

# Extracting Spatiotemporal Flood Information from News Texts Using Machine Learning for a National Dataset in China

Shengnan Fu[1], David M. Schultz[2,3], Heng Lyu[1], Zhonghua Zheng[2,3], and Chi Zhang[1]

[1]School of Infrastructure Engineering, Dalian University of Technology, Dalian, 116024, China
[2]Centre for Crisis Studies and Mitigation, The University of Manchester, M13 9PL, United Kingdom
[3]Centre for Atmospheric Science, Department of Earth and Environmental Sciences, The University of Manchester, M13 9PL, United Kingdom

**Correspondence:** Heng Lyu (lyuheng@dlut.edu.cn)

**Abstract.** Urban floods present a threat in China, demanding an understanding of their spatiotemporal distribution. Current flood datasets primarily offer provincial-scale insights and lack temporal continuity, which leads to a challenge in detailed analysis. To create a consistent national dataset of flood events, this study introduces a machine learning framework by applying online news media as a primary data source to construct a county-level dataset of urban flood events from 2000 to 2022. Using
the Bidirectional Encoder Representations from Transformers (BERT) model, we achieved robust performance in information extraction, with an F1 score of 0.86 and an exact match score of 0.82. Further, a combined model of Bidirectional Long Short-term Memory (BiLSTM) networks with a Conditional Random Field (CRF) layer effectively identified flood locations. Our analysis reveals that the temporal trend of flooded cities in the news-based dataset is similar to the *China Flood and Drought Bulletin*. Furthermore, the consistency of flood events in the news with the typhoon trajectory in two cases, and the connection
between flood occurrences and flood conditioning factors, confirm the accuracy of spatial distribution. The validated news-based dataset analyzes urban floods in China from both temporal and spatial perspectives. First, this dataset shows the seasonal characteristics of flood events, which are concentrated in the summer. From 2000 to 2022, the peak year for floods was 2010, and excluding the influence of peak year, the overall temporal trend of total flood occurrence shows an increase. Spatially, the distribution of floods decreases from southeast to northwest, with Guangxi Province having the highest number of floods.
Additionally, the Yangtze and Pearl River basins are most frequently affected by urban floods. The subtropical climate zone is the most susceptible to flooding. This study provides an automated and effective method for constructing a national flood event dataset and reveals the spatiotemporal characteristics of urban flooding in China.

## 1 Introduction

Floods have been a recurring challenge in China throughout its history, with efforts to manage them spanning four millennia
(Feng et al., 2023; Jiang et al., 2023). Cao et al. (2022) found that China and the United States were the two countries with the highest urban flood exposure, together accounting for approximately 61.5% of the global increase in urban flood exposure. In China, more than three urban flooding events happened annually in 137 cities from 2008 to 2010 (Zhang et al., 2018). In recent years, Zhengzhou had torrential rainfall and subsequent flooding on 21 July 2021, resulting in 380 deaths and direct economic





losses of 168 billion dollars (Dong et al., 2022). In another example, Shenzhen experienced short-duration and extremely heavy
precipitation on 11 April 2019, leading to floods and 11 deaths (Zhang et al., 2023c). Thus, urban flooding is an important risk
factor affecting urban property and public safety in China.

To develop more targeted strategies to mitigate flood damage, understanding the historical urban flood distribution is crucial.
Analyzing the temporal and spatial distribution of floods helps in identifying flood-prone areas. By analyzing the climate and
geographical features of flood-prone regions, what causes the floods can be discovered, leading to possible flood-control strate-
gies (Ahemaitihali and Dong, 2022; Zhang et al., 2023a). Previous studies about the distribution of floods has demonstrated
the reliance on the historical flood datasets (Zhao et al., 2018; Wu et al., 2019; Xu and Tang, 2021).

Although having a database of floods in China is desirable, no single dataset provides comprehensive information about
floods at a scale smaller than the provincial level in China. The existing official Chinese datasets are summarized in Table 1.
However, each of these datasets have their own specific applicability and limitations in different scopes. Notably, the *Annual*
*Report of Chinese Hydrology* is more suitable for basin flooding studies than urban flooding, and it only began publication
in 2021. The *China Flood and Drought Bulletin* lists both urban flooding and basin flooding information but only offers the
spatial distribution of accumulated disaster data at a provincial scale since 2006. The *China Meteorological Disaster Yearbook*
only includes records of severe events that result in significant losses, leading to a smaller number of records of the more
frequent urban floods that cause minor damage. Additionally, the information on the website of the China National Disaster
Reduction Center is relatively detailed and similar to news reports, as both are documented at the time of disaster occurrence.
However, the data collected before 2018 became inaccessible due to a change in the website that followed the creation of the
Emergency Management Department of China in 2018.

In addition to the official Chinese datasets, there are also several natural disaster datasets recording flood events created by
other governments or organizations (Table 2). Each dataset has a different research perspective, which makes it impossible to
create a long-term statistical analysis of historical urban floods in China. The Emergency Events Database (EM-DAT) includes
detailed information on global disaster events, such as timing, location, and losses, but it only includes severe events that cause
damage of a certain scale. The *Natural Disaster Data Book* published by the Asian Disaster Reduction Center is an analysis
of EM-DAT data specifically for the Asian region. The Dartmouth Flood Observatory focuses on global flood events, offering
more detailed information with locations pinpointed to latitude and longitude. However, its records in China are insufficient,
with the total number of flood events recorded from 1985 to the present not exceeding 400 cases. Additionally, both the
Copernicus Emergency Management Service and the Global Flood Monitoring System are based on real-time satellite data
monitoring, which is more suitable for flood forecasting or real-time inundation progression simulation than historical flood
statistical analysis.

The existing datasets clearly show that there are gaps in the urban flood records, lacking continuous records at a finer spatial
resolution, such as at the county or neighborhood level. However, given that China has a land area of approximately 9.6 million
square kilometers (Zhang et al., 2023b), the provincial or prefecture-level administrative regions typically cover large areas
that contain diverse meteorological and geographical characteristics (Wang et al., 2013; Shang et al., 2023). This diversity may
lead to bias in the studies on comparing the flood characteristics of different cities in the same province or analyzing the flood





causes. In contrast, county-level administrative divisions narrow down the scope, offering a more homogeneous perspective
that could improve the accuracy of such flood datasets. Therefore, a continuously updated national urban flood dataset on a
county level that includes records covering at least the past 20 years should be built to bridge the gap.

For the urban flood dataset construction, researchers often supplement governmental data with remote-sensing imagery
(Huang and Jin, 2020; Shahabi et al., 2020) and field-survey data (Eini et al., 2020; Darabi et al., 2021). Remote-sensing
images offer the potential to infer the progression of disasters, but information retrieval through remote sensing is plagued by
uncertainties due to factors such as cloud cover (Datla et al., 2010; Donovan et al., 2019). On the other hand, field surveys
provide highly accurate, first-hand data. However, the process is both time-consuming and labor-intensive (Surampudi and
Yarrakula, 2020; Feng et al., 2022), making it challenging to support the collection of historical flood events on a large scale.
Given these limitations, selecting proper data sources and related processing techniques to collect national urban flood records
remains a challenge.

Against this backdrop, digital news media data emerges as a promising alternative. News data offers timely, authentic, and
extensive coverage of disaster events (Williamson, 2019; Antwi et al., 2022). Moreover, media coverage tends to focus on the
current impact of disasters on humans, infrastructure, and the environment, and on the regions related to the events (Houston
et al., 2012), which are exactly the elements needed to construct a disaster dataset. Some studies about natural hazards have
revealed the power of media data to supplement the shortcomings of traditional natural science data sources (Avellaneda
et al., 2020; Lai et al., 2022). For instance, Yang et al. (2023) analyzed the clustering of multiple natural disasters including
earthquakes, floods, droughts, etc. in China based on news data. Similarly, Liu et al. (2018) used news media data to extract
characteristics of natural disasters, uncovering a significant coexistence of meteorological and geological disasters. Our study
would extend these efforts by developing a county-level historical flood database based on news data, focusing on urban floods.

To extract flood events from news data, event extraction techniques in the field of natural language processing can offer
support (Xiang and Wang, 2019; Olivetti et al., 2020). Early event extraction approaches based on pattern matching (Bui et al.,
2013) have given way to machine learning models that can automatically extract semantic features, offering more nuanced
understanding and flexibility in handling diverse event types (Sha et al., 2016; Liu et al., 2020). Some researchers have proposed
machine reading comprehension methods (He et al., 2018; Farooq et al., 2020), achieving event extraction in a question-answer
format, which can effectively generalize to previously unlabeled event types.

The innovative application of the question-answer format offers an advancement over using solely Named Entity Recognition
(NER) methods for identifying the locations of flood disasters. The NER is an information extraction method for finding
and sorting named entities into pre-defined tags (persons, locations, and organizations). It can automatically extract place
names from texts by analyzing the structure and grammar of sentences. Previous studies on disaster information extraction
commonly adopted NER methods such as the bidirectional long short-term memory network (BiLSTM) combined with a
Conditional Random Field (CRF) layer, which were trained with a large number of place name tags (Kundzewicz et al., 2019;
Yan et al., 2024). The NER methods identify all place names in the text regardless of their contextual background, leading to
the recognition of place names not related to the flood disaster. For instance, in the sentence "Many volunteers from Beijing
went to the disaster area in Shanghai", "Beijing" is not a disaster-affected location, but it would still be output by the NER





model as a result. In contrast, the question-answer model could distinguish between disaster-affected and unaffected areas
by posing targeted questions (Sun et al., 2021; Zhang and Zhang, 2023). Currently, the best-performing machine reading
comprehension models are pre-trained language models (Yoon et al., 2019; Li et al., 2021). The pre-trained models capture
deep semantic relationships between sentences through extensive semi-supervised training on a large corpus, achieving an
accurate understanding of contextual information, thereby completing question-answering tasks.

Among the pre-trained models, the Bidirectional Encoder Representations from Transformers (BERT) (Devlin et al., 2018)
is a seminal model which has proven to be effective and widely adopted. The strength of the BERT model lies in the innovative
use of bidirectional pre-training on large-scale unlabeled text data like Wikipedia and books, allowing it to be effective for
information extraction across diverse fields (Xiong, 2020; Suwaileh et al., 2020). Some cases illustrate the accuracy of BERT
in identifying events from unstructured text, identifying speech for transcription, and building a question-answer system with
a small number of samples (Wang et al., 2021; Huang et al., 2021). These successful applications show the potential of BERT
to identify the flood events information from news text data.

The capabilities of the language model, combined with the reliability of news data, provide us with an opportunity to
construct a new flood event datasets to address the absence of datasets at the county level. Therefore, this present study
aims to develop a national county-level urban flood dataset from 2000–2022 based on news data through a machine learning
framework. The performance of the BERT model in the field of flood disaster knowledge is examined and the spatiotemporal
characteristics of urban floods in China based on news records are analyzed.

The remainder of this paper is organized as follows: Section 2 introduces the data used including a Chinese text dataset
used to train the BERT model, as well as news data and validation data. Section 3 describes the processes of flood information
extraction by the BERT model and flood location recognition using a BiLSTM network combined with a CRF layer. Section
4 shows the performance of the BERT model, as well as the accuracy evaluation of extracted flood information and the spa-
115 tiotemporal distribution of the flood information. Section 5 discusses the findings and limitations of the study. Finally, Section
6 outlines the summary of our key contributions and results.

## 2 Data Preparation

Three kinds of datasets were used in this study. Section 2.1 describes a Chinese machine reading comprehension data called
CMRC2018 (Cui et al., 2018) used to train the BERT model. Section 2.2 explains the news data used to extract information on
urban flooding. Section 2.3 interprets the validation data selected to assess whether the extracted flood information is accurate.

### 2.1 CMRC2018

The CMRC2018 dataset is a span extraction dataset for Chinese machine reading comprehension, consisting of nearly 20000
real-world questions annotated by human experts on Wikipedia paragraphs. The task of reading comprehension is to obtain
the corresponding answer from the given context and question, and span extraction indicates that the content of the answer is
all in the context, and the length of the span is determined by the distance between the start and end positions of the answer.





**Table 1.** Summary of official flood disaster statistics reports

| Name | Period | Flood Records | Update Frequency | Source |
|---|---|---|---|---|
| *Annual Report of Chinese Hydrology* | 2021– | Records of basin/river floods in various provinces and cities | Annual | Ministry of Water Resources of the People's Republic of China |
| *China Flood and Drought Bulletin* | 2006– | The flood disaster situations in various provinces include population, economic, and crop losses | Annual | Ministry of Water Resources of the People's Republic of China |
| *China Meteorological Disaster Yearbook* | 2004– | Record criteria as events causing over 50,000 hectares of agricultural damage, 10 deaths, or 14 million USD in direct economic losses | Annual | China Meteorological Administration |
| Reports on official website of China National Disaster Reduction Center | 2011– | Detailed records of the time, location and damage of flood events (Data prior to 2018 is not available) | Real-time | National Disaster Reduction Center of China |

The dataset used in this study contains 2282 training samples. Each sample consists of a group (C, Q, A), where C represents context, Q represents questions, and A represents answers. The answer to each question should be a span extracted from context. Figure 1 shows an example including a context describing a model's resume, as well as a question about the content of the context and the corresponding answer. In this study, the CMRC2018 dataset was used to fine-tune the BERT model to adapt to the Chinese machine reading comprehension task.





**Table 2.** Summary of global natural disaster datasets

| Name | Period | Flood Records | Update Frequency | Source |
|---|---|---|---|---|
| The Emergency Events Database (EM-DAT) | 1900– | Time, location and damage of global flood events that resulted in a certain number of deaths or economic losses | Annual | Centre for Research on the Epidemiology of Disasters |
| *Natural Disaster Data Book* | 2002– | Statistical and analytical perspectives of flood events in Asia (data retrieved from EM-DAT) | Annual | Asian Disaster Reduction Center |
| Dartmouth Flood Observatory (DFO) | 1985– | Time, location and extent of global flood events using satellite observations | Continuously but irregularly updated | University of Colorado Boulder |
| Copernicus Emergency Management Service (CEMS) | Real-time | Ongoing and upcoming flood events information from satellites to support flood forecasting at national, regional and global levels | Real-time | European Union Copernicus Programme |
| Global Flood Monitoring System (GFMS) | Real-time | Flood inundation extent and depth based on precipitation satellite data and flood model simulation | Every 3 hours | University of Maryland and NASA |

## 2.2 News Data

The news used in this study was collected from two Chinese newspaper databases covering the whole of China, including WiseNews (https://wisenews.wisers.net/) and the newspaper database of the China National Knowledge Infrastructure (CNKI) website (https://navi.cnki.net/knavi/newspapers/index). WiseNews is a full-text news database that provides access to more than 600 newspapers, magazines, and websites from China. Its news coverage dates back to 1998 and is updated daily to the present. The CNKI database is sourced from more than 500 important newspapers in China covering the period from 2000 to the present. The main difference between these two databases is that CNKI has certain selection criteria for selecting academic and informative documents and therefore returns different amounts of results for the same searching keywords. The newspaper sources collected by the WiseNews database could cover the newspaper sources collected by CNKI.



---

[Context]
Anya Russell is a model from St. Petersburg, Russia. She was the runner-up of the 10th season of the American Super Model Rookie Contest. At the age of 17, Anya participated in informal fashion shows for brands such as Chanel, Louis Vuitton, and Fendi. Anya said during the semi-finals interview that she is passionate about the modeling industry, so she participated in the American Super Model Rookie Competition. She performed outstandingly in the competition, having been nominated for first place five times and achieving her best performance ever (2.64) in average order

[Question]
What competition did Anya Russell participate in and win the runner-up?
[Answer]
Season 10 of the American Super Model Rookie Contest

---

**Figure 1.** An example in CMRC dataset.

CNKI was selected as the primary source to manually extract the spatiotemporal information of the floods due to its high concentration of academic and informative content. In total, 2730 pieces of news from 2000–2021 were collected by setting the keywords of the subject as ("flood" OR "flood disaster") and the keywords in any full text as ("city" OR "county" OR "district"). After a manual review to remove duplicates and irrelevant entries, including those referring to flash floods which occur suddenly in mountainous areas and are not the focus of this study, the final dataset consisted of 253 relevant news articles.

These relevant news articles were then segmented into paragraphs and reorganized into 633 distinct samples. Among them, 503 samples were used to fine-tune the BERT model, alongside data from the CMRC2018 dataset, enhancing the model's ability to accurately extract flood disaster information. The remaining 130 samples served as a test set to evaluate the model's performance.

       Building on the method successfully applied to CNKI data, the WiseNews database subsequently employed the fine-tuned
BERT model for analyzing flood-related news. To refine the search and improve data quality, terms "floods and beasts" and "flash flood" were excluded due to their tendency to retrieve unrelated news based on the experience of processing CNKI news data. "Floods and beasts" is an idiom in Chinese, which is often used as a metaphor for frightening things. This idiom contains the word "flood", so some unrelated news items that used it to describe other disasters or bad phenomena may be included in the search results. Therefore, the refined search strategy used was ("flood" OR "flood disaster") AND ("city" OR "county" OR





"district") NOT ("flash flood" OR "floods and beasts"). This search returned a total of 46118 news items, with a time frame
from 2000–2022.

## 2.3  Validation Data

The *China Flood and Drought Bulletin* (Table 1) was used to assess the accuracy of flood events reported in news texts.
Although these bulletins provide annual flood event and loss data at the provincial level, they are not directly comparable with

the county-level flood information derived from news sources in this study. However, bulletins from 2006 to 2018 include data
on the number of cities affected by floods, which were used to verify the city-level flood occurrences identified in this study.

To further evaluate the spatial accuracy of flood event reporting, this study analyzed the coverage of typhoon-related urban
flooding events during two specific typhoon periods. Stronger typhoons tend to cause more extreme rainfall, which leads to
urban flooding (Chan et al., 2021; Liu et al., 2021). If the flood-affected areas reported in the news are consistent with the

typhoon trajectories and potential impact areas, this correlation would confirm the spatial accuracy of the news-based flood
data. The typhoon path and intensity information used in this study were sourced from the National Institute of Informatics
(NII) in Japan (https://kaken.nii.ac.jp/).

Additionally, this present study selected two primary flood conditioning factors that quantify topography and precipitation
to evaluate the characteristics of news-derived flood distribution. According to hydrological research, areas with more precip-

itation and lower-lying topography are more prone to urban flooding (Guo et al., 2021; Chen et al., 2023; Wang et al., 2019).
If the flood locations identified from news reports tend to have higher precipitation levels and are concentrated in lower-lying
areas, this would be consistent with established hydrological principles. In this study, the average daily precipitation from 2000
to 2022 for each county was calculated based on the ERA5-Land dataset. Topography was quantified using elevation data (Zhu
et al., 2023) from the Shuttle Radar Topography Mission (SRTM) with a 90-meter grid spacing.

## 3  Methods


In this study, a machine learning framework was adopted to extract spatiotemporal flood event information from news texts.
Section 3.1 describes how flood information was extracted from related news texts using BERT model through a question-
answer format. Section 3.2 shows the flood location recognition methods based on a NER model. The overview of the
framework is illustrated in Figure 2.

## 3.1  Flood Information Extraction


The BERT machine reading comprehension technique was used to extract the time and location of the disaster from news texts.
Section 3.1.1 introduces the data preparation and Section 3.1.2 presents the BERT model construction and application. Section
3.1.3 explains the evaluation metrics for model performance.



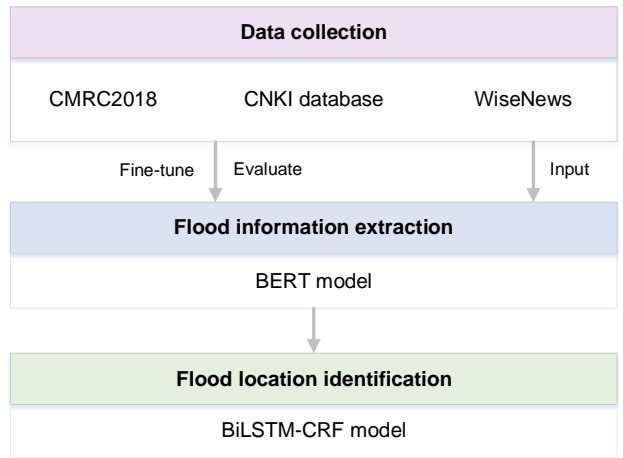

**Figure 2.** The framework of flood information collection and extraction.

### 3.1.1 Data Preparation

Data pre-processing in this study included both data cleaning and splitting. Initially, news documents downloaded from the database contained extra characters such as copyright information, news web links, and numerous empty lines, which were irrelevant to the news content. These irrelevant content was efficiently removed using the regular expression ("re") module in Python. Moreover, given that news documents can be lengthy and may exceed the input length limitations of the BERT model, each news text was divided into multiple samples.

After pre-processing, flood news texts were manually annotated to align with the CMRC2018 data format. For the training samples, the content of each news article was labeled as the context, and three questions were proposed per context, addressing the flood information. The questions included the following "Question 1: What natural disaster occurred? Question 2: When did the disaster occur? Question 3: Where was the disaster located?" Answers were manually annotated based on the news content. Finally, 503 samples from CNKI news were formatted into a training set, of which 80% were combined with the

CMRC2018 dataset to fine-tune the model, while the remaining 20% of the samples were used as a validation set to adjust hyperparameters. The remaining CNKI samples and all the WiseNews samples were processed into test samples mirroring the training sample format, except that the answers were not annotated. For the CNKI test samples, the answers generated by the model were also manually evaluated to assess the accuracy. Once the model's efficacy was confirmed through these assessments, it was subsequently applied to the WiseNews samples.

### 3.1.2 BERT Model Construction and Application


BERT, which is designed to pre-train deep bidirectional representations from large unlabeled data sets (Devlin et al., 2018), was introduced to extract flood information in this study. The model structure is a multi-layer bidirectional transformer encoder,





which is an attention mechanism that learns contextual relations between words. Unlike other traditional bidirectional language models where the contextual representation of each token is a concatenation of the forward and reverse representations, the
transformer encoder reads the entire sequence of words at once. There are two procedures for constructing a BERT model: pre-training and fine-tuning.

For pre-training, the model is trained on unlabeled text data using two unsupervised training strategies. First, BERT proposes a masked language model, inspired by the cloze task (Taylor, 1953), in which 15% of the input tokens are randomly masked by a special label [mask], and these masked tokens are then predicted. Secondly, BERT adopts next-sentence prediction into the
training process. A binary next-sentence prediction task is pre-trained to enable the model capable of understanding sentence relations. The model takes in pairs of sentences as input and attempts to identify if the second sentence within the input pair is the subsequent one in the original context.

Following pre-training, the downstream task for fine-tuning in this study involves extracting answers from the context based on posed questions. In typical applications, BERT models predict where answers start and end in a text by adding a classification
layer known as softmax. The process begins with tokenization, where the text is divided into smaller units called tokens, which can represent words, phrases, or punctuation marks. Then, the model calculates the probability of each token being the start and end of the answer, separately. This allows the model to identify a continuous text fragment as the answer, which is suited for scenarios where a single optimal answer is required. However, the locations affected by a flood event are generally not unique, and the descriptions of multiple disaster areas in the news may be scattered in discontinuous statements. This requires
a method that can extract multiple answers for our study.

To enhance the model's ability to manage multiple answers, a BIEO (Beginning, Inside, End, Outside) tagging layer is integrated into the input (Li et al., 2019). This modification enables the model to predict one of four possible tags for each token: "Beginning" for the starting token of an answer, "Inside" for intermediate tokens within an answer, "End" for the final token of an answer, and "Outside" for tokens that do not form part of the answer. This approach allows the model to recognize
multiple independent answer fragments within a paragraph, since each answer fragment is marked by an explicit start and end and connected by an intermediate tag. This enhanced capability makes the model exceptionally suitable for complex, multi-answer scenarios like analyzing fragmented disaster reports. Figure 3 shows the overall structure of the model used in this present study. Here, the context and query processed through the BIEO tagging layer, and their semantic information is learned via the BERT structure. Finally, the probabilities that each token belongs to BIEO of the answer are determined through a linear
layer and softmax function. The BERT-base model was fine-tuned for three epochs with a learning rate of $5 \times 10^{-5}$ and a batch size of 8, which were determined to be the most effective combination among the tested settings. Results of other combinations can be found in Table A1 in Appendix.

To build the set of urban flooding events, the output of the BERT model needs to be further confirmed and collated. The first step was to check whether the news contained a flooding event. If the answer to Question 1, "What disaster happened," contains
the keywords "flood" or "flood disaster" and does not contain the words "will," which indicates the forecast, the news sample is considered to describe a flood disaster event. As the long text news was split into different samples in the pre-processing procedure, the news could be confirmed as long as one of the answers of multiple samples belonging to the same news met the





conditions. For time information, once the answer to Question 1 was confirmed to be a flood event, then the answer to Question 2 was the time of the flood, and the year information is the year of the news release if the answer did not include the year. For location information, similar to confirming time information, if the answer to Question 1 is confirmed, then the answer to Question 3 contains the flood locations.

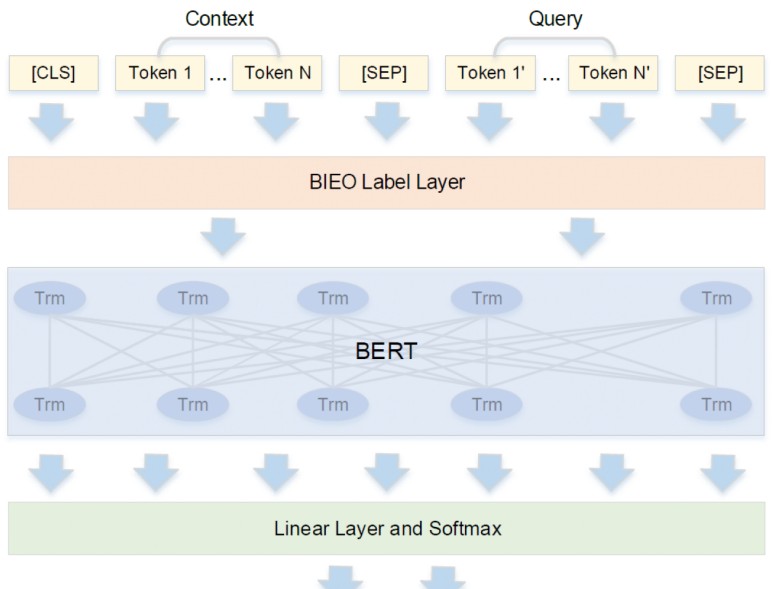

**Figure 3.** The structure of the BERT model proposed in this study. [CLS] and [SEP] are special markers that identify the beginning and end of the input text. [CLS] stands for "classification", and adding a [CLS] token at the beginning of the input text allows the model to learn a representation for the whole sentence. [SEP] stands for "separator", used mainly to separate different sentences or pieces of text. The Trm inside BERT represents the transformer architecture.

### 3.1.3 Evaluation metrics

In this study, two evaluation metrics were used to assess the effectiveness of flood information extraction. These metrics are common across a number of disciplines, yet use different names (Brooks et al., 2024). Initially, the identification of flood events was treated as a classification problem, using precision, recall, and the F1 score to evaluate the accuracy of predictions. The metrics were calculated as:

$$Precision = \frac{TP}{TP + FP} \tag{1}$$

$$Recall = \frac{TP}{TP + FN} \tag{2}$$






$$F1score = \frac{2 \times Precision \times Recall}{Precision + Recall} \tag{3}$$

For the computation of these indexes, Table 3 explains the classification outcomes in terms of True Positives (TP), False Positives (FP), False Negatives (FN), and True Negatives (TN). Among the metrics, precision is defined as the ratio of correctly predicted flood news to all predictions labeled as flood events. Recall measures the proportion of correctly predicted flood news out of all actual flood events. F1 score combines precision and recall scores as the harmonic mean, providing a balanced view of the model's performance. A higher F1 score indicates superior model performance.

**Table 3.** Classification outcomes for flood identification

| True Condition | Prediction Result | Label |
|---|---|---|
| Flood | Flood | True Positive (TP) |
| Flood | Non-flood | False Negative (FN) |
| Non-flood | Flood | False Positive (FP) |
| Non-flood | Non-flood | True Negative (TN) |

Furthermore, for the extraction of flood event information, two matching criteria were applied (Rajpurkar et al., 2016). The first index is called exact match, which measures the matching degree between the prediction and ground truths. The score is 1 for the EM of both the time and location information extracted. Otherwise, the score is 0. There is usually more than one disaster location in one flood event and maybe the model can output several but not completely accurate locations. Therefore, a fuzzy match was used to evaluate the location extraction using precision, recall, and F1 score. Unlike the classical formula, the precision and recall were calculated as:

$$Precision = \frac{P}{M} \tag{4}$$

$$Recall = \frac{P}{N} \tag{5}$$

Where $P$ represents the number of accurately extracted flood locations, $M$ is the total number of predicted flood locations and $N$ is the total number of actual flood locations observed in the texts. The F1 score was computed using the classical formula as Equation 3, providing a balanced view of the quality and completeness of the predicted locations.

## 3.2 Flood Location Recognition

In most flood location answers, there is not only the name of the place itself but also the characters before or after it (e.g., "it occurred in Hangzhou"). Therefore, a BiLSTM–CRF model was adopted to further extract the place names from the answers.



The model was trained on the Microsoft Research Asia corpus, which is an available data set for the Chinese NER. The characters in the Microsoft Research Asia corpus were tagged as named entities representing persons, locations, and organizations. The model framework used in this present study is detailed in Fu et al. (2022).

After identifying the flood locations, it was essential to verify and revise the list of places in accordance with the latest national administrative divisions. This involved updating any names of districts or counties that had been renamed to reflect the most current terminology. After that, flood locations were matched with the administrative division shape file and visualized using ArcGIS.

## 4    Results

The results section consists of three primary parts: Section 4.1 evaluates the BERT model's performance using different metrics. Section 4.2 shows the accuracy of the number and spatial distribution of floods extracted from news using different categories of validation data. Section 4.3 presents the content of the urban flood dataset constructed, highlighting both the temporal distribution in the volume of related news articles and flood events and the spatial distribution of floods across county regions, as well as various basin and climate zones.

### 4.1    The performance of the BERT model

The effectiveness of flood-event recognition and flood-information extraction is presented in Table 4. The BERT model demonstrates excellent performance in identifying whether an event is a flood, achieving an impressive F1 score of 0.98. On the other hand, the overall performance of flood-information extraction is satisfactory, with an F1 score of 0.86 for location extraction and an EM of 0.82 for both time and location extraction. The high F1-score and EM values (over 0.80) demonstrate that integrating domain-specific knowledge enabled the model to respond to questions about flood information.

**Table 4.** The performance of the BERT model (EM index was not applied to evaluate event identification)

|                               | Precision | Recall | F1-score | EM  |
| ----------------------------- | --------- | ------ | -------- | --- |
| Flood-event identification    | 0.98      | 0.98   | 0.98     | N/A |
| Flood-information extraction   | 0.96      | 0.78   | 0.86     | 0.82 |


### 4.2    Accuracy evaluation of the flood information

To validate the news-based flood dataset, a comparative analysis was conducted using records from the *China Flood and Drought Bulletin*. The comparison of annual flooded cities from the *China Flood and Drought Bulletin* with those identified in news sources between 2006 and 2018 is displayed in Figure 4. Detailed yearly comparisons are summarized in Table A2 in

Appendix. The similarity between these time series indicates that the news-based method provides a reasonable approximation





of historical flood-event records. However, the difference also suggests a bias, especially in the years that experienced peak flood occurrences.

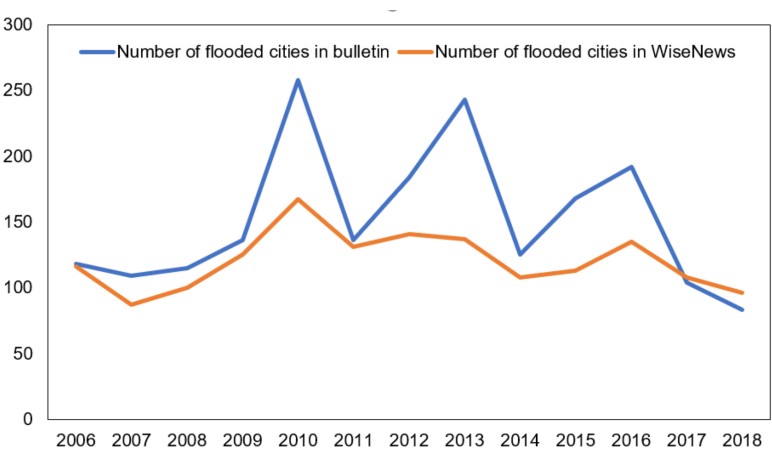

**Figure 4.** The comparison between the time series of number of flooded cities extracted from news and *China Flood and Drought Bulletin* for each year.

The bias between news-based flood data and the *China Flood and Drought Bulletin* may be attributed to media attention. The news-derived flood distribution was compared with regional Gross Domestic Product (GDP) distribution to assess whether economic development introduces a bias in media attention. By clustering the regional GDPs of each county using the K-means method, all counties were classified to 3 groups. Figure 5 presents the box of normalized flood occurrence and GDP values for each group, which reveals a trend of higher GDP regions reporting more flood events. This trend suggests that economic development influences media coverage and public attention. The research by Lu et al. (2022) also indicates that economically developed and urbanized cities receive more media attention. Additionally, media attention towards urban floods is often driven by the severity of the events. For example, Anhui with low media coverage has experienced huge cumulative economic losses from floods in the *China Flood and Drought Bulletin* over the years. This lack of attention is not due to single disaster, but rather a series of smaller floods affecting the area. This selective reporting leads to a lower occurrence in news-based dataset than the *China Flood and Drought Bulletin*.

Consider that the bias sourced from media coverage may also affect the spatial distribution of floods, the subsequent analysis focuses on verifying the accuracy of spatial distribution. Due to the lack of direct spatial distribution information on floods at the county level across China, two kinds of indirect data proven to have certain relationships with flood events, have been investigated. First, this present study focused on a targeted verification of the news extraction method by examining its performance during two severe typhoon events. The reason for selecting Typhoons Lekima and In-fa is that both events caused severe economic losses and casualties and had an extensive spatial impact, spanning multiple provinces. Typhoon Lekima, a super tropical cyclone that originated from the Western North Pacific in 2019, was one of the costliest natural disasters in China, causing 14 million victims with at least 71 deaths and costing 9.22 billion dollars in damages (Qi et al., 2021). It made landfall



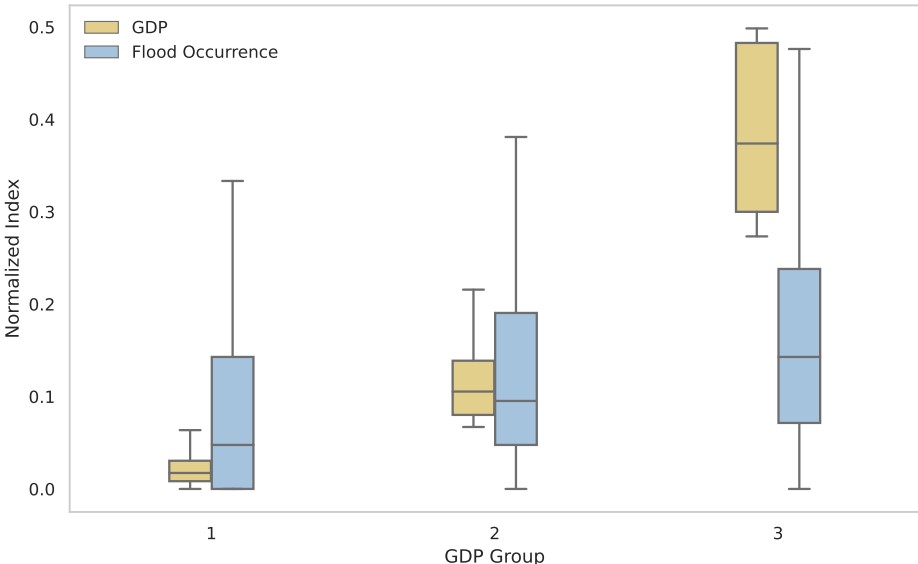

**Figure 5.** The box of flood occurrence and GDP by GDP groups. The vertical axis shows the value after normalizing GDP and flood occurrence between 0 and 1.

in Wenling, Zhejiang Province, on 10 August, then slowly moved north along China's east coast, making a second landfall in Qingdao, Shandong Province, on 11 August, before finally dissipating in the Bohai Gulf on 13 August (Zhou et al., 2022). This typhoon lingered over the land for 44 hours, causing heavy rainfall in many areas, particularly in Shandong where the maximum accumulated rainfall exceeded 400mm (Gao et al., 2023). Figure 6a shows the route of Lekima, along with flood events extracted from the news during the typhoon's progression. The flood locations between 10 and 13 August were closely dispersed on both sides of the Lekima's trajectory, and the sequence of occurrence was also consistent with the time of the typhoon trajectory. Another typhoon in 2021, called In-Fa, also demonstrated this consistency, as shown in Figure 6b. On 25 July, Typhoon In-fa made landfall in Zhoushan, Zhejiang. On 26 July, it made a second landfall in Pinghu, Zhejiang, then gradually moved northward, weakening into a moderate cyclone, and was dissipated on 30 July. The flood locations reported in the news during the typhoon's movement were consistent with the route of In-Fa. These findings indicate the effectiveness of the news-based method in capturing the distribution of flood events in the aftermath of major storms.

Another indirect evaluation of the spatial distribution of flood events at the county level across China was exploring the relationship between flood occurrence and flood conditioning factors, including precipitation and elevation. Precipitation is a direct contributor to urban floods and heavy rain leads to surface water accumulation. Elevation influences how water flows and accumulates in an area, with lower areas more prone to flooding after heavy rains due to water flowing downward. In this study, the average daily precipitation from 2000 to 2022, derived from the precipitation grids within each county-level administrative boundary, was selected. Figure 7 displays the distribution of elevation and precipitation for flood-affected locations extracted in this study. The intensity of the data points' color reflects the total number of flooding events at those locations, with darker col-

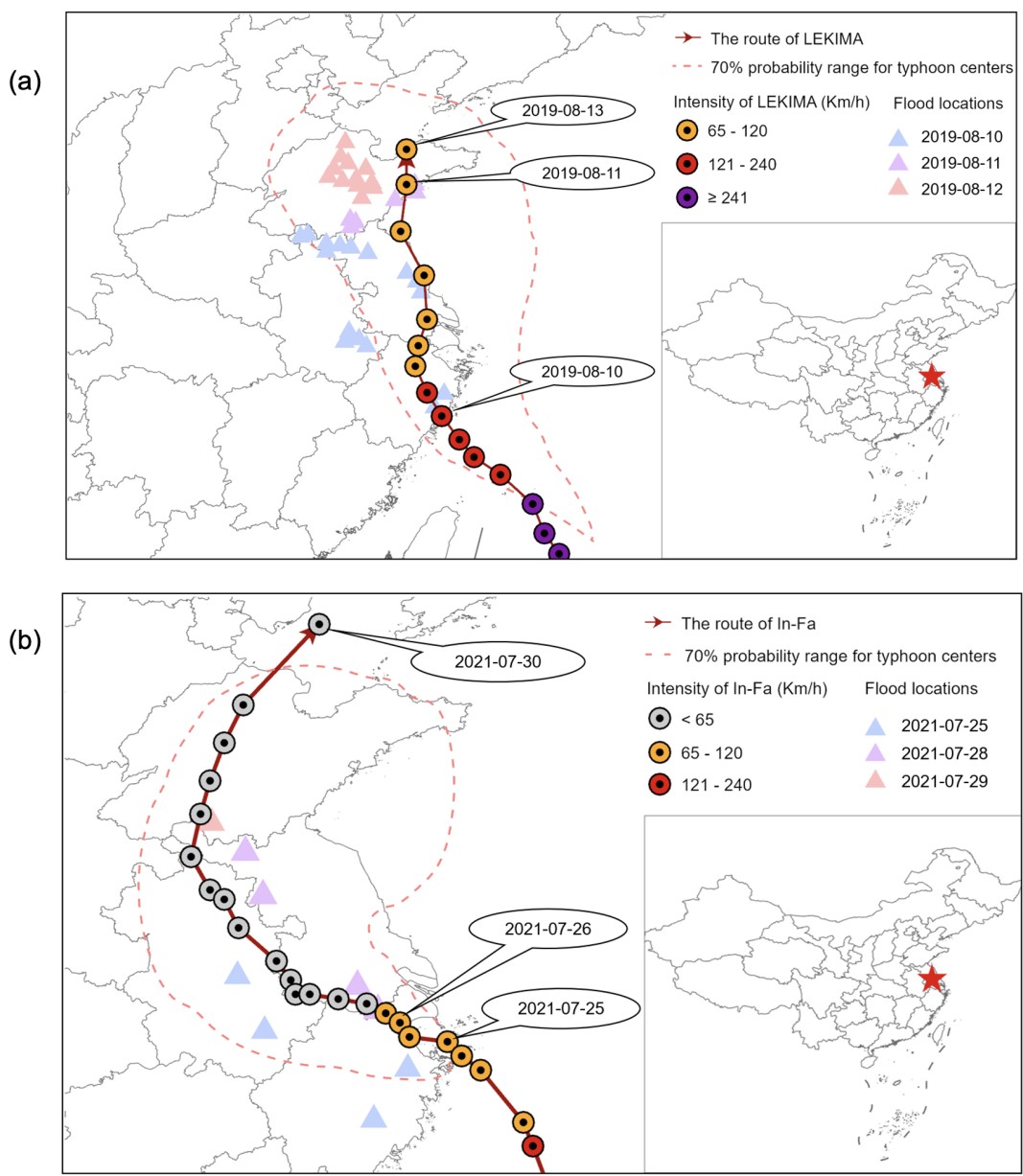

**Figure 6.** The typhoon routes and related flood locations. The red solid line in the figure represents the actual path of the typhoon movement, while the range within the dashed line is the range of the typhoon center predicted at the beginning of the typhoon landing, and the forecast accuracy is 70% probability. (a) Typhoon Lekima; (b) Typhoon In-Fa.

ors denoting a higher occurrence. The points with darker colors, which correspond to high-occurrence flood-affected locations, are predominantly situated in areas of higher precipitation and lower elevation. The spatial distribution of news-sourced floods is concentrated in these highly sensitive areas, demonstrating consistency with established hydrological principles. Despite the





systematic bias caused by media attention, two indirect validation methods above indicate that the spatial distribution of floods extracted from the news is reasonably accurate.

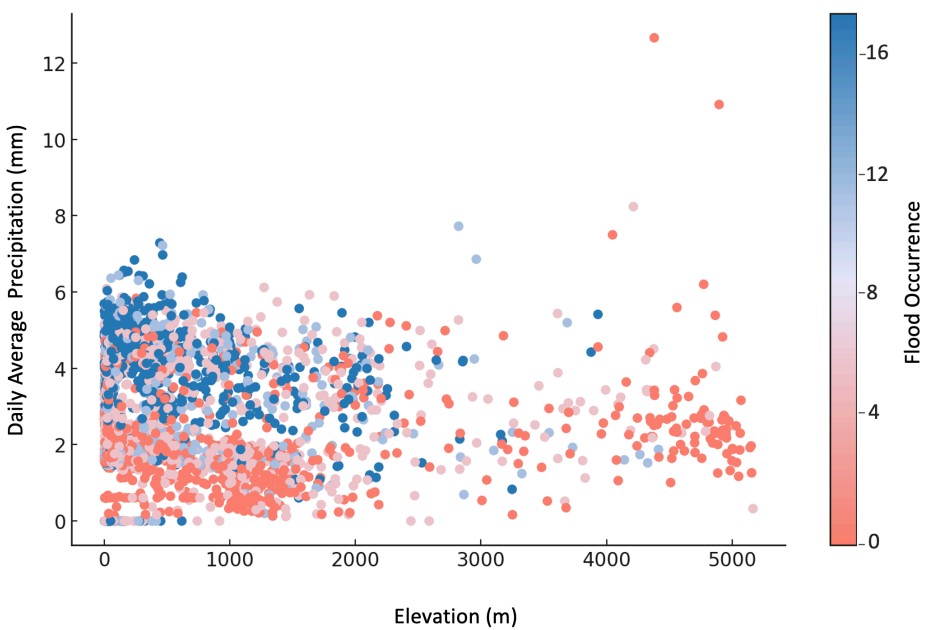

**Figure 7.** The distribution of daily average precipitation and elevation of each county and related flood occurrence. The x-axis and y-axis show the elevation and precipitation values for each county, respectively, and the color of the dots shows the corresponding number of flood occurrence for each county according to the color scale.

## 4.3    The urban flood dataset

After using a mixed strategy on the accuracy evaluation of the news-based flood information, this section introduces the temporal and spatial distributions of the verified national urban flood dataset. The dataset records urban flood events reported in news articles from 2000 to 2022, including the timing of these events at the monthly level and the affected areas at the county level. Currently, the dataset records a total of 2048 counties affected by flood disasters during these years, with a total occurrence of 7559 times. Section 4.3.1 presents the temporal distribution, which shows the monthly changes in the number of news articles about urban floods and the number of flood events in county-level regions. Section 4.3.2 shows the spatial distribution of the total number of floods in county-level regions, along with the trend of floods across different basins and climate zones.



### 4.3.1 Temporal distribution of flood events

Figure 8 shows the temporal distribution of historical flood locations in China from 2000 to 2022. It displays the flood occurrence each month of every year, with darker colors indicating more frequent floods. The comparison of monthly flood occurrences across different years shows little variation, which reveals an evident seasonal cycle. The summer months (June–August) experienced a higher occurrence of flooding, accounting for 74% of total flooding events. In contrast, the winter season (December–February) which recorded few flood events, only accounted for 1%. This is impacted by the climate's tendency for

precipitation to be concentrated in the summer. The seasonal characteristics are consistent with the findings of Xu and Tang (2021)'s analysis of multi-disaster data from 2011 to 2019 in China, which showed that floods predominantly occurred from April to September, with the highest frequency in July. Our study found June to be the most frequent month for floods, possibly due to the different study periods.

Over these years, the total number of flood events per year did not show a continuous increase but rather a rise and fall, with a

peak in 2010. This pattern is consistent with the trends observed in the *China Flood and Drought Bulletin* (Figure 4). Data from the early 2000s show a notably low occurrence of flood events due to a low volume of news data. Excluding this minimum, the year 2003 emerges as having the least number of flood events. The year 2010 stood out with the highest occurrence of floods, significantly impacting 525 counties. After 2010, excluding the impact of peak values, the overall trend shows a slight increase again.

The number of flood-related news reports each month of every year is displayed in Figure 9, showing characteristics similar to the temporal distribution of flood events. The year 2010 was marked by an exceptionally high volume of news, particularly from June through August, and a notable surge in May. July frequently emerges as the month with the highest reportage, emphasizing the heightened flood risk during this period.

### 4.3.2 Spatial distribution of flood events

The spatial distribution of accumulated flood frequency by the county level in China from 2000 to 2022 is shown in Figure 10, whereas the frequency has been classified into 5 levels by the natural breaking method (0–1, 1–5, 5–9, 9–13, 13–22). Level 2 accounts for the largest percentage (50%) followed by Level 1 (29%) and Level 3 (16%), while Levels 3 and 4 are also extremely low (4% and 1%). Most of the county areas in China have experienced flood events. Regions experiencing the most frequent floods include the level 5 and level 4 areas, which are located in the southwest interior of China, such as Sichuan Province;

the northwest, including Lanzhou Province; and the southern coastal regions like Guangdong Province, Fujian Province, and Guangxi Province. This pattern is similar to the findings of Liang et al. (2019), whose analysis of historical meteorological data revealed a higher concentration of floods in southern provinces including Guangdong, Hainan, Guangxi, and Fujian.

The province with the most floods was Guangxi, with 960 reported flood events. Guangxi is located in the southwest of China, between $20°54'$–$26°23'$N, $104°29'$–$112°04'$E, in the South Asian tropical monsoon climate zone (Nie et al., 2012; Gao

et al., 2020b). The region is characterized by high temperatures, a long summer, short winter, and distinct wet and dry seasons. The annual average temperature is 22.5°C, and the annual average precipitation is 1806 mm (Qiu et al., 2021). Studies indicate




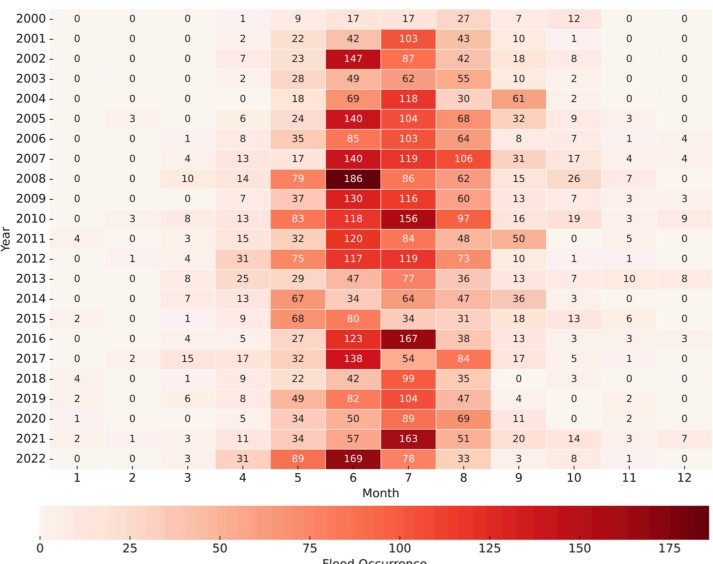

**Figure 8.** A heat map showing in each cell the number of flood occurrence for each month and each year.

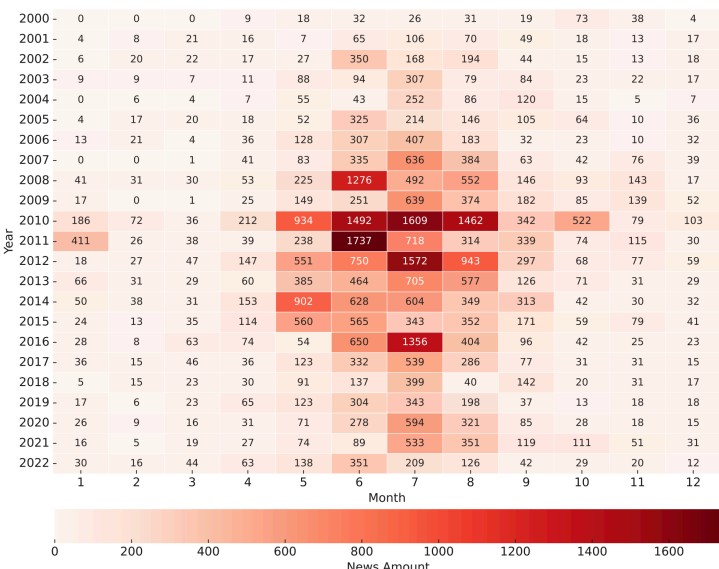

**Figure 9.** A heat map showing in each cell the number of news reports for each month and each year.

that Guangxi experienced flood disasters caused by heavy rainfall frequently (Li et al., 2023; Ma et al., 2023a). Qin et al. (2021) observed an increase in flood hazards in Guangxi since the 1990s and predicted that future precipitation in the region tends to intensify.





To further examine the spatial characteristics of historical flood events in China, an analysis was conducted on the distribution across various basin divisions and climate zones, using the Theil–Sen estimator for robust trend detection. The Theil–Sen estimator is a non-parametric method that calculates the slope among points, offering a robust way to track changes in flood occurrences over time (Kemter et al., 2023). The spatial distribution and trend analysis of flood occurrence within China's river basins shown in Figure 11 reveal a distinct pattern: a higher occurrence in the east than in the west, and a higher occurrence in

the south compared to the north. The Yangtze River Basin (III) and the Pearl River Basin (V) exhibit a notably higher frequency of flooding, with 3617 and 1794 events respectively, suggesting a pronounced vulnerability to flooding within these populous and economically pivotal regions. In contrast, the Continental Basin (I) and the Southwest Basin (II) display a relatively lower frequency of flood events. Trend directions are also reflected in this figure, symbolically represented by triangles, where the orientation indicates an increasing or decreasing trend in flood frequency. Notably, although the Yangtze River Basin (III) and

the Pearl River Basin (V) exhibit the highest frequency of floods, the overall trend is decreasing, whereas other basins show different levels of increasing trends. Furthermore, adjacent basins do not necessarily share similar trends; for instance, the Yellow River Basin (IV) presents a more pronounced increasing trend, whereas the adjacent Huaihe River Basin (VII) exhibits a nearly zero slope, implying the potential influence of localized environmental or anthropogenic factors.

    Figure 12 offers the spatial distribution and trend of flood occurrence concerning China's diverse climate zones. The sub-

tropical zones, specifically the South Subtropical Zone (IV), North Subtropical Zone (V), and Central Subtropical Zone (III), are characterized by a higher frequency of flooding, with 1336, 1597, and 3222, respectively. These zones, which endure the bulk of flood events, exhibit contrasting trends. Whereas the South Subtropical Zone (IV) indicates a trend towards an increase in flood events, the Central Subtropical Zone (III) and the North Subtropical Zone (V) show a slight decrease, highlighting the non-monotonic nature of flood trends across climatic gradients. In addition, the South Temperate Zone (I) displays a relatively

higher frequency and an increasing trend of flood events. Conversely, the North Temperate Zone (VII), the Central Tropical Zone (IX), and the Plateau Climate Zone (II) experience a minimal frequency of flood events and a near-zero trend coefficient, suggesting a relatively lower impact of flooding.

## 5   Discussion

The dataset constructed in this study serves as a supplement to official datasets, particularly in filling the gaps regarding the

distribution of flood event distribution at the county level. It offers robust data support for future research on urban flooding, enabling a detailed understanding of flood dynamics across different areas. Notably, the temporal and spatial distribution characteristics revealed consistent with the findings of previous studies, offering corroborative evidence on the dynamics of urban flood events.

    The temporal distribution revealed an increasing trend in urban flood events over time when the peak value is excluded

from consideration. The increase of urban floods is mainly due to an increase in extreme rainfall events coupled with the rapid urbanization (Wu et al., 2021; Cheng et al., 2020). On the one hand, urban floods are intensified by changes in precipitation patterns due to global warming, which increases both the intensity and frequency of rainfall (Kong et al., 2021). Kundzewicz





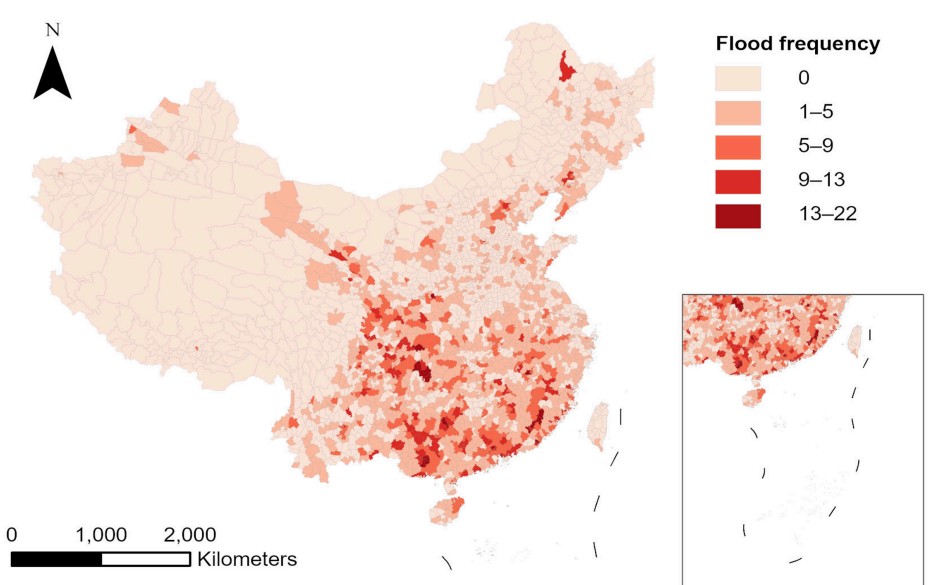

**Figure 10.** The accumulated flood occurrence by the county level in China from 2000–2022.

et al. (2019) found an increase in the annual number of heavy-rain days (daily precipitation ≥ 50 mm) in China from 1961 to 2017. On the other hand, rapid urbanization in China intensified the imperviousness of urban surfaces leading to the increasing

of urban flood risk (Wang et al., 2022; Du et al., 2015). Urbanization leads to changes from natural surfaces such as soil and vegetation to impervious materials like asphalt, which prevents rainfall from effectively seeping into the ground, accelerating the speed of runoff and thus increasing flood risk (Huong and Pathirana, 2013; Luo and Zhang, 2022). Notably, China has been the largest contributor to the global expansion in high-hazard settlements in flood zones, reflecting the impact of its rapid urban growth (Rentschler et al., 2023).

Shifting focus to spatial distribution, this paper discusses the flood distribution pattern from two perspectives including basins and climate zones. Overall, the southern and eastern regions of China are more affected by flooding, which is consistent with previous findings in flood risk and peak precipitation distribution (Sun et al., 2024; Gao et al., 2020a). Specifically, the Yangtze and Pearl River basins are the most frequently flooded areas, which is similar to the findings of Fang et al. (2021). The Yangtze and Pearl River basins are two of China's most economically developed and densely populated regions (Pan et al.,

2023; Ma et al., 2023b), facing significant environmental challenges due to their vulnerability to extreme rainfall events and associated flood disasters (Yang et al., 2022; Xu et al., 2021). However, our trend analysis indicates a decrease in flood events in both the Yangtze and Pearl River basins, contrasting with Zhou et al. (2017)'s findings of increased extreme rainfall across all cities in the Yangtze Basin. This difference is attributed to the impact of a peak year within our study period, with 2010





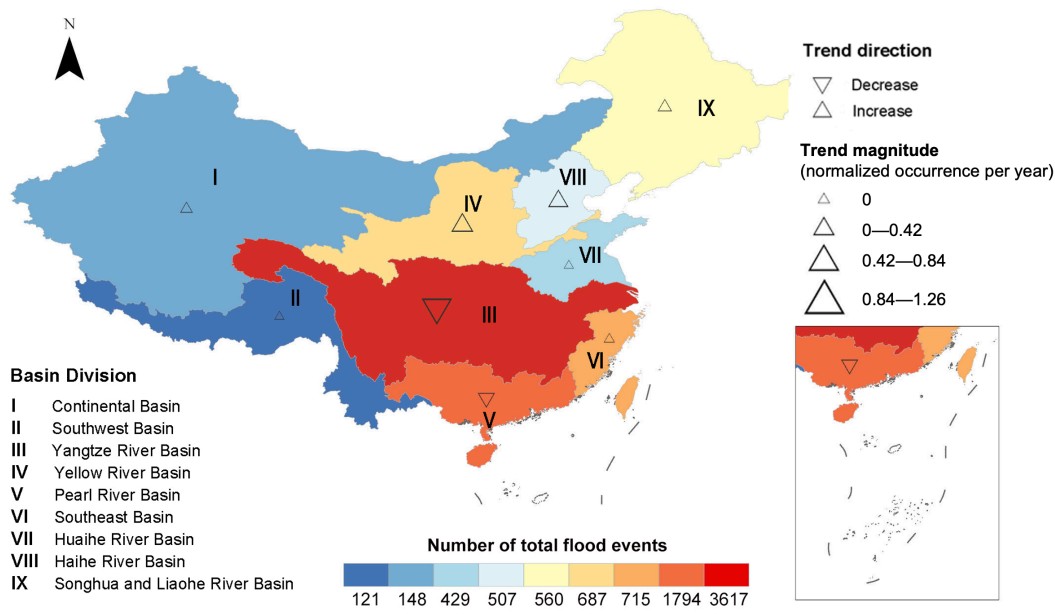

**Figure 11.** The accumulated flood occurrence and trend in different basins from 2000–2022. Roman numerals are used to represent different basins. Color indicates the number of flood events, with shades closer to red denoting higher occurrence. Triangles indicate trend characteristics, where downward and upward triangles represent decreases and increases, respectively, and the size is used to characterize the value of the slope.

being identified as a significant peak time primarily due to the event count in these river basins. The negative growth trend

calculated in this paper needs to be verified by more studies, and it cannot be interpreted as a reduction in flood.

Subtropical regions received the most frequent flood events in this study. Subtropical regions with their high temperatures and high humidity, particularly during summer and autumn, are especially prone to short-term heavy precipitation caused by convective activities (Li et al., 2022; Kotz et al., 2023). These climatic conditions make subtropical zones the primary contributors to the overall flood event count in China.

Despite the valuable insights provided by the spatial and temporal analysis in this study, it is crucial to acknowledge the limitations imposed by our reliance on news data, which may introduce systemic errors due to media attention. Future research should aim to integrate other data sources, such as social media and remote-sensing data, to improve the data quality. Furthermore, the subsequent analysis of the extracted information in this present study is limited to the presentation of spatiotemporal distribution, thus failing to highlight the unique value of county-level data in revealing regional flood characteristics. Future

research could focus on utilizing the dataset constructed in this present study for more detailed analysis. Attribution analysis of



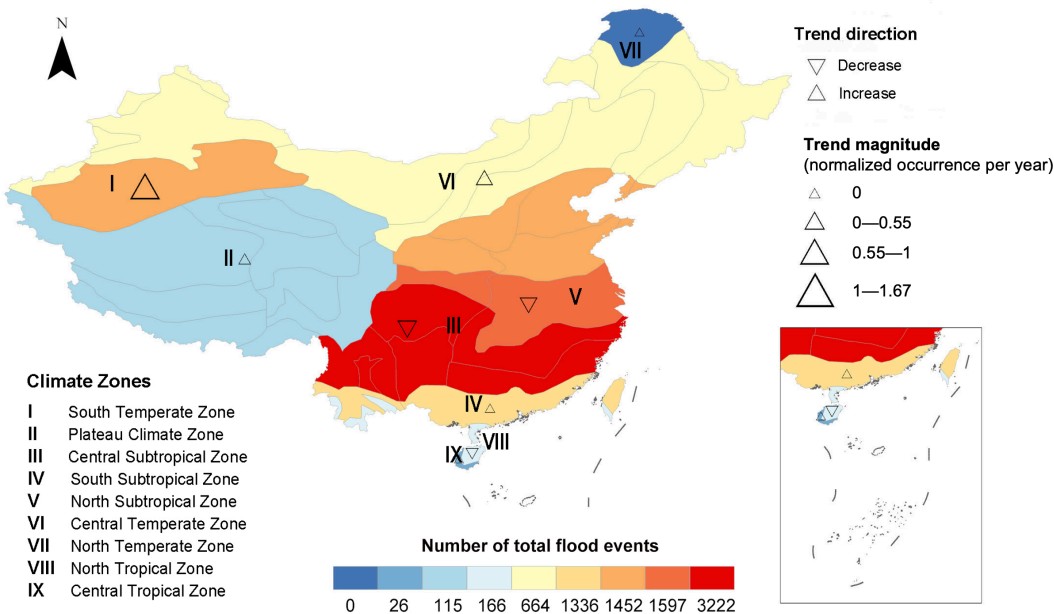

**Figure 12.** The accumulated flood occurrence and trend in different climate zones from 2000–2022. Roman numerals are used to represent different climate zones. Color indicates the number of flood events, with shades closer to red denoting higher occurrence. Triangles indicate trend characteristics, where downward and upward triangles represent decreases and increases, respectively, and the size is used to characterize the value of the slope.

floods could be studied to explore the main contributing factors in different areas. Additionally, by analyzing changes in land use and urban planning in specific counties, a more comprehensive understanding of how various factors interact at the local level to cause flood events can be achieved.

## 6 Conclusions

This study constructed a national urban flood dataset at the county level in China based on news media data through a question-answer format BERT model, providing an analysis of flood events in China and highlighting their temporal and spatial patterns. The BERT model was initially employed for spatiotemporal information extraction, achieving an F1 score of 0.86 and an exact match score of 0.82. Flood locations were then identified using a named entity recognition model that combines a BiLSTM network with a CRF layer. The established dataset consists of records from 2048 counties affected by flood disasters between





2000 and 2022, with a total number of 7559 events. This dataset is hosted on a GitHub repository and will be updated with the latest findings from flood-related news articles, thereby enhancing its utility for future research.

Although lacking authoritative county-level flood distribution data for validation, the initial accuracy of the trend in extracted information was confirmed by comparing the number of flooded cities with those reported in the *China Flood and Drought Bulletin*. Subsequent case studies of two widespread typhoon events established a strong correlation between flood distribution

and typhoon paths. Further analysis showed that flood-prone regions are typically in areas with high precipitation and low elevation, aligning with established hydrological principles. These validations confirm the accuracy of the dataset records constructed in this study, which can serve as a supplement to authoritative datasets and provide support for future research on urban floods.

Building on the insights from the flood event sets, this study further examines the temporal and spatial patterns of flood

occurrence in China. Temporally, the total flood occurrence from 2000 to 2022, excluding the influence of peak year, shows an upward trend. The peak in flood occurrence is identified in 2010, when 525 counties were affected. In addition, the seasonal characteristics shows that the rainy climate in summer leads to summer floods accounting for 74% of the total, highlighting the importance of being prepared for floods during this period. Spatially, flood occurrence was decreased from southeast to northwest. The southeastern regions like Guangdong Province, Fujian Province, and Guangxi Province and the southwest of

interior China like Sichuan Province, are the most frequently flooded areas. The Yangtze River Basin and Pearl River Basin, as economically developed and populated areas, were identified as areas particularly prone to flooding. As for climate zones, most of the subtropical zones across China experienced more floods than other climate zones.

*Data availability.* The national flood dataset constructed in this present study is accessible on a Github repository (https://github.com/shengnan0218/China-urban-flood-dataset.git) and will be continuously updated with new findings from flood-related news articles, en-

hancing its value for ongoing research. In addition, the DEM data used in this study was downloaded from http://www.gscloud.cn. The precipitation was derived from ERA5 data (Hersbach et al., 2020) (https://cds.climate.copernicus.eu).

**Appendix A: Appendix**

Table A1 shows the F1 score values used to evaluate BERT model performance during the hyperparameter tuning process, with a different calculation method from the main text. Precision and recall are calculated as:

$$Precision = \frac{O}{L} \tag{A1}$$

$$Recall = \frac{O}{S} \tag{A2}$$



O is defined as the maximum overlapping character length between the model's output and the standard annotated answer, L is the length of the output answer and S is the length of the standard answer. Finally, the F1 score is refer to Formula 3 in Section 485 3.

**Table A1.** The performance of BERT model with different hyperparameters during fine-tuning process

| Learning rate | Batch size | F1 score |
|---------------|------------|----------|
| $1 \times 10^{-5}$ | 4 | 83.35 |
| $1 \times 10^{-5}$ | 8 | 81.86 |
| $5 \times 10^{-5}$ | 4 | 83.81 |
| $5 \times 10^{-5}$ | 8 | 84.82 |

Table A2 shows the annual number of flooded cities in *China Flood and Drought Bulletin* and WiseNews.

**Table A2.** The comparison between the number of flooded cities in *China Flood and Drought Bulletin* and WiseNews

| Year | Number of flooded cities in bulletin | Number of flooded cities in WiseNews |
|------|--------------------------------------|--------------------------------------|
| 2006 | 118 | 116 |
| 2007 | 109 | 87 |
| 2008 | 115 | 100 |
| 2009 | 136 | 125 |
| 2010 | 258 | 167 |
| 2011 | 136 | 131 |
| 2012 | 184 | 141 |
| 2013 | 243 | 137 |
| 2014 | 125 | 108 |
| 2015 | 168 | 113 |
| 2016 | 192 | 135 |
| 2017 | 104 | 108 |
| 2018 | 83 | 96 |

*Author contributions.* Shengnan Fu conducted the investigation, developed the methodology, handled coding, and wrote the original draft. Heng Lyu provided guidance on methodology, and reviewed and edited the manuscript. David M. Schultz and Zhonghua Zheng also guided the methodology and coding, respectively, and participated in manuscript editing. Chi Zhang supervised the project and provided funding.





*Competing interests.*  The authors declare that they have no known competing financial interests or personal relationships that could have appeared to influence the work reported in this paper.

*Acknowledgements.*  This work was supported by several funding sources. The research was funded by. This research was partially supported by the fund of the National Key R&D Program of China (Grant No. 2022YFC3090601), National Science Foundation for Distinguished Young Scholars (Grant No. 51925902), and Key Fund of National Natural Science Foundation of China (Grant No. U2240204).We also
thank the China Scholarship Council for supporting Shengnan Fu's studies in Manchester, which facilitated this project. Additionally, David M. Schultz was partially supported by the Natural Environment Research Council UK (Grants NE/W000997/1 and NE/X018539/1).



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
