# Peer review of "Creating a National Urban Flood Dataset of China from News Texts (2000-2022) at the County Level"

_Hydrology and Earth System Sciences, 2024_

## Author Comment (AC1)

Dear Reviewer,

Thank you very much for your time involved in reviewing the manuscript and providing valuable feedback. Those comments are constructive for revising and improving our manuscript. We have taken the time to think through all of your comments and will carefully revise the manuscript to address each comment:

**General Comment 1. Flood Query Keywords**

*The flood query was limited to "flood" and "flood disasters" (L142, L154), while many other terms could hint at flood events in news items, e.g., "typhoon," "cyclone," "mud," "heavy rainfall," "inundated areas,"… Query terms are an essential aspect of event detection and this could be seen as a restriction limiting the detection power of the proposed approach. It raises some questions: Should this be documented as a limitation? Is it a decision to limit the size of the corpus? Does the Q&A approach prevent that concern?*

Thanks for bringing up this important point. However, the other keywords included may raise the dataset too large. For example, we tried using "heavy rainfall" as the query term and found that only around 10% news returned reported flood events. Most of these news texts are related to meteorological early warning information. Therefore, the current query was determined to limit the corpus to the most relevant content. Even if the Q&A approach can distinguish between relevant and irrelevant information, the benefits of large corpus are far less than the burden of running the model.

**General Comment 2. Flood Types and Multi-Hazard Concerns**

*The paper focuses on urban floods, excluding other types of floods, yet flood types are interrelated and very often not mutually exclusive. Hence, referring, for instance, to the Hazard Information Profiles (HIPs, https://www.preventionweb.net/drr-glossary/hips ), an urban flood could also be related to a flash flood (despite the exclusion of the query of "flash flood," L151), a riverine flood, a coastal flood, a groundwater flood. Floods are also secondary hazards associated with other hazards, such as a flood that could result from a Typhoon, heavy rainfall, a storm surge, an intense monsoon etc. Floods are also associated with geo-hazards such as landfall (See GLC studies). I found the Typhoon case study in the paper interesting. It also illustrates the multi-hazard nature of floods well. As in GLC studies, I would be interested in having the authors' view on multi, cascading, and co-occurring type issues, the possibilities of detecting multi-type floods, and the challenges, limitations, and perspectives concerning their proposed approach.*

Though we agree with this perspective, this article mainly focuses on urban flooding, especially its temporal and spatial information. There are two considerations regarding the reviewer's comment.

First, in future studies, it can be continuously mined as new contents in our database about whether it is transformed by other flood types, and its complex causes. We could extract multi-hazard information to add a column in the dataset to show what weather event caused the floods and a column to show the floods resulting in what geo-hazards such as landfall. We believe that the Q&A method can effectively identify the causal relationships between floods and other hazards only if news data can include this kind of information. For example, we manually checked 100 samples describing 52 events and 6 events mentioned that this flood caused by a typhoon. Therefore, the feasibility of disaster causality analysis based on news data needs to be further studied and confirmed. Our future research will also add other data sources to increase the data potential.

Second, some recent studies have used news media reports to extract information on various meteorological and geological disasters. However, most of these studies just classify news by rules rather than analyze the causality between disasters and did not subdivide flood into different flood types. For instance, Yang et al. (2023) applied a rule-based approach to extract 15 types of disaster information from news texts. Specifically, the rule implies that if any of these disaster names appear in the text, the news is categorized accordingly, and then the prefecture-level administrative names are used to match the location information in the news. Another example (Liu et al., 2018) also utilized keyword positioning and rule-based named entity recognition methods to identify disaster types and locations in the news. Both of above studies used this rule: if one news report mentions multiple disaster types at the same time, it is determined that the news event is multi-disaster co-occurrence. This method will introduce biases when a news just mentions two hazards but in different events that have no direct relationship. In the future, we could employ the language model to test the efficiency of extracting multi-type floods and other related hazards form news-based data, but the performance should be examined.

For these two reasons, addressing single- or multi-hazard information from the dataset is challenging and would require considerable thought to overcome these limitations.

This is a valuable point raised by the reviewer. So, in the revised version, we will add further discussion on this issue.

Yang, Chenchen, Han Zhang, Xunhua Li, Zongyi He, and Junli Li. "Analysis of spatial and temporal characteristics of major natural disasters in China from 2008 to 2021 based on mining news database." Natural Hazards 118, no. 3 (2023): 1881-1916.

Liu, Xiao, Haixiang Guo, Yu-ru Lin, Yijing Li, and Jundong Hou. "Analyzing spatial-temporal distribution of natural hazards in China by mining news sources." Natural Hazards Review 19, no. 3 (2018): 04018006.

**General Comment 3. A More Balanced Discussion: Trend Analyses vs. Gap Filling Potential**

*The manuscript extensively discusses spatiotemporal trend analysis, necessitating more caution and clarity on trends influencing factors. I understand the need to illustrate trends in the resulting dataset, but, in my opinion, this matter could be more efficiently summarized, and the paper could be more descriptive and less assertive in the interpretation. Some analyses are simplistic and do not go deep enough. Rather than make the paper even longer, I invite the authors to distinguish more between the essential and the accessory and, if anticipated, to cover in greater depth the spatiotemporal analysis of events and cross-referencing with third-party data in other papers (see GLC studies).*

*Some figures may be grouped, e.g., maps in different pannels of one figure, allowing not only to focus on the trends of the output data but also on how the output data compares to other datasets, which is currently limited to Figure 4, despite the numerous datasets being listed in the introduction. The reader has little clue as to what gap is being filled. In particular, the Chinese bulletin appears as a more exhaustive dataset (although coarser). This point may be worth further discussion.*

*Note regarding temporal trends:*

*Trends in hazard occurrences are complex, influenced by variations in hazard intensity and alteration of environmental susceptibility, as well as demographic shifts that alter exposure or vulnerability. Moreover, climatic cycles (e.g., ENSO or other climate indices) can distort linear trend estimations over brief periods due to their cyclical nature.*

*The complexity is further compounded when analyzing trends from news data. Changes in reporting capacity, especially in remote areas, along with new communication technologies like satellite and social media, may introduce significant biases. The proliferation of the internet during the 1990s and 2000s has notably impacted flood event reporting (Gall et al., 2009; Kron et al., 2012; Delforge et al., 2023). Kron et al., 2012 illustrate well the challenges in building a hazard database with flood examples. These works underscore the necessity for standardized flood event definitions to mitigate discrepancies in reporting scales. In the case of news scraping, the framing by journalists can significantly alter the perceived frequency, spatial representation, and the type of events.*

*In conclusion, the total number of flood events is a highly relative figure. It is essential to acknowledge that while flood hazards are natural phenomena, flood disasters and their reporting are social phenomena with potentially distinct and diverging trend patterns. Given these complexities, attributing trends depicted in the news (i.e., social variables, not physical ones) to climate change or land use changes requires careful consideration.*

We greatly appreciate the reviewer's detailed and insightful feedback. Your comments are invaluable in refining our analysis and ensuring our conclusions are both accurate and impactful. In response to your comments mentioned above, we have taken the following considerations:

First, regarding distinguishing more between the essential and the accessory, we will focus on highlighting the characteristics of the spatial distribution while streamlining the discussion of temporal trends, particularly simplifying the analysis of the influence of natural factors. In addition, our study focuses on urban floods, and the fundamental data is derived from news reports, which have a strong social dimension. Therefore, it is necessary to analyze the flood trend in different population density and economically developed areas to provide conclusions from an urban and social perspective. We will include this information in the revised version, with a detailed explanation provided in the latter part of this response.

Second, cross-referencing with third-party data in other papers or comparing to other datasets is challenging because of the absence of proper data. Therefore, we can only find some relevant data for comparison in certain regions. We have created a line chart for reference (Figure 1 below), to analyze the correlation of the direct economic losses provided by the Guangxi Provincial Government website due to floods after 2016, and the scale of disaster represented by the number of news-extracted flood-affected counties. These two indicators exhibit relatively consistent trends, which can to some extent suggest that the coverage of news data in certain regions is fairly good. However, these two indicators do not represent the same physical quantity, we think this figure may not suit for inclusion in the main text but can be provided as supplementary material for reference.

[Figure]

Figure 1. The time series of the number of news-extracted flooded counties and direct economic loss in Guangxi from 2016-2022.

Third, regrading what gap we have filled, it should be explained first that the *China Flood and Drought Bulletin* only provides the number of flooded cities in a general overview paragraph, without presenting their spatial distribution or specific inventory. The spatial distribution of flood loss information in the bulletin is limited to the province level, which encompasses multiple city-level areas. While our dataset is not comprehensive, it is the first county-level dataset on a national scale, and its time trends are largely consistent with authoritative data.

As for the temporal analysis, we agree that there are inherent limitations to using media data for temporal analysis. Overall, we will make the following adjustments in the revision:

In Section Temporal distribution of flood events and the relevant part of other sections, we will revise our statements on the temporal trends to reduce subjective interpretations and to clarify the limitations of news media data:

The temporal distribution of urban flood events in our dataset reveals an overall increasing trend over time. While this may reflect broader patterns of environmental change, such as the increase in extreme rainfall events driven by global warming and the effects of rapid urbanization, these trends should not be interpreted in isolation. Media data, which forms the basis of our dataset, is subject to various biases, as introduced by previous studies such as Gall et al. (2009) and Kron et al. (2012).

From the perspective of media communication studies, agenda-setting theory posits that by choosing which events to report on, the media effectively signals to the public which issues are important (Leidecker-Sandmann et al., 2023). Through the quantity and depth of coverage, the media can shape the level of public attention given to certain events. In the context of disaster reporting, the

government may influence the direction of media coverage to control public attention on specific disasters (Bai, 2022). For example, during the COVID-19 pandemic, research on government crisis communication showed that media agenda-setting was significantly influenced by government press conferences (Hayek, 2024). Crisis communication theory further explains the government can swiftly steer public opinion in the aftermath of a disaster, reducing the spread of negative emotions and maintaining social stability (Zhou et al., 2023). As a result, the variability in disaster reporting by the media may be influenced by multiple factors, including government policies, public interest, and the media's own resource allocation, leading to a situation where the volume of media reports is not necessarily consistent with the actual number of disaster events.

In Section Spatial distribution of flood events and relevant part in Section Discussion, we will streamline the results from different sub-regional analyses and group Figure 10, 11, and 12 into one figure as following picture (Figure 2) according to your suggestion:

[Figure]

Figure 2. The spatial distribution of flood occurrence.

Furthermore, additional analysis on population density and Gross Regional Product (GRP) will be included as follows:

[Figure]

Figure 3. The analysis of flood event trends across Chinese provinces from 2000 to 2022, shown in relation to (a) population density and (b) Gross Regional Product (GRP).

The background maps display average annual Gross Regional Product (GRP) in billion USD and population density in people per square kilometer, respectively, with darker shades indicating higher values. Overlaid on these maps are Theil-Sen estimated trends for the number of flood events, where the direction of the triangle represents whether the trend is increasing or decreasing, and the size of the triangle corresponds to the magnitude of the trend. Provinces without a significant trend are not marked.

Overall, most provinces exhibit an increasing trend in flood events, particularly in the northern, and western regions of China. These areas, including provinces such as Heilongjiang, Shandong, and Chongqing, are characterized by varying levels population density, both higher and lower, according to Figure 3(a). The provinces that exhibit a decreasing trend in flood events are primarily located in the central and southeastern regions, particularly in provinces like Jiangsu, Fujian, and Guangdong, which are notable for their higher population densities. This suggests that the rising flood events are not strictly tied to population density.

As for the trends in relation to economic output in Figure 3(b), the provinces with increasing flood trends are mostly those with lower to moderate GRP, such as those in the northern and western parts of China, despite Shandong and Zhejiang. These regions may not have received the same level of economic investment in flood control infrastructure as the more developed eastern provinces, which might explain the rising trend in flood events. On the other hand, the central and eastern provinces showing a decreasing trend, such as Jiangsu, Guangdong, and Sichuan, are among the most economically developed in China. This suggests that the availability of economic resources has allowed for more comprehensive flood management strategies, reducing the frequency of flood events in these areas.

It is important to note that several provinces with high population densities and significant economic development, specifically Jiangsu and Guangdong, exhibit a decreasing trend in flood events. These regions have experienced a high number of flood events over these years, with a notable peak around 2010. The estimated decrease in flood trends may be related to this peak, where the number of flood events was significantly higher than in other years, possibly skewing the trend calculations downward. Additionally, as regions frequently affected by flooding and characterized by high economic output and population density, substantial investments in flood management infrastructure and policies may have been made, also contributing to the observed decline in flood events. Jia et al. (2022) have highlighted the significant investments in flood management infrastructure in China's economically developed regions. They compared the 1998 and 2020 floods in Yangtze River Basin regions, which are economically developed regions in China. Their analysis reveals that significant improvements in risk management, including engineering defenses, environmental recovery, forecasting and early warning, and emergency response have led to a substantial reduction in flood disaster losses in Yangtze River Basin regions.

Gall, M., Borden, K. A., and Cutter, S. L.: When Do Losses Count?: Six Fallacies of Natural Hazards Loss Data, Bulletin of the American Meteorological Society, 90, 799–810, https://doi.org/10.1175/2008BAMS2721.1, 2009.

Kron, W., Steuer, M., Löw, P., and Wirtz, A.: How to deal properly with a natural catastrophe database – analysis of flood losses, Natural Hazards and Earth System Sciences, 12, 535–550, https://doi.org/10.5194/nhess-12-535-2012, 2012.

Leidecker-Sandmann, Melanie & Koppers, Lars & Lehmkuhl, Markus. (2023). Correlations between the selection of topics by news media and scientific journals. PloS one. 18. e0280016. 10.1371/journal.pone.0280016.

Bai, Sheng. Mainstream Media Agenda Setting in Disaster Events. Journal of Emergency Management and Disaster Communications 3, no. 2 (2022): 83-98. https://doi.org/10.1142/S2689980922500038.

Hayek, Lore. (2024). Media Framing of Government Crisis Communication During Covid-19. Media and Communication. 12. 10.17645/mac.7774.

Zhou S, Yu W, Tang X, Li X. Government crisis communication innovation and its psychological intervention coupling: Based on an analysis of China's provincial COVID-19 outbreak updates. Front Psychol. 2023 Jan 26;13:1008948. doi: 10.3389/fpsyg.2022.1008948. PMID: 36778169; PMCID: PMC9909028.

Jia, Huicong & Chen, Fang & Pan, Donghua & Du, Enyu & Wang, Lei & Wang, Ning & Yang, Aqiang. (2021). Flood risk management in the Yangtze River basin

—Comparison of 1998 and 2020 events. International Journal of Disaster Risk Reduction. 68. 102724. 10.1016/j.ijdrr.2021.102724.

**General Comment 4. Analyses of GDP**

*The manuscript highlights the GDP as the primary driver of media attention. However, the boxes in Figure 5 do not seem to show any significant difference between the occurrence of floods for different GDP groups. So, to highlight a possible effect of GDP on media attention, it is vital to use GDP per capita (see GLC studies).*

*The population is a critical factor in media attention and hazard exposure. More densely populated cities should receive more media attention in the event of a flood. It is likely the primary factor explaining the spatial patterns in the dataset. It is likely to be correlated with GDP, as well as other factors such as elevation, distance to river or coast, or climate (see G5). Therefore, controlling that factor when investigating some effects is essential.*

We agree with your perspective. Our initial motivation for conducting the GDP clustering analysis was to explain how regional economic development might influence the biases in media data. However, after carefully considering the reviewers' comments and reviewing literature on media communication themes, we have decided to remove this section. Relying solely on economic development or population density to explain the biases in media data is not convincing enough. In the revised version, we will modify our explanation of the biases introduced by media data as mentioned in the response to G3.

Moreover, we will add the analysis of the flood trend in different population density and economically developed areas as mentioned in the response to G3.

**General Comment 5. Analyses of Flood Susceptibility**

*Figure 7 and the underlying analysis of flood susceptibility present some issues and do not bring much to the paper. The proposed pattern is not very neat (the points also overlap with no transparency), likely because the chosen indicators are quite remote proxies of flood susceptibility and should not be presented as acknowledged indicators in hydrology (the supporting references are weak).*

*Average daily precipitation depicts a hydrological equilibrium rather than an extreme event. Naturally, arid regions are less susceptible (also less populated, hence, exposed). However, the indicator becomes less relevant to other hydrological systems with higher precipitation averages (a mixture of blue and red dots). Likewise, elevated areas are also likely to be less populated and then less exposed, and the elevation effect tends to disappear at a lower elevation.*

*Flow accumulation or topographical wetness indices could have been more reliable indicators of flood susceptibility.*

*I would recommend removing this analysis given its low informative value and also because these variables are related to climate variability, which is already pictured in Figure 12. See GLC studies for comparisons.*

Thank you for pointing out the issue with the selection of flood susceptibility factors. We agree that the factors initially chosen were not appropriate. Average rainfall reflects the general characteristics of a region, but flood disasters are often associated with extreme rainfall. Additionally, discussing the impact of elevation alone is not convincing given the large extent of the study area. We will remove this section in the revision.

**General Comment 6. Flood Events Dataset Resolution**

*While the final dataset is reported at the county-month level, the reader is left with little insight into the level of detail directly resulting from the information extraction process, which remains unclearly described. Based on Figures 4 and 6, it appears that information at the city-daily level was collected. It seems that a much more precise dataset could have been shared without much additional effort, raising questions about the motivation behind disaggregating the data to such a coarser level.*

We are sorry that our description may confuse readers especially the term "county". First, we think the administrative level in China should be introduced:

The provincial level is the highest level of administrative division in China, and it consists of: Provinces, Autonomous Regions, Municipalities, Special administrative Regions (Hong Kong and Macau); The second level is prefectural level including: Prefecture-level Cities (just cities in the usual sense), Autonomous Prefectures, Leagues (found in Inner Mongolia); The third level is county level including: Counties, County-level Cities (smaller cities under the jurisdiction of a prefecture-level city), Districts, Banners (found in Inner Mongolia); The forth level is township level including: Towns, Townships (typically more rural areas), Subdistricts; And the last level is village level including: Villages, Communities.

Therefore, a county is a finer administrative division than a city, with one city typically comprising several county-level areas.

The locations extracted from news reports typically include only the county-level area name or the county name with the specific flooded street or building. Therefore, we standardized the spatial information by using county names.

Second, most of the data can be extracted to specific day information, but some can only be extracted to month, so at first, in order to unify the data set, we set the time resolution as month. In the revised version, we will change the events with day information to be accurate to day.

As for the figures you mentioned, Figure 6 is indeed the flood events with daily information within two typhoon event months. However, in Figure 4, we used a line plot just to show the temporal trends of news-reported flooded cities amount and those reported in bulletins. The data is aggregated annually rather than daily.

**General Comment 7. Data Content, FAIR Principles, and Reusability**

*Also, given that a central outcome of the paper is a dataset, alignment with FAIR principles (https://www.go-fair.org/) should be particularly encouraged. Regarding the data shared, GitHub is not considered FAIR as it does not allow for persistent identifiers. Also, a few additional data could greatly increase the reusability of the dataset, e.g., precise column descriptions in the readme, the reference for the administrative unit shapefile to link the data with the post-code or administrative units as described in the paper (L275-278), using international time standards, and possibly translate region names to English to maximize reuse in the global context.*

*Regarding reproducibility, the data and code availability section could be improved. Input news data and their conditions of (re-)use are not described in this section. Tools and libraries being used to develop the approach are not referred to (except references to the Python "Re" module at L187). There is no comment about whether or not the developed models are accessible and under which conditions of use.*

*There are no links or references to the news articles that have been used to construct the dataset. Sharing the links could drastically increase the paper's outreach and support future research and NLP applications to extract additional information, such as flood impact variables or associated hazard types, without redeveloping an NLP flood event detection model. Annotated corpora are also valuable datasets in the context of NLP for future benchmarking. Consider commenting on that dataset as well.*

Thanks for your helpful suggestions. We will change the dataset sharing website to Zenodo, which is an open-access repository that allows researchers to share and preserve their datasets. It is operated by CERN and OpenAIRE and provides features like DOIs for citations, which supports the FAIR principles. Furthermore, we will add a column describing post-code and precise column descriptions in the readme, and translate region names to English. As for the administrative unit shapefile to link the data with the post-code, we will add the reference in Section Data availability.

About the input news data, we will check the link of the news platform and share it in Section Data availability. Then, readers can retrieve the news using the query as we described in Section News data and download the data according to the data management rules of the WiseNews platform. We can share the annotated flood-related corpora in our open-access dataset.

We will refer to the tools and libraries being used to develop the approach, such as Tensorflow in the revised version. Besides, we could provide the developed model to the readers who contact to us.

**Specific Comment 1.**     *L8: "similar" could be more nuanced.*

We will re-organize this sentence to describe the comparison between news-based floods and bulletin data:
Our analysis reveals that while there are notable differences in the magnitude reported events in peak years, the temporal trend of flooded cities in the news-based dataset broadly aligns with that in the *China Flood and Drought Bulletin*.

**Specific Comment 2.**     *L9:10: "the connection between…": the connection does not support accuracy and the analysis is oversimplistic (See G5).*

We agree that the flood susceptibility indicators were insufficiently appropriate. As response to Comment G5, we will remove this analysis in the revision.

**Specific Comment 3.**     *L43 (and after): "natural disaster" is a controversial terminology often avoided by Disaster Risk experts, acknowledging that a disaster is not natural (as opposed to natural hazards).*

We will correct it to 'natural hazard' in the revised version of the manuscript.

**Specific Comment 4.**     *L43-L52: Table 2 could distinguish between catalogs from remote and social sensing, e.g., that DFO is based on remote sensing, EM-DAT on the collection of text documents and manual extraction of the information. Some missing recent initiatives could be worth mentioning, e.g., a global remote sensing catalog is the global flood database and a global catalog obtained from social media:*

- *Tellman, B., Sullivan, J.A., Kuhn, C. et al. Satellite imaging reveals increased proportion of population exposed to floods. Nature 596, 80–86 (2021). https://doi.org/10.1038/s41586-021-03695-w*

- *J.A. de Bruijn, H. de Moel, B. Jongman, M.C. de Ruiter, J. Wagemaker, J.C.J.H. Aerts. A global database of historic and real-time flood events based on social media. Scientific Data, 6 (1) (2019), p. 311, 10.1038/s41597-019-0326-9*

- *G.R. Brakenridge. Global Active Archive of Large Flood Events. Dartmouth Flood Observatory, University of Colorado, USA. http://floodobservatory.colorado.edu/ Archives/ (Accessed xxx)*

- *Delforge, D., Wathelet, V., Below, R., Lanfredi Sofia, C., Tonnelier, M., Loenhout, J. van, and Speybroeck, N.: EM-DAT: the Emergency Events Database, preprint, https://doi.org/10.21203/rs.3.rs-3807553/v1, 2023.*

We will add a column to distinguish between databases from remote and social sensing and the recent datasets in Table 2 as follows:

| | Name | Period | Flood Records | Update Frequency | Source |
|---|---|---|---|---|---|
| Social sensing | The Emergency Events Database (EM-DAT) | 1900-- | Time, location and damage of global flood events that resulted in a certain number of deaths or economic losses | Continuously | Centre for Research on the Epidemiology of Disasters |
| | Natural Disaster Data Book | 2002-- | Statistical and analytical perspectives of flood events in Asia (data retrieved from EM-DAT) | Annual | Asian Disaster Reduction Center |
| | Global Flood Monitor | 2014-2023 | A real-time overview of ongoing flood events based on filtered Twitter data | Pause | IVM - VU University Amsterdam and FloodTags |
| | Floodlist | 2016-- | Dates, locations, magnitude and damages of each flood events based on news | Real-time | FloodList (funding from Copernicus) |
| Remote sensing | Dartmouth Flood Observatory (DFO) | 1985-- | Time, location and extent of global flood events using satellite observations | Continuously | University of Colorado Boulder |
| | Global Flood Awareness System | Real-time | Ongoing and upcoming flood events information from satellites to support flood forecasting at national, regional and global levels | Real-time | Copernicus Emergency Management Service (CEMS) |

| | | | | |
|---|---|---|---|---|
| Global Flood Monitoring System (GFMS) | Real-time | Flood inundation extent and depth based on precipitation satellite data and flood model simulation | Every 3 hours | University of Maryland and NASA |
| The Global Flood Database | 2000-2018 | flood extent and population exposure for 913 large flood events | unknown | Floodbase |

**Specific Comment 5.** *L65: Beyond cloud cover for optical imagery, mapping urban flood is challenging per se.*

It is true that mapping urban flooding is inherently challenging, and we just listed one source of uncertainty. In the revised version, we will modify this sentence to emphasize that mapping urban flooding is already a technical challenge per se.

**Specific Comment 6.** *L75: "Yang et al. (2023)" Such a paper of high relevance should be re-discussed later in the discussion section, among others, to identify (see Overview).*

Thank you for the helpful suggestion. As the response to G2, we will add the discussion on multiple hazards, particularly those related to flooding, and reviewing these highly relevant papers.

**Specific Comment 7.** *L77: The authors acknowledge the multi-hazard nature of floods here and after, but the issue is not discussed in light of their own work (see G2).*

We will add the discussion on the multi-hazard nature of floods in the revised version (see the response to G2)

**Specific Comment 8.** *L90: "Conditional Random Fields (CRF) layer" appears to be a central part of the methodology appearing multiple times in the paper; however, it lacks a clear explanation of what it is and why it is used.*

Sorry for this unclear statement. CRF model is a type of discriminative probabilistic model used to predict sequences of labels for sequences of input samples. It considers the context (i.e., neighboring labels) to make more accurate predictions. The CRF layer was part of the named entity recognition (NER) method in our approach.

We used BERT to extract initial answers including spatiotemporal information of floods and then, adopted an NER method called BiLSTM-CRF model to identify the location names in the answers. In the NER model, a BiLSTM layer is adopted to extract features from the input character vectors. And then, the CRF layer uses

the output from BiLSTM to compute the most likely sequence of labels considering the dependencies between labels.

**Specific Comment 9.** *L110:116: since the paper follows a conventional structure, it is unnecessary to detail it in the introduction.*

We will remove these explanations in revised version according to your suggestion.

**Specific Comment 10.** *Table 2: EM-DAT is continuously updated (see Delforge et al., 2023). I would also refer to the Global Flood Awareness System (https://global-flood.emergency.copernicus.eu/), the flood component of CEMS, instead of CEMS. See also S4.*

We will update Table 2 according to your comments (See response to S4).

**Specific Comment 11.** *L134: check url link (404 error).*

We will check this issue and re-share the data link (https://www.wisers.com/wisesearch)

**Specific Comment 12.** *Figure 1: I appreciate the availability of an example. However, consider selecting a more topic-appropriate example or asking for a where/when the question for more relevance.*

We will provide a more topic-appropriate example in the revised version, such as followings:

**Context:** Faced with a once-in-50-years "7·23" catastrophic flood, the city committee and government treated the disaster as a command and time as life, mobilizing the entire city, resettling the affected people, and actively conducting post-disaster epidemic prevention. In this event, many houses in Luzhou's Linjiang were submerged, and at least 100,000 to 200,000 people needed to be relocated, which is unprecedented in Luzhou's recent decades of history.

**Question1:** What disaster event occurred?

**Answer1:** "7·23" catastrophic flood.

**Question2:** When did it happen?

**Answer2:** "7·23".

**Question3:** Where did the disaster occur?

**Answer3:** Luzhou.

**Specific Comment 13.**    *L142, L151, and L154: See G1.*

We could add sentences to explain our query term determination if needed:

Our study focuses on the mining of flood events, although other meteorology-related terms such as "typhoon," "cyclone," "heavy rainfall," may be related to flood events, but there are very few flood event news only mentioning flood-causing terms like typhoon. At the same time, we examined the results of a separate query of "heavy rain", and only 10% were reported flood events, most of which were meteorological warnings. Therefore, in order to control the relevance of corpus and improve the efficiency of model, this study limited the current search terms.

**Specific Comment 14.**    *L145-148: The description of the data and its processing, including test/train split, may be confusing. It may be more appropriate to move to the method section.*

We will move the description of the data processing to the method section.

**Specific Comment 15.**    *L157: "Validation" unless China Flood and Drought Bulletin is considered a gold standard, I think referring to comparative data and cross-comparison instead of validation is more appropriate.*

We agree with you. We will replace "Validation" in revised version according to your suggestion.

**Specific Comment 16.**    *L168-L174: oversimplistic view of hydrology and weak references. See G5.*

We will remove this part.

**Specific Comment 17.**    *L190-199: This section could indicate the total/train/test sample sizes more clearly.*

Sorry for unclear explanation. The total of the CNKI news samples was 633, and these samples were divided into three parts: 402 samples for fine-tuning BERT (alongside with CMRC2018); 101 samples as validation set for adjusting hyperparameters; 130 samples for testing. We will re-organize the data-processing part in method section and remove current related descriptions in data section.

**Specific Comment 18.**    *L235: words should be singular in "and does contain the words 'will'…". Also, I wonder if this approach successfully separated actual events from forecasts? Is there any language specificity in Chinese invoved here?*

We don't think it is related to the language specificity in Chinese, the forecasts should include the words representing future state. Therefore, we can take two steps to distinguish actual events and forecasts. Firstly, the answer to Question 1 could contain the flood-related events. The answer is usually just one short sentence which defines the events described in the news. Secondly, we identify the words representing future state and remove the corresponding events.

**Specific Comment 19.** *Figure 3: Is [SEP] a requirement given the specificity of the Chinese language?*

No, [SEP] is a special token used for BERT model not just for Chinese language tasks. In a Question-Answering (Q&A) task using BERT, the [SEP] token is essential. It separates the question from the context or passage from which the answer needs to be extracted. The typical input format for BERT in a Q&A task is: [CLS] Question [SEP] Context [SEP]

This structure helps BERT understand the boundaries and relationships between the question and the context, facilitating accurate extraction of the answer.

**Specific Comment 20.** *L243: In the first sentence, correct "flood information extraction" into "(i) flood event detection and (ii) flood information extraction" for clarity.*

We will address this issue in the revised version.

**Specific Comment 21.** *L259: it is not clear to me how Exact Match behaves in case of multiple locations, zero if any error? What is it clearly meant by the location data? City? County? How is location handled before the flood location recognition is explained in section 3.2? Perhaps 3.2 should be explained before.*

Yes, if any one of multiple locations was not identified then the score for this sample is zero.

The location data specifically refers to county-level region name.

Sorry for unclear structure. We did not do any further processing of the answer information before flood location name recognition. We agree that Section 3.2 should be explained before and will address this issue in the revised version.

**Specific Comment 22.** *L276: consider adding the reference of the used administrative unit shapefile. See also G7.*

We will add this reference in the revised manuscript.

**Specific Comment 23.** *L285, section 4.1. The performance seems good in an absolute manner, but the reader has no clue how this performs in relation to the*

*context of social sensing of flood or in the context of Chinese NLP. This is quite important to document.*

Thanks for pointing that out. We will add the discussions on the performance of our NLP method.

The performance of BERT model in this current study are competitive within the broader field of information extraction and Chinese NLP. For instance, Yang et al. (2022) adopted a BERT-based model for Chinese named entity recognition (NER) and achieved 94.78% and 62.06% F1 values on the MSRA (created by Microsoft Research Asia, is a well-structured and annotated collection of text for NER tasks) and Weibo (A Chinese social media platform) datasets, respectively. This significant disparity in performance highlights the challenges in semantic understanding in social media data compared to more structured datasets like MSRA. In addition, Kim et al. (2022) developed a question answering method for infrastructure damage information retrieval from textual data using BERT and achieved F1-scores of 90.5% and 83.6% for the hurricane and earthquake datasets, respectively.

Yang, Ruisen, Yong Gan, and Chenfang Zhang. 2022. "Chinese Named Entity Recognition Based on BERT and Lightweight Feature Extraction Model" Information 13, no. 11: 515. https://doi.org/10.3390/info13110515

Kim, Y., Bang, S., Sohn, J., & Kim, H. (2022). Question answering method for infrastructure damage information retrieval from textual data using bidirectional encoder representations from transformers. Automation in construction, 134, 104061.

**Specific Comment 24.** *Figure 4: Bulletin seems more exhaustive. This could be discussed more and the authors could highlight better complementarities between data collection approaches, e. g., how would the proposed approach improve Chinese bulletin?*

The spatial distribution of flood loss information provided in the bulletin is only at the provincial scale, and the number of flooded cities is mentioned in the paragraph describing the overall extent of the disaster. However, it does not provide a specific list of flooded cities or time information of each event. We have provided a more detailed list of affected counties. Additionally, we visualized the year-on-year differences in the data to offer a clearer view of interannual variations. As shown in the figure below, despite some degree of underestimation, the temporal trends in our data align closely with those reported in the Bulletin.

[Figure]

**Specific Comment 25.** *L298-L308: The analysis of media attention due to GDP biases is not significat and do not control for the population bias (see G4).*

We will remove the analysis of media attention due to GDP biases and detailed explanation is in response to G4.

**Specific Comment 26.** *L313-314: The two case studies were selected as the author assumed a good coverage because of their important hazard magnitude and impact. This is a known bias and an issue worth mentioning, as small-impact disasters tend to be less well-covered and documented. See Kron et al., 2012, Gall et al. 2009, and Delforge et al. 2023 and references therein for more insights about hazard catalog biases.*

We quite agree with you. We selected these two cases to show that our dataset can cover the more impactful events. However, it is true that small-impact events receive much less media attention, which is one of the limitations of our data set based on media data. We also appreciate the references you provided, and we will discuss the bias caused by media data as the response to G3.

**Specific Comment 27.** *L328-339 + Figure 7. These selected indicators are bad proxies of flood susceptibility, and I do not see how this analysis validates something about the spatial distribution of floods (see G5). Consider removing.*

We will remove this part.

**Specific Comment 28.** *L340: how the information was structured prior to harmonizing the data into the urban flood dataset is unclear. See also G6.*

The detailed description is in the response to G6. We will add the explanation in the revision.

**Specific Comment 29.** *Figures 8 and 9, it would be great to have an additional column or a time series on the Y axis with the annual total. This could help identify pluriannual cycles as a result of climate indices. Consider adding the total number of occurrences and items in the figure caption.*

According to your suggestions, we will modify these figures as followings:

[Figure]

Figure 8. Flood occurrence heatmap by year and month.

[Figure]

Figure 8. Flood-related news heatmap by year and month.

**Specific Comment 30.** *L354: "seasonality" instead of "climate's tendency" could be more appropriate.*

We will address this issue in the revised version.

**Specific Comment 31.** *L390: "exposure" or "susceptibility" (the environmental side of vulnerability) is maybe more appropriate than vulnerability because the latter also encompasses social vulnerability.*

We will replace vulnerability with susceptibility in the revised version.

**Specific Comment 32.** *Maps Figures 10, 11, and 12 could be grouped into a multipanel figure for conciseness. Consider adding population density as well since it drives hazard exposure. DEM and river networks may also be considered as information to include (parsimoniously).*

First, we will group these figures into a multi-panel figure as you suggested and include the population density and the Gross Regional Product (GRP) related analysis. The detailed description is in the response to G3.

Regarding the suggestion to include DEM and river networks, we appreciate the idea but believe these factors, while relevant, do not directly align with the primary focus of our analysis. Incorporating DEM and river networks would introduce additional complexity that may not substantially contribute to the core

findings or enhance the validation of our flood distribution data. We agree with your opinion in G5 that in areas with very high elevation, the low population exposure naturally leads to fewer reported flood events, so conducting spatial analysis with DEM as the sole base layer does not have significant meaning. Similarly, the independent analysis of river networks is not particularly meaningful.

**Specific Comment 33.**    *L409: The comparison with other datasets is quite limited, and the Chinese bulletin seems more exhaustive if one can trace the original data. To what extent the proposed dataset fills gaps is thus not very well documented (see G1). Adding more than one catalog from Table 1 and 2 in Figure 4 for comparison can improve this discussion.*

There is no other Chinse national-level dataset describing the inventory of urban floods. The *Chinese Flood and Drought Bulletin* just shows the number of flooded cities for each year without specific flooded cities inventory and in recent years, even the numbers have not been published. Additionally, no other datasets from Table 1 and 2 could provide the number of flooded cities or counties across China so that we cannot add more than one catalog in Figure 4. The absence of such comparable data itself highlights that our dataset fills a gap in urban flood data on a national scale in China.

The spatial distribution of flood loss information in the bulletin is limited to the province level, which encompasses multiple city-level areas. Our dataset, despite its limitations, offers more granular information by identifying specific flooded areas at the county level, which is smaller than the city level. There may be biases inherent in the news data, but we believe that our dataset serves as a valuable reference in the absence of more detailed and comprehensive data sources.

**Specific Comment 34.**    *L473: The data availability section does not include the input news data accessibility information. In line with HESS recommendations and FAIR standards, I also encourage the authors to share information about code and model availabilities.*

We will ensure the maximum possible sharing of data and code. The detailed explanation is in the response to G7.

**Specific Comment 35.**    *L414-L416: this sentence (and the section in general) looks like the authors do their best to fit in the context of climate change and urbanization, even excluding some peak values to retrieve a positive trend. Trends, in particular for disaster news, are much more complex than trends observed on physical variables and include important social drivers and biases. The discussion is oversimplified, and the authors should take more distance and*

*inquire about the biases arising from social sensing of hazards. See G3 and references.*

We agree that trends for disaster news are much more complex than trends observed on physical variables and include important social drivers and biases. We will add the discussion on the biases arising from social sensing of flood hazard and the detailed description is mentioned in response to G3.

**Specific Comment 36.** *L445: Perspectives are neither exhaustive nor detailed. Consider adding more relevant perspectives, differentiating those related to the method (NLP-detection, extraction) and those related to the valorization of the resulting dataset.*

We will re-organize the discussions of limitations and future directions on the method and the resulting dataset according to your suggestion.

The current study employs a BERT model for question-answering tasks, which has proven efficient in information extraction. However, with the rapid advancement in large language models (LLMs), newer models such as ChatGPT offer significant improvements in various NLP tasks, including text classification, question-answering, and text generation, achieving state-of-the-art results. For instance, Colverd et al. (2023) have successfully used several LLMs, including GPT-3.5, GPT-4, and PaLM-Text-Biso, to generate flood disaster impact reports by extracting and curating information from the web. They found a notable correlation between the scores assigned by GPT-4 and human evaluators when comparing generated reports to human-authored ones. Furthermore, Hu et al. (2023) proposed a method fusing geospatial knowledge of locations with GPT models to extract location descriptions from disaster-related social media messages, demonstrating a 40% improvement over typically used Named Entity Recognition (NER) approaches. Given these advancements, our future research will explore the use of LLMs to extract nuanced information from flood-related text data, which includes distinguishing flood types, causes, and the specific losses associated with each flooding event.

The subsequent analysis of the resulting dataset (constructed from the extracted information) in this present study also has limitations, which fail to fully leverage the advantages of county-level data in revealing regional flood characteristics. Future research could involve attribution analysis of floods to explore the main contributing factors in different areas. Additionally, by analyzing changes in land use and urban planning in specific counties, a more comprehensive understanding of how various factors interact at the local level to cause flood events can be achieved. Moreover, leveraging advanced machine learning models, such as deep learning and ensemble methods, could enhance the predictive capabilities of flood risk evaluation. By addressing these aspects,

future studies can significantly improve the utility of county-level flood data, offering better-informed strategies for flood mitigation and resilience planning.

Colverd, Grace, Paul Darm, Leonard Silverberg, and Noah Kasmanoff. "Floodbrain: Flood disaster reporting by web-based retrieval augmented generation with an llm." arXiv preprint arXiv:2311.02597 (2023).

Hu, Yingjie, Gengchen Mai, Chris Cundy, Kristy Choi, Ni Lao, Wei Liu, Gaurish Lakhanpal, Ryan Zhenqi Zhou, and Kenneth Joseph. "Geo-knowledge-guided GPT models improve the extraction of location descriptions from disaster-related social media messages." International Journal of Geographical Information Science 37, no. 11 (2023): 2289-2318.

**Specific Comment 37.**     *L473: data and code availabilities: see G7.*

The detailed explanation is in the response to G7.

**Specific Comment 38.**     *Table A2: Same as Figure 4. It may be removed, in my opinion.*

We initially want to share the raw form data for Figure 4 incase that some readers are interested. However, the information indeed is duplicated. We will remove it in the revise version.

---

## Author Comment (AC2)

**Comment 1:** *From the abstract, but also the rest of the paper, the level of detail that this flood information dataset has is unclear. A spatial scale is mentioned as 'county-level', but that can vary quite a lot depending on where the reader is from. Connecting this to a typical length scale (1, 10, 100, … kilometres?) will make it clearer to a potential end-user whether this dataset is useful.*
  *Similarly: what kind of information is present about the flooding? Is it just spatial extent? Or also indications of amounts of water, timing or duration, damages done, etc etc. This should be immediately clear from the first reading, in both the abstract, as well as the results section.*
*Related to this, table 1 is an overview of current flood disaster reports, which also doesn't contain any information on the kind of data that's in there. Giving both your, and the existing datasets that level of detail can make it clear what the advantage of this new methodology is in comparison to the existing ones. Also, the validation data described in section 2.3 suffers from this lack of information.*

First, we would like to explain the relationship between the different administrative levels in China:

The provincial level is the highest level of administrative division in China, and it consists of: Provinces, Autonomous Regions, Municipalities, Special administrative Regions (Hong Kong and Macau); The second level is prefectural level including: Prefecture-level Cities (just cities in the usual sense), Autonomous Prefectures, Leagues (found in Inner Mongolia); The third level is county level including: Counties, County-level Cities (smaller cities under the jurisdiction of a prefecture-level city), Districts, Banners (found in Inner Mongolia); The forth level is township level including: Towns, Townships (typically more rural areas), Subdistricts; And the last level is village level including: Villages, Communities.

Regarding the specific spatial scope, the size of county-level administrative regions varies greatly across different parts of the country. The smallest county-level administrative region is Jing'an District in Shanghai, which covers only 8 square kilometers, while the largest is Ruoqiang County in the Xinjiang Uygur Autonomous Region, covering 206,903 square kilometers. The average area of a county-level administrative region nationwide is 3,459 square kilometers.

This is important for enhancing the readability of the article and dataset. We will include a brief explanation of the spatial scope in the revised version.

Second, our dataset provides information solely on the timing and names of the affected areas. Unfortunately, it does not include details such as the spatial extent, water volume, or damages caused by the flooding. This limitation arises from the nature of the data source—we relied on news reports rather than scientific papers, which typically do not provide the physical measurements or quantitative details often found in more specialized studies. In the future, we will introduce more data

source to update and correct the dataset and add these kinds of flood event details.

We will make this clearer in the abstract and results sections of the revised manuscript to ensure that readers understand the dataset.

Additionally, we will update the flood record information in Table 1 like followings:

| Name | Period | Flood Records | Update Frequency | Source |
|---|---|---|---|---|
| Annual Report of Chinese Hydrology | 2021-- | Number of basin/river floods and flooded river list | Annual | Ministry of Water Resources of the People's Republic of China |
| China Flood and Drought Bulletin | 2006-- | The population, economic, and crop losses in each province | Annual | Ministry of Water Resources of the People's Republic of China |
| China Meteorological Disaster Yearbook | 2004-- | Time, flooded district, damage of major flood events, the record criteria as events causing over 50,000 hectares of agricultural damage, 10 deaths, or 14 million USD in direct economic losses | Annual | China Meteorological Administration |
| Reports on official website of China National Disaster Reduction Center | 2011-- | Records of the time, location and damage of flood events (Data prior to 2018 is not available) | Real-time | National Disaster Reduction Center of China |

**Comment 2:** *The approach used seems quite specific for the Chinese language, using several specifically trained models and training input. It's worthy of discussion of your approach also works for a completely different language group to apply this methodology in other data-scarce regions (e.g. the Global South).*

Our approach was indeed fine-tuned using a Chinese corpus, which means that applying this methodology to a completely different language would require retraining the model with a suitable corpus in that language. This is because BERT

models are language-specific, and the fine-tuning process is critical to adapting the model to the nuances of the target language. Moreover, our method relies on the availability of news data, which may be less abundant or even scarce in certain regions. This potential lack of media data could limit the applicability of our approach in those areas.

This point is important, and we will include this discussion in the revision. While the model we trained cannot be directly transferred to regions with different languages, the technical approach we have developed can be applied in any region and serves as a reference.

**Comment 3:** *Also, regarding the approach: the media used are all newspaper databases, and only 2 different ones. Why is social media not included, or other sources of information? This seems to limit the potential of the method, since using one type of media source might be fairly uniform in its wording and phrasing, and perhaps not always covering all instances of floods. Furthermore, the restrictive choices on the keywords to select these articles might make the whole model biased: was there any form of testing with broader search terms, synonyms or other idioms for instance (like in L 152)? The model is strongly influenced by the choices for the training data obviously, but it seems to me like some additional testing of the influence of that training data is necessary.*

First, social media data is often non-public and may involve privacy issues, which can impose limitations on its use. On the other hand, due to concerns about data quality control, news articles were selected in the hope of obtaining more accurate results. Data with varying language styles might negatively impact the model's performance.

Regarding the problem of keywords, we tested other keywords as retrieval methods and found that the other keywords included may raise the dataset too large. For example, we tried using "heavy rainfall" as the query term and found that only around 10% news returned reported flood events. Most of these news texts are related to meteorological early warning information. Therefore, the current query was determined to limit the corpus to the most relevant content. Even if the Q&A approach can distinguish between relevant and irrelevant information, the benefits of large corpus are far less than the burden of running the model. The idiom "Floods and beasts" was determined after analyzing CNKI news data, and other intrusive idioms are rarely seen.

We agree that the model is strongly influenced by the choices for the training data. In constructing the training set, we randomly selected samples rather than using data from consecutive years or a single newspaper source, aiming to help the model learn more diverse features.

Moreover, we realized this is an important suggestion, and we are currently experimenting with different random sample combinations for cross-validation. If possible, we will include the cross-validation results in the revised version.

**Comment 4:** *Reading through the methodology it seems like a lot of manual preprocessing is still required, including manually annotating news texts. Ow much of a bottleneck is that for operational purposes is that, if you really will have a constantly updating database? This requires some discussion since it directly impacts the applicability of this dataset.*

Thank you for raising this important point. The manual preprocessing, including the annotation of news texts, is primarily required during the initial fine-tuning of the model, as well as for adjusting hyperparameters. This step is essential for ensuring the accuracy and effectiveness of the model. However, once the model has been trained, it does not need to be retrained for future applications. Future data can be directly processed into the test set format and used as input for the model without additional manual preprocessing. Therefore, this process does not represent a significant problem for the operational application of a constantly updating database.

**Comment 5:** *L 235: the exclusion of any texts wit the word 'will' seems like it can introduce giant margins of error. I get the reasoning to exclude forecasts, but if 'will' is used in a different context in a text that is actually related to flooding (e.g. 'damaged roads will re-open in 4 days') is then the whole text still excluded?*

This screening measure is based on BERT's answer to what disaster event happened, which is a classification of the disaster events described in the news. For example, the statement you mentioned, 'damaged roads will re-open in 4 days,' is related to the impact of the event and would not typically appear in the answer content. The responses to the Question 1 are focused on the type of the event, and they are usually very brief and do not include the details of the event. Therefore, this exclusion will not introduce big biases.

**Comment 6:** *The choice of GDP as a clustering method is odd to me. Why not use population density instead? That does correlate somewhat with GDP (so you still get reports on economic losses) but the loss of human life also hugely matters in disaster reporting, I'd think.*

Our initial motivation for conducting the GDP clustering analysis was to explain how regional economic development might influence the biases in media data. However, after carefully considering the reviewers' comments and reviewing literature on media communication themes, we have decided to remove this section. Relying solely on economic development or population density to explain the biases in media data is not convincing enough. In the revised version, we will

modify our explanation of the biases introduced by media data as follows:

From the perspective of media communication studies, agenda-setting theory posits that by choosing which events to report on, the media effectively signals to the public which issues are important (Leidecker-Sandmann et al., 2023). Through the quantity and depth of coverage, the media can shape the level of public attention given to certain events. In the context of disaster reporting, the government may influence the direction of media coverage to control public attention on specific disasters (Bai, 2022). For example, during the COVID-19 pandemic, research on government crisis communication showed that media agenda-setting was significantly influenced by government press conferences (Hayek, 2024). Crisis communication theory further explains the government can swiftly steer public opinion in the aftermath of a disaster, reducing the spread of negative emotions and maintaining social stability (Zhou et al., 2023). As a result, the variability in disaster reporting by the media may be influenced by multiple factors, including government policies, public interest, and the media's own resource allocation, leading to a situation where the volume of media reports is not necessarily consistent with the actual number of disaster events.

Moreover, we will add the analysis of the flood trend in different population density and economically developed areas to provide insights from an urban and social perspective as following figure:

[Figure]

Figure x. The analysis of flood event trends across Chinese provinces from 2000 to 2022, shown in relation to (a) population density and (b) Gross Regional Product (GRP).

Overall, most provinces exhibit an increasing trend in flood events, particularly in the northern, and western regions of China. These areas, including provinces such as Heilongjiang, Shandong, and Chongqing, are characterized by varying levels population density, both higher and lower, according to Figure x(a). As for the trends in relation to economic output in Figure 3(b), the provinces with increasing flood trends are mostly those with lower to moderate GRP, such as

those in the northern and western parts of China, despite Shandong and Zhejiang. These regions may not have received the same level of economic investment in flood control infrastructure as the more developed eastern provinces, which might explain the rising trend in flood events.

**Comment 7:** *Figure 6: This figure doesn't seem too relevant to the paper to warrant inclusion. A typhoon is certainly going to lead to flooding but the spatial scale is so wide that it's not a great verification in my opinion.*

We also agree that typhoons undoubtedly cause flooding, but the spatial scale is indeed quite large. Our intention in using Figure 6 was to demonstrate that our dataset successfully identifies disaster areas affected by typhoons, serving as evidence that our dataset can capture events with a broad impact. When considering the biases inherent in news data, a review of other literature revealed that the variability in disaster reporting by the media may be influenced by multiple factors, including government policies, public interest, and the media's own resource allocation, leading to a situation where the volume of media reports is not necessarily consistent with the actual number of disaster events. We speculate that this bias in media data is mainly due to the neglect of smaller-scale or less severe events. Therefore, we used these two significant cases to illustrate that larger-scale events are still likely to be reported. However, we understand that this figure may not adequately demonstrate the accuracy or completeness of our data. If it does not fit well with the structure of the paper, we are willing to move this analysis to the supplementary materials.

**Comment 8:** *Figures 8 and 9: occurrence is here shown without any distinction of severity of flooding, whereas the latter one might be more relevant for actual use of the dataset.*

Thank you for your suggestions regarding Figures 8 and 9. These figures represent the heatmaps of flood occurrences and the number of flood-related news reports by year and month. The primary purpose is to illustrate the temporal distribution of flood events across China. These visualizations help to highlight the years and seasons during which floods are most frequent, offering insights into the timing of flood events over the study period.

We acknowledge that distinguishing the severity of flooding could enhance the relevance of the dataset for certain applications. However, our current dataset does not provide detailed information on the severity of the flooding. Because this information is expressed with more variability and is more unstructured across different news sources, we will try to re-train the model or introduce new methods for extracting this type of information in future research.

**Comment 9:** *L230: I don't understand what the authors mean with '3 epochs' and*

*a learning rate of 5 x 10^-5. Please elaborate.*

The term "3 epochs" refers to the number of times the entire training dataset is passed through the model during the training process. In our case, we trained the model for 3 epochs, meaning the dataset was fed into the model three times, which is a standard approach to ensure that the model learns the patterns effectively without overfitting.

The learning rate of 5 x 10^-5 is a hyperparameter that controls the step size at each iteration while moving toward a minimum of the loss function. A smaller learning rate, such as the one we used, allows the model to converge more slowly and steadily, reducing the risk of overshooting the optimal parameters. This value was chosen based on preliminary experiments to balance the learning speed and model performance.

We will clarify these points in the revised manuscript to make them more understandable.

**Comment 10:** *L 275: 'verify and revise': is this part of the preprocessing? What does this mean, exactly?*

The process of "verifying and revising" is not part of the preprocessing but rather a post-processing step. The location names are generated by the BiLSTM-CRF model, and since the data spans from 2000 to 2022, it includes periods during which several regions in China underwent administrative adjustments or renaming. To ensure accuracy and relevance when associating these locations with the administrative division shapefile for spatial visualization in ArcGIS, I updated the names to reflect the most current administrative divisions. This step was crucial for maintaining consistency and ensuring that the visualizations accurately represent the latest geographical boundaries.

**Comment 11:** *Figure 4: Any idea what causes the large biases? This is hardly discussed.*

Identifying specific biases is challenging because the *China Flood and Drought Bulletin* provides only the total number of flooded cities, without listing specific locations. This limitation makes it impossible to pinpoint which specific events or regions are underreported in our dataset.

However, we can hypothesize that the biases stem from the intrinsic characteristics of news data. Some previous studies have also reflected the bias of constructing disaster catalogs with reports (Gall et al., 2009; Kron et al., 2012). From the perspective of media communication studies, agenda-setting theory posits that by choosing which events to report on, the media effectively signals to the public which

issues are important (Leidecker-Sandmann et al., 2023). Through the quantity and depth of coverage, the media can shape the level of public attention given to certain events. In the context of disaster reporting, the government may influence the direction of media coverage to control public attention on specific disasters (Bai, 2022). For example, during the COVID-19 pandemic, research on government crisis communication showed that media agenda-setting was significantly influenced by government press conferences (Hayek, 2024). Crisis communication theory further explains the government can swiftly steer public opinion in the aftermath of a disaster, reducing the spread of negative emotions and maintaining social stability (Zhou et al., 2023). As a result, the variability in disaster reporting by the media may be influenced by multiple factors, including government policies, public interest, and the media's own resource allocation, leading to a situation where the volume of media reports is not necessarily consistent with the actual number of disaster events.

Although our data consistently underestimates the number of flooded cities each year, likely due to the influence of the factors mentioned above, the trend in our data closely follows the overall temporal pattern observed in the *China Flood and Drought Bulletin*. This suggests that our dataset can still be effectively used to explore variations in flood events across regions. Then, it can be leveraged to analyze potential influencing factors of these changes, such as socioeconomic changes, climate change, alterations in land surface characteristics, and modifications in flood control measures, ultimately providing recommendations for flood management.

On the other hand, in future research, we plan to incorporate more available data sources to continuously update and validate this dataset. By expanding the dataset's coverage and adding descriptions of damages, its comprehensiveness will be improved.

Gall, M., Borden, K. A., and Cutter, S. L.: When Do Losses Count?: Six Fallacies of Natural Hazards Loss Data, Bulletin of the American Meteorological Society, 90, 799–810, https://doi.org/10.1175/2008BAMS2721.1, 2009.

Kron, W., Steuer, M., Löw, P., and Wirtz, A.: How to deal properly with a natural catastrophe database – analysis of flood losses, Natural Hazards and Earth System Sciences, 12, 535–550, https://doi.org/10.5194/nhess-12-535-2012, 2012.

Leidecker-Sandmann, Melanie & Koppers, Lars & Lehmkuhl, Markus. (2023). Correlations between the selection of topics by news media and scientific journals. PloS one. 18. e0280016. 10.1371/journal.pone.0280016.

Bai, Sheng. Mainstream Media Agenda Setting in Disaster Events. Journal of Emergency Management and Disaster Communications 3, no. 2 (2022): 83-98.

https://doi.org/10.1142/S2689980922500038.

Hayek, Lore. (2024). Media Framing of Government Crisis Communication During Covid-19. Media and Communication. 12. 10.17645/mac.7774.

Zhou S, Yu W, Tang X, Li X. Government crisis communication innovation and its psychological intervention coupling: Based on an analysis of China's provincial COVID-19 outbreak updates. Front Psychol. 2023 Jan 26;13:1008948. doi: 10.3389/fpsyg.2022.1008948. PMID: 36778169; PMCID: PMC9909028.

---

## Author Comment (AC3)

**Comment 1**:*The authors provide a comprehensive introduction to existing natural disaster datasets that record flood events created by official sources, other governments, or organizations. However, the manuscript would benefit from a more detailed discussion on how this study specifically addresses the gaps in these existing datasets. It is essential to clearly state the novelty and significance of your work in the context of existing datasets. For instance, do the deficiencies in these existing datasets affect the analysis, modeling, and prediction of flood events to some extent? How does the new dataset you have developed alleviate these issues at both theoretical and practical application levels?*

We will add more statements of the novelty and significance of our dataset in the revision: We construct the first county-level urban flood inventory across China from 2000. The existing datasets cannot present the floods distribution at county-level or provide the information across China. The most comprehensive and authoritative data on flood disasters published annually in China is found in the *China Flood and Drought Bulletin*. However, this data primarily focuses on the economic losses, casualties, and agricultural damages at the provincial level. While the Bulletin provides an overall description that includes the number of affected cities, it does not present the specific inventory of these cities. Our dataset shows trends that are largely consistent with those reported in the *Bulletin*, indicating that it can reliably reflect changes in flood events across different regions. Our data is originally at the county level and it can also be resampled and aggregated to city or provincial levels for research on flood dynamics and their influencing factors across different spatial scales.

**Comment 2**:*Line 143 "After a manual review to remove duplicates and irrelevant entries, including those referring to flash floods which occur suddenly in mountainous areas and are not the focus of this study, the final dataset consisted of 253 relevant news articles". The data preparation section needs more details. Please explain the criteria used for manually reviewing and removing irrelevant news articles from the CNKI database. Additionally, discuss any potential biases or limitations introduced by this manual selection process.*

We will explain more in the data preparation section:

The lead author manually reviewed the news data from CNKI. During this process, duplicate reports of the same flood events from different regional newspapers were removed, as well as articles containing search keywords but actually reporting on flood prevention measures, flood season warnings, and other non-flood events. Additionally, since this paper focuses on urban flooding, events related to flash floods and landslides were also excluded. The researchers in our group has double-checked this reviewing results.

Although we have conducted a double-check, manual interpretation may still introduce biases due to subjective differences in cognition. In future research, we

plan to incorporate large language models for correlation analysis or involve more domain experts to cross-validate the accuracy of our correlation assessments.

**Comment 3:** *Similarly, Line145 "These relevant news articles were then segmented into paragraphs and reorganized into 633 distinct samples. Among them, 503 samples were used to fine-tune the BERT model, alongside data from the CMRC2018 dataset, enhancing the model's stability to accurately extract flood disaster information. The remaining 130 samples served as a test set to evaluate the model's performance." Please clarify how the 503 samples were selected from the 633 distinct samples, and explain why the remaining 130 samples were used to evaluate the model's performance. This selection process is currently unclear and confusing.*

Thank you for your insightful feedback. Regarding the selection of the 503 samples from the 633 distinct samples, we would like to clarify that this process was done through random sampling, without any subjective selection. The intent behind this random division was to ensure that the training and testing datasets were representative of the entire dataset, thereby minimizing any potential bias.

The remaining 130 samples were designated as the test set after the training samples were randomly selected, without any manual intervention or predefined criteria. This allows for an independent evaluation of the model's ability to generalize to new, unseen data.

Additionally, we are currently experimenting with randomly selecting different training samples for retraining. If possible, we will describe the results of cross-validation in the revised version.

**Comment 4:** *For the identification of flood locations, I have a general question. From my understanding, news media reports about flooding occurrences typically mention the affected city or, at most, the district. However, actual urban flooding can occur at the street level or even smaller scales. Could you please provide a detailed explanation of how the BiLSTM-CRF model was trained and applied to recognize flood locations?*

Thank you for raising this important point. It is indeed true that news reports often only mention the affected city or district, with only some reports specifying smaller-scale locations like streets or buildings. However, our goal is to obtain results at the county or district level rather than more granular details.

To clarify our approach, we used the BERT model to identify the flood-affected areas mentioned in each news report firstly. The answer sentences typically include cities, counties or districts, and in some cases, specific streets or

buildings. After obtaining these mentions, the BiLSTM-CRF model was employed to extract the pure location names from BERT's output.

The BiLSTM-CRF model was trained using the MSRA named entity recognition corpus, a widely used dataset developed by Microsoft Research Asia, which contains a large amount of annotated Chinese sentences with named entities such as location names, person names, and organization names. Then, we standardized the spatial information by using county/district names (in China, counties and districts are the same administrative level and both included in cities).

**Comment 5:** *Regarding the performance of the BERT model (Table 4), it appears that the authors have only examined results based on a binary classification (flood vs. non-flood). If this is the case, the task seems too simple and lacks sufficient novelty. Could the authors also provide an evaluation of the model's performance in identifying the time and location of flood events?*

Table 4 does not only present the results of categorizing events as flood or non-flood. Only the first row is for classification. The second row shows the evaluation results of spatiotemporal information extraction. Two evaluation systems were used for the recognition of time and location information. This is explained in Section 3.1.3, Evaluation Metrics:

"The first index is called exact match, which measures the matching degree between the prediction and ground truths. The score is 1 for the EM of both the time and location information extracted. Otherwise, the score is 0. There is usually more than one disaster location in one flood event and maybe the model can output several but not completely accurate locations. Therefore, a fuzzy match was used to evaluate the location extraction using precision, recall, and F1 score. Unlike the classical formula, the precision and recall were calculated as:

$$Precision = \frac{P}{M}$$

$$Recall = \frac{P}{N}$$

Where P represents the number of accurately extracted flood locations, M is the total number of predicted flood locations and N is the total number of actual flood locations observed in the texts."

Table 4. The performance of the BERT model (EM index was not applied to evaluate event identification)

|  | Precision | Recall | F1-score | EM |
|---|---|---|---|---|
| Flood-event identification | 0.98 | 0.98 | 0.98 | N/A |
| Flood-information extraction | 0.96 | 0.78 | 0.86 | 0.82 |

**Comment 6:** *It seems that the number of identified flooded cities is significantly underestimated by the news media compared to the China Flood and Drought Bulletin (Figure 4). The authors suggest this discrepancy is related to the low attention given to low GDP areas. However, this raises a significant concern about the reliability of the developed dataset. As mentioned in section 4.3, the dataset records urban flood events reported in news articles from 2000 to 2022. If the news media is so inaccurate that it fails to record a large number of flood events, how can the authors ensure the reliability of the data generated from these news sources?*

We visualized the year-on-year difference in the number of flooded cities between the two datasets to provide a more intuitive representation of interannual variations. As shown in the figure below, the trend in our data is largely consistent with that reported in the Bulletin. This suggests that our dataset can reliably reflect changes in flood events across different regions, though news media may underestimate the number of cities affected by floods.

[Figure]

On the other hand, while the *China Flood and Drought Bulletin* provides a summary of the number of flooded cities, it does not offer a detailed inventory of specific locations. The spatial distribution of flood disaster loss information in the bulletin is limited to the province level, which encompasses multiple city-level areas. There may be biases inherent in the news data, but we contend that our dataset serves as a valuable reference in the absence of more detailed and comprehensive data sources.

In the future, we will introduce more data source to improve the data coverage, such as social media data, available disaster reports in some cities or provinces and so on.

**Comment 7**:*Figure 6 is not directly related to your results, I think you can put it into supplementary materials.*

We will put this part into supplementary materials.

---

## Author Comment (AC4)

**Comment 1:** *It's a bit surprising that this work is still based on BERT and doesn't mention anything about the emerging large language model (LLM) techniques (e.g., GPT-4). Please comment on this choice and discuss potential improvements if newer techniques could be used.*

When we designed this study, emerging LLMs had not yet developed to their current impressive state. Later, we compared our approach with GPT-3.5, and found that the accuracy of information extraction from the same test corpus was similar. On the other hand, considering this might be a project requiring long-term maintenance, we decided to continue using BERT, an open-source model that performs adequately, rather than switching to another model.

Moreover, we will add the discussion of potential improvements if newer techniques could be used as follows:

With the rapid advancement in large language models (LLMs), newer models such as GPT-4 offer significant improvements in various NLP tasks, including text classification, question-answering, and text generation, achieving state-of-the-art results. For instance, Colverd et al. (2023) have successfully used several LLMs, including GPT-3.5, GPT-4, and PaLM-Text-Biso, to generate flood disaster impact reports by extracting and curating information from the web. They found a notable correlation between the scores assigned by GPT-4 and human evaluators when comparing generated reports to human-authored ones. Furthermore, Hu et al. (2023) proposed a method fusing geospatial knowledge of locations with GPT models to extract location descriptions from disaster-related social media messages, demonstrating a 40% improvement over typically used Named Entity Recognition (NER) approaches. Given these advancements, our future research will explore the use of LLMs to extract nuanced information from flood-related text data, which includes distinguishing flood types, causes, and the specific losses associated with each flooding event.

**Comment 2:** *Given the focus of this dataset on cities, the analysis of the contributed dataset seems somewhat less pertinent. For instance, the large-scale climate zone analysis is rather off-topic. Instead, one would expect to see if such a dataset could be linked with urban-specific features (e.g., built-up area, urban volumetric density, GDP) to reveal more city-scale findings.*

We agree that our dataset should be more linked with urban-specific features. We will add the population density and regional GDP-related analysis in the revision as follows:

[Figure]

Figure x. The analysis of flood event trends across Chinese provinces from 2000 to 2022, shown in relation to (a) population density and (b) Gross Regional Product (GRP).

The background maps display average annual Gross Regional Product (GRP) in billion USD and population density in people per square kilometer, respectively, with darker shades indicating higher values. Overlaid on these maps are Theil-Sen estimated trends for the number of flood events, where the direction of the triangle represents whether the trend is increasing or decreasing, and the size of the triangle corresponds to the magnitude of the trend. Provinces without a significant trend are not marked.

Overall, most provinces exhibit an increasing trend in flood events, particularly in the northern, and western regions of China. These areas, including provinces such as Heilongjiang, Shandong, and Chongqing, are characterized by varying levels population density, both higher and lower, according to Figure x(a). The provinces that exhibit a decreasing trend in flood events are primarily located in the central and southeastern regions, particularly in provinces like Jiangsu, Fujian, and Guangdong, which are notable for their higher population densities. This suggests that the rising flood events are not strictly tied to population density.

As for the trends in relation to economic output in Figure x(b), the provinces with increasing flood trends are mostly those with lower to moderate GRP, such as those in the northern and western parts of China, despite Shandong and Zhejiang. These regions may not have received the same level of economic investment in flood control infrastructure as the more developed eastern provinces, which might explain the rising trend in flood events. On the other hand, the central and eastern provinces showing a decreasing trend, such as Jiangsu, Guangdong, and Sichuan, are among the most economically developed in China. This suggests that the availability of economic resources has allowed for more comprehensive flood management strategies, reducing the frequency of flood events in these areas.

It is important to note that several provinces with high population densities and significant economic development, specifically Jiangsu and Guangdong, exhibit a decreasing trend in flood events. These regions have experienced a high number of flood events over these years, with a notable peak around 2010. The estimated decrease in flood trends may be related to this peak, where the number of flood events was significantly higher than in other years, possibly skewing the trend calculations downward. Additionally, as regions frequently affected by flooding and characterized by high economic output and population density, substantial investments in flood management infrastructure and policies may have been made, also contributing to the observed decline in flood events. Jia et al. (2022) have highlighted the significant investments in flood management infrastructure in China's economically developed regions. They compared the 1998 and 2020 floods in Yangtze River Basin regions, which are economically developed regions in China. Their analysis reveals that significant improvements in risk management, including engineering defenses, environmental recovery, forecasting and early warning, and emergency response have led to a substantial reduction in flood disaster losses in Yangtze River Basin regions.

Jia, Huicong & Chen, Fang & Pan, Donghua & Du, Enyu & Wang, Lei & Wang, Ning & Yang, Aqiang. (2021). Flood risk management in the Yangtze River basin —Comparison of 1998 and 2020 events. International Journal of Disaster Risk Reduction. 68. 102724. 10.1016/j.ijdrr.2021.102724.

**Minor Comment 1:** *Line 375: "Lanzhou Province" - Lanzhou is \*\*not\*\* a province but the capital city of Gansu Province.*

Sorry for this mistake. We will address this issue in the revision.

**Minor Comment 2:** *The dataset should be archived more appropriately following the FAIR principle as suggested by reviewer 1. In addition, the GitHub repo needs more necessary README info, such as a description of the dataset, citation, etc. Also, `xlsx` is not recommended for simple tabular formats—please consider publishing this dataset in `csv` for better accessibility to allow better open research.*

Thanks for this suggestion. We will change the dataset sharing website to Zenodo, which is an open-access repository that allows researchers to share and preserve their datasets. It is operated by CERN and OpenAIRE and provides features like DOIs for citations, which supports the FAIR principles. We will improve our README file to describe the dataset including the data source, the data resolution, time span, and so on. In addition, we will change the data files into 'csv' format.

---

## Author Response (AR1)

**Dear** Referee #1,

Thank you very much for your time involved in reviewing the manuscript and providing valuable feedback. Those comments are constructive for revising and improving our manuscript. All the revisions related to your comments are noted in red in the marked-up manuscript. Our responses to your comments are as follows:

**General Comment 1. Flood Query Keywords**

**The flood query was limited to "flood" and "flood disasters" (L142, L154), while many other terms could hint at flood events in news items, e.g.,**

**"typhoon," "cyclone," "mud," "heavy rainfall," "inundated areas,"… Query terms are an essential aspect of event detection and this could be seen as a restriction limiting the detection power of the proposed approach. It raises some questions: Should this be documented as a limitation? Is it a decision to limit the size of the corpus? Does the Q&A approach prevent that**

**concern?**

Thanks for bringing up this important point. However, the other keywords included may raise the dataset too large. For example, we tried using "heavy rainfall" as the query term and found that only around 7% news returned reported flood events. Most of these news texts are related to meteorological early warning information. Therefore, the current query was determined to limit the corpus to the most relevant content. Even if the Q&A approach can distinguish between relevant and irrelevant information, the benefits of large corpus are far less than the burden of running the model.

We have added the explanation in Lines 119-125:

*"Although other meteorology-related terms such as "typhoon," "cyclone," "heavy rainfall," may also be associated with flood events, there were few cases where flood-related news mentioned only flood-causing terms like typhoon. For instance, a separate query using the term "heavy rain" yielded only about 7% relevant reports on actual flood events, with the majority of results being*

*meteorological warnings. To ensure a relevant dataset and improve model efficiency, this study limited the search terms to those most directly related to flooding."*

**General Comment 2. Flood Types and Multi-Hazard Concerns**

**The paper focuses on urban floods, excluding other types of floods, yet**

**flood types are interrelated and very often not mutually exclusive. Hence, referring, for instance, to the Hazard Information Profiles (HIPs, https://www.preventionweb.net/drr-glossary/hips ), an urban flood could also be related to a flash flood (despite the exclusion of the query of "flash**

flood," L151), a riverine flood, a coastal flood, a groundwater flood. **Floods are also secondary hazards associated with other hazards, such as a flood that could result from a Typhoon, heavy rainfall, a storm surge, an intense monsoon etc. Floods are also associated with geo-hazards such as landfall (See GLC studies). I found the Typhoon case study in the paper interesting. It also illustrates the multi-hazard nature of floods well. As in GLC studies, I would be interested in having the authors' view on multi, cascading, and co-occurring type issues, the possibilities of detecting multi-type floods, and the challenges, limitations, and perspectives concerning their proposed approach.**

Though we agree with this perspective, this article mainly focuses on urban flooding, especially its temporal and spatial information. This is a valuable point raised by the reviewer. So, in the revised version, we have added further discussion on this issue in Lines 413-422:

*"Recognizing that urban flooding often occurs in conjunction with other disasters, recent studies have attempted to extract multi-hazard information from news media reports. However, most of these studies use rule-based methods for classification, rather than analyzing causal relationships between disasters or subdividing floods into specific types. For instance, Yang et al. (2023a) applied a rule-based approach to extract 15 types of disaster information from news texts, categorizing reports based on specific disaster terms and matching location information using prefecture-level administrative names. Similarly, Liu et al. (2018) used keyword positioning and rule-based named entity recognition (NER) to identify disaster types and locations in news reports. In both cases, a report mentioning multiple disaster types is considered indicative of multi-disaster co-occurrence, but this approach can introduce biases if the mentioned hazards are unrelated. Future research should explore the use of language models to enhance extraction of diverse flood types and related hazards from news data, potentially increasing accuracy by identifying causal links."*

**General Comment 3. A More Balanced Discussion: Trend Analyses vs. Gap Filling Potential**

**The manuscript extensively discusses spatiotemporal trend analysis, necessitating more caution and clarity on trends influencing factors. I understand the need to illustrate trends in the resulting dataset, but, in my opinion, this matter could be more efficiently summarized, and the paper could be more descriptive and less assertive in the interpretation. Some analyses are simplistic and do not go deep enough. Rather than make the paper even longer, I invite the authors to distinguish more between the essential and the accessory and, if anticipated, to cover in greater depth the spatiotemporal analysis of events and cross-referencing with third-party data in other papers (see GLC studies).**

**Some figures may be grouped, e.g., maps in different pannels of one figure, allowing not only to focus on the trends of the output data but also on how the output data compares to other datasets, which is currently limited to Figure 4, despite the numerous datasets being listed in the introduction. The reader has little clue as to what gap is being filled. In particular, the**
**Chinese bulletin appears as a more exhaustive dataset (although coarser). This point may be worth further discussion.**

**Note regarding temporal trends:**

**Trends in hazard occurrences are complex, influenced by variations in**
**hazard intensity and alteration of environmental susceptibility, as well as demographic shifts that alter exposure or vulnerability. Moreover, climatic cycles (e.g., ENSO or other climate indices) can distort linear trend estimations over brief periods due to their cyclical nature.**

**The complexity is further compounded when analyzing trends from news**
**data. Changes in reporting capacity, especially in remote areas, along with new communication technologies like satellite and social media, may introduce significant biases. The proliferation of the internet during the 1990s and 2000s has notably impacted flood event reporting (Gall et al., 2009; Kron et al., 2012; Delforge et al., 2023). Kron et al., 2012 illustrate well**
**the challenges in building a hazard database with flood examples. These works underscore the necessity for standardized flood event definitions to mitigate discrepancies in reporting scales. In the case of news scraping, the framing by journalists can significantly alter the perceived frequency, spatial representation, and the type of events.**

**In conclusion, the total number of flood events is a highly relative figure. It is essential to acknowledge that while flood hazards are natural phenomena, flood disasters and their reporting are social phenomena with potentially distinct and diverging trend patterns. Given these complexities, attributing trends depicted in the news (i.e., social variables, not physical**
**ones) to climate change or land use changes requires careful consideration.**

We greatly appreciate the reviewer's detailed and insightful feedback. Your comments are invaluable in refining our analysis and ensuring our conclusions are both accurate and impactful. In response to your comments mentioned
above, we have taken the following considerations:

First, regarding distinguishing more between the essential and the accessory, we have decided to focus on highlighting the characteristics of the spatial distribution while streamlining the discussion of temporal trends, particularly simplifying the analysis of the influence of natural factors. In addition, our study focuses on urban floods, and the fundamental data is derived from news reports, which have a strong social dimension. Therefore, it is necessary to analyze the flood trend in different population density and economically developed areas to provide conclusions from an urban and social perspective. We have included this information in the revised version, with a detailed explanation provided in the latter part of this response.

Second, cross-referencing with third-party data in other papers or comparing to other datasets is challenging because of the absence of proper data. Therefore, we can only find some relevant data for comparison in certain regions. We have created a line chart for reference (Figure 1 below), to analyze the correlation of the direct economic losses provided by the Guangxi Provincial Government website due to floods after 2016, and the scale of disaster represented by the number of news-extracted flood-affected counties. These two indicators exhibit relatively consistent trends, which can to some extent suggest that the coverage of news data in certain regions is fairly good. However, these two indicators do not represent the same physical quantity, we think this figure may not suit for inclusion in the main text.

[Figure]

Figure 1. The time series of the number of news-extracted flooded counties and direct economic loss in Guangxi from 2016-2022.

Third, regrading what gap we have filled, it should be explained first that the *China Flood and Drought Bulletin* only provides the number of flooded cities in a general overview paragraph, without presenting their spatial distribution or specific inventory. The spatial distribution of flood loss information in the bulletin is limited to the province level, which encompasses multiple city-level areas.

While our dataset is not comprehensive, it is the first county-level dataset on a national scale, and its time trends are largely consistent with authoritative data.

As for the temporal analysis, we agree that there are inherent limitations to using media data for temporal analysis.

Overall, we have made the following adjustments in the revision:

In Section 5.1, Main Findings (Lines 398-405) and the relevant part of other sections (Lines 306-317), we have revised our statements on the temporal trends to reduce subjective interpretations and clarify the biases introduced by media data:

[revised manuscript text omitted]

*"Regarding the societal aspects, several provinces with high population densities*
*and significant economic development, specifically Jiangsu and Guangdong, exhibit a decreasing trend in flood events. These regions have experienced a high number of flood events over these years, with a notable peak around 2010. The decrease in floods since may be related to this peak. Additionally, as regions frequently affected by flooding and characterized by high economic output and*
*population density, substantial investments in flood management infrastructure and policies may have been made, also contributing to the observed decline in flood events. Jia et al. (2022) have highlighted the investments in flood management infrastructure in China's economically developed regions. They compared the 1998 and 2020 floods in the Yangtze River Basin regions, which*
*are economically developed regions in China. Their analysis reveals that improvements in risk management, including engineering defenses, environmental recovery, forecasting and early warning, and emergency response have led to a substantial reduction in flood disaster losses in Yangtze River Basin regions."*
**General Comment 4. Analyses of GDP**

**The manuscript highlights the GDP as the primary driver of media attention. However, the boxes in Figure 5 do not seem to show any significant difference between the occurrence of floods for different GDP groups. So,**
**to highlight a possible effect of GDP on media attention, it is vital to use GDP per capita (see GLC studies).**

**The population is a critical factor in media attention and hazard exposure. More densely populated cities should receive more media attention in the event of a flood. It is likely the primary factor explaining the spatial patterns**
**in the dataset. It is likely to be correlated with GDP, as well as other factors such as elevation, distance to river or coast, or climate (see G5). Therefore, controlling that factor when investigating some effects is essential.**

We agree with your perspective. Our initial motivation for conducting the GDP clustering analysis was to explain how regional economic development might influence the biases in media data. However, after carefully considering the reviewers' comments and reviewing literature on media communication themes, we have decided to remove this section. Relying solely on economic development or population density to explain the biases in media data is not convincing enough. In the revised version, we have modified our explanation of the biases introduced by media data as mentioned in the response to G3.

Moreover, we added the analysis of the flood trend in different population density and economically developed areas as mentioned in the response to G3.

**General Comment 5. Analyses of Flood Susceptibility**

**Figure 7 and the underlying analysis of flood susceptibility present some issues and do not bring much to the paper. The proposed pattern is not very neat (the points also overlap with no transparency), likely because the chosen indicators are quite remote proxies of flood susceptibility and should not be presented as acknowledged indicators in hydrology (the supporting references are weak).**

**Average daily precipitation depicts a hydrological equilibrium rather than an extreme event. Naturally, arid regions are less susceptible (also less populated, hence, exposed). However, the indicator becomes less relevant to other hydrological systems with higher precipitation averages (a mixture of blue and red dots). Likewise, elevated areas are also likely to be less populated and then less exposed, and the elevation effect tends to disappear at a lower elevation. Flow accumulation or topographical wetness indices could have been more reliable indicators of flood susceptibility.**

**I would recommend removing this analysis given its low informative value and also because these variables are related to climate variability, which is already pictured in Figure 12. See GLC studies for comparisons.**

Thank you for pointing out the issue with the selection of flood susceptibility factors. We agree that the factors initially chosen were not appropriate. Average rainfall reflects the general characteristics of a region, but flood disasters are often associated with extreme rainfall. Additionally, discussing the impact of elevation alone is not convincing given the large extent of the study area. We have removed this section in the revision.

**General Comment 6. Flood Events Dataset Resolution**

**While the final dataset is reported at the county-month level, the reader is**
**left with little insight into the level of detail directly resulting from the**
**information extraction process, which remains unclearly described. Based**
**on Figures 4 and 6, it appears that information at the city-daily level was**
**collected. It seems that a much more precise dataset could have been**
**shared without much additional effort, raising questions about the**
**motivation behind disaggregating the data to such a coarser level.**

We are sorry that our description may confuse readers especially the term
"county". First, we think the administrative level in China should be introduced:

The provincial level is the highest level of administrative division in China, and it
consists of: Provinces, Autonomous Regions, Municipalities, Special
administrative Regions (Hong Kong and Macau); The second level is prefectural
level including: Prefecture-level Cities (just cities in the usual sense),
Autonomous Prefectures, Leagues (found in Inner Mongolia); The third level is
county level including: Counties, County-level Cities (smaller cities under the
jurisdiction of a prefecture-level city), Districts, Banners (found in Inner Mongolia);
The forth level is township level including: Towns, Townships (typically more rural
areas), Subdistricts; And the last level is village level including: Villages,
Communities.

Therefore, a county is a finer administrative division than a city, with one city
typically comprising several county-level areas. We have also clarified this
information in Lines (62-66):

"*In China, the administrative structure consists of several levels: the highest is the*
*provincial level, followed by the prefectural level (i.e., cities in the usual sense),*
*and then county-level. Given China's vast area of approximately 9.6 million*
*square kilometers, flood characteristics exhibit significant spatial variability across*
*provincial and prefectural regions (Wang et al., 2013; Shang et al., 2023). With*
*around 2,844 county-level areas, each spanning roughly 1,000-3,000 square*
*kilometers, this scale offers a more granular perspective for analyzing flood*
*patterns across diverse locations.*"

The locations extracted from news reports typically include only the county-level
area name or the county name with the specific flooded street or building.
Therefore, we standardized the spatial information by using county names.

Second, most of the data can be extracted to specific day information, but some
can only be extracted to month, so at first, in order to unify the data set, we set
the time resolution as month. In the revised version, we have modified the events
with day information to be accurate to day.

As for the figures you mentioned, Figure 6 (original version) is indeed the flood events with daily information within two typhoon event months. However, in Figure 4 (original version), we used a line plot just to show the temporal trends of news-reported flooded cities amount and those reported in bulletins. The data is aggregated annually rather than daily.

**General Comment 7. Data Content, FAIR Principles, and Reusability**

**Also, given that a central outcome of the paper is a dataset, alignment with FAIR principles (https://www.go-fair.org/) should be particularly encouraged. Regarding the data shared, GitHub is not considered FAIR as it does not allow for persistent identifiers. Also, a few additional data could greatly increase the reusability of the dataset, e.g., precise column descriptions in the readme, the reference for the administrative unit shapefile to link the data with the post-code or administrative units as described in the paper (L275-278), using international time standards, and possibly translate region names to English to maximize reuse in the global context.**

**Regarding reproducibility, the data and code availability section could be improved. Input news data and their conditions of (re-)use are not described in this section. Tools and libraries being used to develop the approach are not referred to (except references to the Python "Re" module at L187). There is no comment about whether or not the developed models are accessible and under which conditions of use.**

**There are no links or references to the news articles that have been used to construct the dataset. Sharing the links could drastically increase the paper's outreach and support future research and NLP applications to extract additional information, such as flood impact variables or associated hazard types, without redeveloping an NLP flood event detection model. Annotated corpora are also valuable datasets in the context of NLP for future benchmarking. Consider commenting on that dataset as well.**

Thanks for your helpful suggestions. We have changed the dataset sharing website to Zenodo, which is an open-access repository that allows researchers to share and preserve their datasets. It is operated by CERN and OpenAIRE and provides features like DOIs for citations, which supports the FAIR principles. Furthermore, we have added a column describing post-code and precise column descriptions in the readme, and translate region names to English. As for the administrative unit shapefile to link the data with the post-code, we have added the reference in Section Data availability and shared it in the dataset.

About the input news data, we have checked the link of the news platform and re-shared it. Then, readers can retrieve the news using the query as we described in

Section News data and download the data according to the data management rules of the WiseNews platform. We could share the annotated flood-related corpora if readers contact us.

We have referred to the tools and libraries used to develop the approach, such as Tensorflow in the revised version. Besides, the code and trained model is
available from the corresponding author upon request.

The modified Section Data availability (Lines 480-486) as follows:

*"The national flood dataset constructed in this present study is accessible on Zenodo (10.5281/zenodo.14000094). The BERT model used in this study was based on pre-trained weights from Google's BERT repository*
*(https://storage.googleapis.com/bert_models/2018_11_03/chinese_L-12_H-768_A-12.zip). The following key libraries and tools were used: TensorFlow (v1.12), NumPy, Pandas and Scikit-Learn. Readers can retrieve the news using the query as we described in Section News data and download the data according to the data management rules of the WiseNews platform. The*
*population used in this study was provided by Landscan (https://doi.org/10.48690/1529167) and the GRP data was from China Statistical Yearbook. The code, trained model and the annotated flood-related corpora is available from the corresponding author upon request."*

**Specific Comment 1.    L8: "similar" could be more nuanced.**

We have re-organized this sentence to describe the comparison between news-based floods and bulletin data in Lines 9-11:
*"Our analysis reveals that the temporal trend of flooded cities in our news-based dataset broadly aligns with that in the China Flood and Drought Bulletin, despite notable differences in the magnitude of reported events during peak years."*

**Specific Comment 2.    L9:10: "the connection between…": the connection does not support accuracy and the analysis is oversimplistic (See G5).**

We agree that the flood susceptibility indicators were insufficiently appropriate. As response to Comment G5, we have removed this analysis in the revision.

**Specific Comment 3.    L43 (and after): "natural disaster" is a controversial**
**terminology often avoided by Disaster Risk experts, acknowledging that a disaster is not natural (as opposed to natural hazards).**

We have corrected it to 'natural hazard' thoroughly in the revised version.

**Specific Comment 4.    L43-L52: Table 2 could distinguish between catalogs from remote and social sensing, e.g., that DFO is based on remote**

**sensing, EM-DAT on the collection of text documents and manual extraction of the information. Some missing recent initiatives could be worth mentioning, e.g., a global remote sensing catalog is the global flood database and a global catalog obtained from social media:**

- **Tellman, B., Sullivan, J.A., Kuhn, C. et al. Satellite imaging reveals increased proportion of population exposed to floods. Nature 596, 80–86 (2021). https://doi.org/10.1038/s41586-021-03695-w**

- **J.A. de Bruijn, H. de Moel, B. Jongman, M.C. de Ruiter, J. Wagemaker, J.C.J.H. Aerts. A global database of historic and real-time flood events based on social media. Scientific Data, 6 (1) (2019), p. 311, 10.1038/s41597-019-0326-9**

- **G.R. Brakenridge. Global Active Archive of Large Flood Events. Dartmouth Flood Observatory, University of Colorado, USA. http://floodobservatory.colorado.edu/ Archives/ (Accessed xxx)**

- **Delforge, D., Wathelet, V., Below, R., Lanfredi Sofia, C., Tonnelier, M., Loenhout, J. van, and Speybroeck, N.: EM-DAT: the Emergency Events Database, preprint, https://doi.org/10.21203/rs.3.rs-3807553/v1, 2023.**

We have added a column to distinguish between databases from remote and social sensing and the recent datasets in Table 2 as follows:

| Data Source | Name | Period | Flood Records | Update Frequency | Source |
|---|---|---|---|---|---|
| Social sensing | The Emergency Events Database (EM-DAT) | 1900-- | Time, location and damage of global flood events that resulted in a certain number of deaths or economic losses | Continuously | Centre for Research on the Epidemiology of Disasters |
| | Natural Disaster Data Book | 2002-- | Statistical and analytical perspectives of flood events in Asia (data retrieved from EM-DAT) | Annual | Asian Disaster Reduction Center |
| | Global Flood Monitor | 2014-2023 | A real-time overview of ongoing flood events based on filtered Twitter data | Pause | IVM - VU University Amsterdam and FloodTags |

| | | | | | |
|---|---|---|---|---|---|
| | Floodlist | 2016-- | Dates, locations, magnitude and damages of each flood events based on news | Real-time | FloodList (funding from Copernicus) |
| Remote sensing | Dartmouth Flood Observatory (DFO) | 1985-- | Time, location and extent of global flood events using satellite observations | Continuously | University of Colorado Boulder |
| | Global Flood Awareness System | Real-time | Ongoing and upcoming flood events information from satellites to support flood forecasting at national, regional and global levels | Real-time | Copernicus Emergency Management Service (CEMS) |
| | Global Flood Monitoring System (GFMS) | Real-time | Flood inundation extent and depth based on precipitation satellite data and flood model simulation | Every 3 hours | University of Maryland and NASA |
| | The Global Flood Database | 2000-2018 | flood extent and population exposure for 913 large flood events | unknown | Floodbase |

**Specific Comment 5.** *L65: Beyond cloud cover for optical imagery, mapping urban flood is challenging per se.*

It is true that mapping urban flooding is inherently challenging, and we just listed one source of uncertainty. In the revised version, we have modified this sentence in Lines 68-70:

*"While remote-sensing images have the potential to infer disaster progression, mapping urban floods presents inherent challenges, such as uncertainties caused by cloud cover"*

**Specific Comment 6**. **L75: "Yang et al. (2023)" Such a paper of high relevance should be re-discussed later in the discussion section, among others, to identify (see Overview).**

Thank you for the helpful suggestion. As the response to G2, we have added the discussion on multiple hazards and reviewing these highly relevant papers.

**Specific Comment 7.    L77: The authors acknowledge the multi-hazard nature of floods here and after, but the issue is not discussed in light of their own work (see G2).**

We have added the discussion on the multi-hazard nature of floods in the revised version (see the response to G2)

**Specific Comment 8.    L90: "Conditional Random Fields (CRF) layer" appears to be a central part of the methodology appearing multiple times in the paper; however, it lacks a clear explanation of what it is and why it is used.**

Sorry for this unclear statement. CRF model is a type of discriminative probabilistic model used to predict sequences of labels for sequences of input samples. It considers the context (i.e., neighboring labels) to make more accurate predictions. The CRF layer was part of the named entity recognition (NER) method in our approach.

We used BERT to extract initial answers including spatiotemporal information of floods and then, adopted an NER method called BiLSTM-CRF model to identify the location names in the answers. In the NER model, a BiLSTM layer is adopted to extract features from the input character vectors. And then, the CRF layer uses the output from BiLSTM to compute the most likely sequence of labels considering the dependencies between labels.

We have added the explanation in Lines 228-231:

*"BiLSTM is a deep learning model that captures context information from sequence data, while CRF is a probabilistic graphical model used for sequence labeling that considers dependencies between labels. In the NER model, a BiLSTM layer is adopted to extract features from the input character vectors. And then, the CRF layer uses the output from BiLSTM to compute the most likely sequence of labels considering the dependencies between labels."*

**Specific Comment 9.    L110:116: since the paper follows a conventional structure, it is unnecessary to detail it in the introduction.**

We have removed these explanations in revised version according to your suggestion.

**Specific Comment 10.    Table 2: EM-DAT is continuously updated (see Delforge et al., 2023). I would also refer to the Global Flood Awareness System (https://global-flood.emergency.copernicus.eu/), the flood component of CEMS, instead of CEMS. See also S4.**

We have updated Table 2 according to your comments (See response to S4).

**Specific Comment 11.    L134: check url link (404 error).**

We have checked this issue and re-shared the data link in Line 111:
(https://www.wisers.com/wisesearch)

**Specific Comment 12.    Figure 1: I appreciate the availability of an example. However, consider selecting a more topic-appropriate example or asking for a where/when the question for more relevance.**

We have provided a more topic-appropriate example in the revised version as the
following figure:

**Context:** Faced with a once-in-50-years "7·23" catastrophic flood, the city committee and government treated the disaster as a command and time as life, mobilizing the entire city, resettling the affected people, and actively conducting post-disaster epidemic prevention. In this event, many houses in Luzhou's Linjiang were submerged, and at least 100,000 to 200,000 people needed to be relocated, which is unprecedented in Luzhou's recent decades of history.

**Question1:** What disaster event occurred?

**Answer1:**    "7·23" catastrophic flood.

**Question2:** When did it happen?

**Answer2:** "7·23".

**Question3:** Where did the disaster occur?

**Answer3:** Luzhou.

**Specific Comment 13.    L142, L151, and L154: See G1.**

The detailed explanation and modification are in the response to G1.

**Specific Comment 14.    L145-148: The description of the data and its**
**processing, including test/train split, may be confusing. It may be more appropriate to move to the method section.**

We have moved the description of the data processing to the method section.
Section 3.1 has changed into "Data Preparation".

**Specific Comment 15.   L157: "Validation" unless China Flood and Drought Bulletin is considered a gold standard, I think referring to comparative data and cross-comparison instead of validation is more appropriate.**

We agree with you. The title of Section 4.2 has changed into "Comparison of The Urban Flood Information".

**Specific Comment 16.   L168-L174: oversimplistic view of hydrology and weak references. See G5.**

We have removed this part.

**Specific Comment 17.   L190-199: This section could indicate the total/train/test sample sizes more clearly.**

Sorry for unclear explanation. The total of the CNKI news samples was 633, and these samples were divided into three parts: 402 samples for fine-tuning BERT (alongside with CMRC2018); 101 samples as validation set for adjusting hyperparameters; 130 samples for testing. We have removed the related descriptions from data section and re-organized the data-processing part in method section (Lines 160-163):

*"The CNKI news articles were then divided into 633 distinct samples. Of these, 503 were randomly selected as training samples, and the remaining 130 samples were set aside as test samples to evaluate the model performance. For training, 80% of the samples (402) were combined with the CMRC2018 dataset to fine-tune the BERT model, while 20% (101) were used for validation to optimize the model's hyperparameters."*

**Specific Comment 18.   L235: words should be singular in "and does contain the words 'will'…". Also, I wonder if this approach successfully separated actual events from forecasts? Is there any language specificity in Chinese invoved here?**

We don't think it is related to the language specificity in Chinese. This screening measure is based on BERT's answer to what disaster event happened, which is classification of the disaster events described in the news. The response refers to the type of the event, and if the event is a forecast, it should include the words representing future state.

**Specific Comment 19.   Figure 3: Is [SEP] a requirement given the specificity of the Chinese language?**

No, [SEP] is a special token used for BERT model not just for Chinese language tasks. In a Question-Answering (Q&A) task using BERT, the [SEP] token is essential. It separates the question from the context or passage from which the answer needs to be extracted. The typical input format for BERT in a Q&A task is: [CLS] Question [SEP] Context [SEP]

This structure helps BERT understand the boundaries and relationships between the question and the context, facilitating accurate extraction of the answer.

**Specific Comment 20.    L243: In the first sentence, correct "flood information extraction" into "(i) flood event detection and (ii) flood information extraction" for clarity.**

We have addressed this issue in Lines 245-246:

*"In this study, two evaluation metrics were used to assess the effectiveness of (i) flood event detection and (ii) flood information extraction"*

**Specific Comment 21.    L259: it is not clear to me how Exact Match behaves in case of multiple locations, zero if any error? What is it clearly meant by the location data? City? County? How is location handled before the flood location recognition is explained in section 3.2? Perhaps 3.2 should be explained before.**

Yes, if any one of multiple locations was not identified then the score for this sample is zero.

The location data specifically refers to county-level region name.

Sorry for unclear structure. We did not do any further processing of the answer information before flood location name recognition. We agree that Section 3.2 should be explained before and have addressed this issue in the revision. The current method is organized as follows:

Section 3.1 is Data Preparation, Section 3.2 is BERT Model Construction and Application, Section 3.2 is Urban Flood Location Recognition and Section 3.4 is Evaluation Metrics.

**Specific Comment 22.    L276: consider adding the reference of the used administrative unit shapefile. See also G7.**

We have added this reference in the revised manuscript as response to G7.

**Specific Comment 23.    L285, section 4.1. The performance seems good in an absolute manner, but the reader has no clue how this performs in**

**relation to the context of social sensing of flood or in the context of Chinese NLP. This is quite important to document.**

Thanks for pointing that out. We added the discussions on the performance of our NLP method in Lines 282-288:

*"The performance of BERT model in this current study are competitive within the broader field of information extraction and Chinese NLP. For instance, Yang et al. (2022) adopted a BERT-based model for Chinese named entity recognition (NER) and achieved 94.78% and 62.06% F1 values on the MSRA (created by Microsoft Research Asia, is a well-structured and annotated collection of text for*
*NER tasks) and Weibo (A Chinese social media platform) datasets, respectively. This significant disparity in performance highlights the challenges in semantic understanding in social media data compared to more structured datasets like MSRA. In addition, Kim et al. (2022) developed a question answering method for infrastructure damage information retrieval from textual data using BERT and*
*achieved F1-scores of 90.5% and 83.6% for the hurricane and earthquake datasets, respectively."*

**Specific Comment 24. Figure 4: Bulletin seems more exhaustive. This could be discussed more and the authors could highlight better complementarities between data collection approaches, e. g., how would**
**the proposed approach improve Chinese bulletin?**

The spatial distribution of flood loss information provided in the bulletin is only at the provincial scale, and the number of flooded cities is mentioned in the paragraph describing the overall extent of the disaster. However, it does not provide a specific list of flooded cities or time information of each event. We have
provided a more detailed list of affected counties. Additionally, we visualized the year-on-year differences in the data to offer a clearer view of interannual variations. As shown in the figure below, despite some degree of underestimation, the temporal trends in our data align closely with those reported in the Bulletin.

We have modified the comparison analysis and the discussion in Lines 295-301 and Lines 391-397:

*"To evaluate the news-based flood dataset, a comparative analysis was conducted using records from the China Flood and Drought Bulletin. The comparison of annual flooded cities from the China Flood and Drought Bulletin*
*with those identified in news sources between 2006 and 2018 is displayed in Figure 4(a). In addition, the year-on-year difference in the number of flooded cities between the two datasets was visualized to provide a more intuitive representation of interannual variations. As shown in Figure 4(b), the trend of the news-based dataset closely follows the overall temporal pattern observed in the*

*China Flood and Drought Bulletin. This suggests that the dataset created in this present study can reliably reflect changes in flood events across different regions, though news media consistently underestimate the number of cities affected by floods."*

[Figure]

*Figure 4. The comparison between the number of flooded cities extracted from news and China Flood and Drought Bulletin for each year. (a) The time series of both datasets; (b) year-over-year changes in both datasets, the y-axis showing the difference in values from the previous year. Positive bars indicate an increase compared to the prior year, whereas negative bars represent a decrease.*

*"The dataset created in this study serves as the first county-level urban flood inventory across China from 2000, addressing a gap in existing datasets that often fail to provide county-level flood distributions or coverage across the country. While China Flood and Drought Bulletin offers authoritative data on flood disasters, focusing on economic losses, casualties, and agricultural damages at the provincial level, it lacks detailed inventories for specific cities. Our dataset shows trends that are largely consistent with those reported in the China Flood and Drought Bulletin, indicating that it can reliably reflect changes in flood events across different regions. This dataset's county-level granularity also allows for resampling and aggregation to city or provincial levels, facilitating deeper analyses of flood dynamics and influencing factors at different spatial scales."*

**Specific Comment 25.     L298-L308: The analysis of media attention due to GDP biases is not significant and do not control for the population bias (see G4).**

We have removed the analysis of media attention due to GDP biases and detailed explanation is in response to G4.

**Specific Comment 26.     L313-314: The two case studies were selected as the author assumed a good coverage because of their important hazard magnitude and impact. This is a known bias and an issue worth mentioning, as small-impact disasters tend to be less well-covered and documented. See Kron et al., 2012, Gall et al. 2009, and Delforge et al. 2023 and references therein for more insights about hazard catalog biases.**

 We quite agree with you. We selected these two cases to show that our dataset can cover the more impactful events. However, it is true that small-impact events receive much less media attention, which is one of the limitations of our data set based on media data. We also appreciate the references you provided, and we have added the discussion of the bias caused by media data as the response to G3.

**Specific Comment 27.     L328-339 +   Figure 7. These selected indicators are bad proxies of flood susceptibility, and I do not see how this analysis validates something about the spatial distribution of floods (see G5). Consider removing.**

We have removed this part in the revision.

**Specific Comment 28.     L340: how the information was structured prior to harmonizing the data into the urban flood dataset is unclear. See also G6.**

The detailed description is in the response to G6.

**Specific Comment 29.     Figures 8 and 9, it would be great to have an additional column or a time series on the Y axis with the annual total. This could help identify pluriannual cycles as a result of climate indices. Consider adding the total number of occurrences and items in the figure caption.**

According to your suggestions, we have modified the figures as followings:

[Figure]

Figure 5. A heat map showing in each cell the number of flood occurrence for each month and each year

[Figure]

Figure 6. A heat map showing in each cell the number of flood occurrence for each month and each year.

**Specific Comment 30.     L354: "seasonality" instead of "climate's tendency" could be more appropriate.**

We have addressed this issue in Line 332.

**Specific Comment 31.     L390: "exposure" or "susceptibility" (the environmental side of vulnerability) is maybe more appropriate than vulnerability because the latter also encompasses social vulnerability.**

We agree with this comment; however, after considering your comment G3 and
adding an analysis of social factors, we decided to remove the basin-related analysis in order to avoid increasing the length of the manuscript. As a result, the entire paragraph where this sentence was located has been removed.

**Specific Comment 32.    Maps Figures 10, 11, and 12 could be grouped into a multipanel figure for conciseness. Consider adding population density as well since it drives hazard exposure. DEM and river networks may also be considered as information to include (parsimoniously).**

First, we have grouped the figures into a multi-panel figure as you suggested and included the population density and the Gross Regional Product (GRP) related analysis. The detailed description is in the response to G3.

Regarding the suggestion to include DEM and river networks, we appreciate the idea but believe these factors, while relevant, do not directly align with the primary focus of our analysis. Incorporating DEM and river networks would introduce additional complexity that may not substantially contribute to the core findings or enhance the validation of our flood distribution data. We agree with your opinion in G5 that in areas with very high elevation, the low population exposure naturally leads to fewer reported flood events, so conducting spatial analysis with DEM as the sole base layer does not have significant meaning. Similarly, the independent analysis of river networks is not particularly meaningful.

**Specific Comment 33.    L409: The comparison with other datasets is quite limited, and the Chinese bulletin seems more exhaustive if one can trace the original data. To what extent the proposed dataset fills gaps is thus not very well documented (see G1). Adding more than one catalog from Table 1 and 2 in Figure 4 for comparison can improve this discussion.**

There is no other Chinse national-level dataset describing the inventory of urban floods. The *Chinese Flood and Drought Bulletin* just shows the number of flooded cities for each year without specific flooded cities inventory and in recent years, even the numbers have not been published. Additionally, no other datasets from Table 1 and 2 could provide the number of flooded cities or counties across China so that we cannot add more than one catalog in Figure 4. The absence of such comparable data itself highlights that our dataset fills a gap in urban flood data on a national scale in China.

The spatial distribution of flood loss information in the bulletin is limited to the province level, which encompasses multiple city-level areas. Our dataset, despite its limitations, offers more granular information by identifying specific flooded areas at the county level, which is smaller than the city level. There may be biases inherent in the news data, but we believe that our dataset serves as a valuable reference in the absence of more detailed and comprehensive data sources.

**Specific Comment 34.    L473: The data availability section does not include the input news data accessibility information. In line with HESS recommendations and FAIR standards, I also encourage the authors to share information about code and model availabilities.**

The detailed explanation is in the response to G7.

**Specific Comment 35.    L414-L416: this sentence (and the section in general) looks like the authors do their best to fit in the context of climate change and urbanization, even excluding some peak values to retrieve a positive trend. Trends, in particular for disaster news, are much more complex than trends observed on physical variables and include important social drivers and biases. The discussion is oversimplified, and the authors should take more distance and inquire about the biases arising from social sensing of hazards. See G3 and references.**

We agree that trends for disaster news are much more complex than trends observed on physical variables and include important social drivers and biases. We have added the discussion on the biases arising from social sensing of flood hazard as the response to G3.

**Specific Comment 36.    L445: Perspectives are neither exhaustive nor detailed. Consider adding more relevant perspectives, differentiating those related to the method (NLP-detection, extraction) and those related to the valorization of the resulting dataset.**

We have re-organized the discussions of limitations and future directions on the method and the dataset according to your suggestion in Section 5.2, Limitations and Future work (Lines 424-454):

*"Despite the valuable insights provided by the spatial and temporal analysis in this study, there are several notable limitations. Our dataset contains information solely on the timing and names of the affected areas, lacking critical details such as the spatial extent, water volume, flood types, causes, damages, or multi-hazard information. This limitation arises from the nature of our data source, as we relied on news reports rather than scientific papers, which typically include such quantitative details. To address this limitation, we plan to incorporate additional data sources, such as disaster yearbooks from each province or city, to enrich our dataset with more comprehensive flood event details, particularly multi-hazard information.*

*Recognizing that urban flooding often occurs in conjunction with other disasters, recent studies have attempted to extract multi-hazard information from news media reports. However, most of these studies use rule-based methods for classification, rather than analyzing causal relationships between disasters or*

*subdividing floods into specific types. For instance, Yang et al. (2023a) applied a rule-based approach to extract 15 types of disaster information from news texts, categorizing reports based on specific disaster terms and matching location information using prefecture-level administrative names. Similarly, Liu et al. (2018) used keyword positioning and rule-based named entity recognition (NER) to identify disaster types and locations in news reports. In both cases, a report mentioning multiple disaster types is considered indicative of multi-disaster co-occurrence, but this approach can introduce biases if the mentioned hazards are unrelated. In future research, we intend to explore the use of language models to enhance the extraction of multi-type floods and related hazards from news data, potentially improving accuracy by examining causal links.*

*Furthermore, the approach used in the present study also has its limitations. We employed a BERT model fine-tuned by a Chinese corpus for question-answering tasks, which has proven efficient in information extraction. However, with the rapid advancement of large language models (LLMs), newer models such as GPT series offer significant improvements in natural language processing tasks. For example, Colverd et al. (2023) successfully used several LLMs, including GPT-3.5, GPT-4, and PaLM-Text-Bison, to generate flood disaster impact reports by extracting information from the web, finding strong correlations between LLM-generated and human-authored reports. Additionally, Hu et al. (2023) proposed a method that combines geospatial knowledge with GPT models to extract location descriptions from disaster-related social media posts, achieving a 40\% improvement over traditional NER approaches. Given these advancements, our future research will explore the use of LLMs to extract nuanced information from flood-related text data, which includes distinguishing flood types, causes, and the specific losses associated with each flooding event. On the other hand, BERT models require language-specific fine-tuning, which can limit adaptability across languages. In contrast, LLMs that adopt a zero-shot strategy (i.e., direct application without the need for fine-tuning) may solve the transferability problem.*"

**Specific Comment 37.    L473: data and code availabilities: see G7.**

The detailed explanation is in the response to G7.

**Specific Comment 38.    Table A2: Same as Figure 4. It may be removed, in my opinion.**

We have removed it in the revise version.

**Dear Referee #2,**

We have taken the time to think through all of your comments and carefully revised the manuscript as you suggested. All the revisions related to your comments are noted in **Bold** in the marked-up manuscript. Our responses to your comments are as follows:

**Comment 1: From the abstract, but also the rest of the paper, the level of detail that this flood information dataset has is unclear. A spatial scale is mentioned as 'county-level', but that can vary quite a lot depending on where the reader is from. Connecting this to a typical length scale (1, 10, 100, … kilometres?) will make it clearer to a potential end-user whether this dataset**
**is useful.**
**Similarly: what kind of information is present about the flooding? Is it just spatial extent? Or also indications of amounts of water, timing or duration, damages done, etc etc. This should be immediately clear from the first reading, in both the abstract, as well as the results section.**
**Related to this, table 1 is an overview of current flood disaster reports, which also doesn't contain any information on the kind of data that's in there. Giving both your, and the existing datasets that level of detail can make it clear what the advantage of this new methodology is in comparison to the existing ones. Also, the validation data described in section 2.3 suffers from**
**this lack of information.**

First, we would like to explain the relationship between the different administrative levels in China:

The provincial level is the highest level of administrative division in China, and it consists of: Provinces, Autonomous Regions, Municipalities, Special administrative Regions (Hong Kong and Macau); The second level is prefectural level including: Prefecture-level Cities (just cities in the usual sense), Autonomous Prefectures, Leagues (found in Inner Mongolia); The third level is county level
including: Counties, County-level Cities (smaller cities under the jurisdiction of a prefecture-level city), Districts, Banners (found in Inner Mongolia); The forth level is township level including: Towns, Townships (typically more rural areas), Subdistricts; And the last level is village level including: Villages, Communities. Regarding the specific spatial scope, the size of county-level administrative regions
ranges from 1,000 to 3,000 square kilometers.

This is important for enhancing the readability of the article and dataset. We have included a brief explanation of the spatial scope in Lines 62-66:
*"In China, the administrative structure consists of several levels: the highest is the*
*provincial level, followed by the prefectural level (i.e., cities in the usual sense), and then county-level. Given China's vast area of approximately 9.6 million square kilometers, flood characteristics exhibit significant spatial variability across provincial and prefectural regions (Wang et al., 2013; Shang et al., 2023). With around 2,844 county-level areas, each spanning roughly 1,000-3,000 square*
*kilometers, this scale offers a more granular perspective for analyzing flood*

*patterns across diverse locations. The county level, which encompasses counties, county-level cities, and districts, offers a more detailed context for analyzing flood phenomena."*

Second, our dataset provides information solely on the timing and names of the affected areas. Unfortunately, it does not include details such as the spatial extent, water volume, or damages caused by the flooding. This limitation arises from the nature of the data source—we relied on news reports rather than scientific papers, which typically do not provide the physical measurements or quantitative details
often found in more specialized studies. In the future, we will introduce more data source to update and correct the dataset and add these kinds of flood event details. We have explained this more in the abstract and results sections of the revised manuscript to ensure that readers understand the dataset:
In Section Abstract (Lines 7-9):

*"The dataset documents the timing and affected county areas of urban floods, revealing that a total of 2,051 county-level regions have been impacted, with 7,595 occurrences recorded."*

In Section 4.3, The Urban Flood Dataset (Lines 320-322):

*"The dataset records urban flood events reported in news articles from 2000 to 2022, including the timing of these events at the day level and the affected areas at the county level."*

Additionally, we have updated the flood record information in Table 1 like followings:

| Name | Period | Flood Records | Update Frequency | Source |
|---|---|---|---|---|
| Annual Report of Chinese Hydrology | 2021-- | Number of basin/river floods and flooded river list | Annual | Ministry of Water Resources of the People's Republic of China |
| China Flood and Drought Bulletin | 2006-- | The population, economic, and crop losses in each province | Annual | Ministry of Water Resources of the People's Republic of China |
| China Meteorological Disaster Yearbook | 2004-- | Time, flooded district, damage of major flood events, the record criteria as events causing over 50,000 hectares of agricultural damage, 10 | Annual | China Meteorological Administration |

| | | deaths, or 14 million USD in direct economic losses | | |
|---|---|---|---|---|
| Reports on official website of China National Disaster Reduction Center | 2011-- | Records of the time, location and damage of flood events (Data prior to 2018 is not available) | Real-time | National Disaster Reduction Center of China |

**Comment 2: The approach used seems quite specific for the Chinese language, using several specifically trained models and training input. It's worthy of discussion of your approach also works for a completely different language group to apply this methodology in other data-scarce regions (e.g. the Global South).**

Our approach was indeed fine-tuned using a Chinese corpus, which means that applying this methodology to a completely different language would require retraining the model with a suitable corpus in that language. This is because BERT models are language-specific, and the fine-tuning process is critical to adapting the model to the nuances of the target language. While the model we trained cannot be directly transferred to regions with different languages, the technical approach we have developed can be applied in any region and serves as a reference.

This point is important, and we have included this statement in the revision (Lines 452-454):

*"On the other hand, BERT models require language-specific fine-tuning, which can limit adaptability across languages. In contrast, LLMs that adopt a zero-shot strategy (i.e., direct application without the need for fine-tuning) may solve the transferability problem."*

**Comment 3: Also, regarding the approach: the media used are all newspaper databases, and only 2 different ones. Why is social media not included, or other sources of information? This seems to limit the potential of the method, since using one type of media source might be fairly uniform in its wording and phrasing, and perhaps not always covering all instances of floods. Furthermore, the restrictive choices on the keywords to select these articles might make the whole model biased: was there any form of testing with broader search terms, synonyms or other idioms for instance (like in L 152)? The model is strongly influenced by the choices for the training data obviously, but it seems to me like some additional testing of the influence of that training data is necessary.**

First, social media data is often non-public and may involve privacy issues, which can impose limitations on its use. On the other hand, due to concerns about data quality control, news articles were selected in the hope of obtaining more accurate results. Data with varying language styles might negatively impact the model's performance.

Regarding the problem of keywords, we tested other keywords as retrieval methods and found that the other keywords included may raise the dataset too large. For example, we tried using "heavy rainfall" as the query term and found that only around 10% news returned reported flood events. Most of these news texts are related to meteorological early warning information. Therefore, the current query was determined to limit the corpus to the most relevant content. Even if the Q&A approach can distinguish between relevant and irrelevant information, the benefits of large corpus are far less than the burden of running the model. The idiom "Floods and beasts" was determined after analyzing CNKI news data, and other intrusive idioms are rarely seen.

We have added more explanation on the keywords selection in Lines 119-125:
*"A total of 2730 news articles from 2000 to 2021 were gathered using the subject keywords ``flood" OR ``flood disaster" and the full-text keywords ``city" OR ``county" OR ``district". Although other meteorology-related terms such as "typhoon," "cyclone," "heavy rainfall," may also be associated with flood events, there were few cases where flood-related news mentioned only flood-causing terms like typhoon. For instance, a separate query using the term ``heavy rain" yielded only about 7\% relevant reports on actual flood events, with the majority of results being meteorological warnings. To ensure a relevant dataset and improve model efficiency, this study limited the search terms to those most directly related to flooding."*

We agree that the model is strongly influenced by the choices for the training data. In constructing the training set, we randomly selected samples rather than using data from consecutive years or a single newspaper source, aiming to help the model learn more diverse features. Moreover, we realized this is an important suggestion, and we have added experiments with different random sample combinations for cross-validation. The method and results of these experiments is in Section 3.2, BERT Model Construction and Application (Lines 214-223) and Section 4.1, The Performance of The BERT Model (Lines 289-293):
*"To evaluate the impact of training data selection on model performance, we also conducted several cross-validation experiments. First, we combined the original training set and validation set to form a comprehensive dataset containing all annotated samples, totaling 503 samples. In each iteration of the cross-validation, we randomly shuffled the comprehensive dataset using different random seeds to ensure data order diversity and experiment reproducibility. Specifically, in five iterations, we set different random seeds (from 0 to 4) and used Python's random module to shuffle the data.*

*After each random shuffle, we selected the first 402 samples (consistent with the size of the original training set) as the new training set. By keeping the training set size consistent, the differences in model performance were solely due to the selection of training data, rather than changes in data size. Then, the BERT model was fine-tuned on each new training set and the Friedman test was used to assess*
*the statistical significance of performance differences between different fine-tuned models."*

*"The cross-validation experiments yielded consistent model performance across different training data selections. Table 5 summarizes the F1-score and EM of*
*flood-information extraction for each model. To statistically assess the differences among the models, we conducted the Friedman test. The test resulted in a p-value of 0.38, indicating that there are no statistically significant differences in performance among the models ($p > 0.05$). This suggests that the model is robust and the training data selection in current study is appropriate."*

**Table 5.** Performance metrics for each cross-validation experiment

| Model | F1-Score | EM |
| --- | --- | --- |
| Model 1 | 85.854 | 81.121 |
| Model 2 | 86.870 | 82.431 |
| Model 3 | 87.017 | 83.617 |
| Model 4 | 85.862 | 81.280 |
| Model 5 | 85.933 | 81.423 |

**Comment 4: Reading through the methodology it seems like a lot of manual preprocessing is still required, including manually annotating news texts. Ow much of a bottleneck is that for operational purposes is that, if you really**
**will have a constantly updating database? This requires some discussion since it directly impacts the applicability of this dataset.**

Thank you for raising this important point. The manual preprocessing, including the annotation of news texts, is primarily required during the initial fine-tuning of the
model, as well as for adjusting hyperparameters. This step is essential for ensuring the accuracy and effectiveness of the model. However, once the model has been trained, it does not need to be retrained for future applications. Future data can be directly processed into the test set format and used as input for the model without additional manual preprocessing. Therefore, this process does not represent a
significant problem for the operational application of a constantly updating database.

The relevant explanation was added in Lines 166-168:
*"For the WiseNews data, it only needs to be formatted as a test set for direct*

*application with the trained model, without requiring manual answer annotations. Future data will follow the same process, further enhancing analytical efficiency."*

**Comment 5: L 235: the exclusion of any texts wit the word 'will' seems like it can introduce giant margins of error. I get the reasoning to exclude forecasts,**
**but if 'will' is used in a different context in a text that is actually related to flooding (e.g. 'damaged roads will re-open in 4 days') is then the whole text still excluded?**

This screening measure is based on BERT's answer to what disaster event
happened, which is a classification of the disaster events described in the news. For example, the statement you mentioned, 'damaged roads will re-open in 4 days,' is related to the impact of the event and would not typically appear in the answer content. The responses to the Question 1 are focused on the type of the event, and they are usually very brief and do not include the details of the event. Therefore,
this exclusion will not introduce big biases.

**Comment 6: The choice of GDP as a clustering method is odd to me. Why not use population density instead? That does correlate somewhat with GDP (so you still get reports on economic losses) but the loss of human life also**
**hugely matters in disaster reporting, I'd think.**

Our initial motivation for conducting the GDP clustering analysis was to explain how regional economic development might influence the biases in media data. However, after carefully considering the reviewers' comments and reviewing
literature on media communication themes, we have decided to remove this section. Relying solely on economic development or population density to explain the biases in media data is not convincing enough. In the revised version, we modified our explanation of the biases introduced by media data in Lines 302-317: *"Identifying specific biases is challenging because the China Flood and Drought*
*Bulletin provides only the total number of flooded cities, without listing inventory of specific locations. This limitation makes it impossible to pinpoint which specific events or regions are underreported in our dataset. However, we can hypothesize that the biases stem from the intrinsic characteristics of news data.*
*Some previous studies have also reflected the bias of constructing disaster*
*catalogs with reports (Gall et al., 2009; Delforge et al., 2023). From the perspective of media communication studies, agenda-setting theory posits that by choosing which events to report on, the media effectively signals to the public which issues are important (Leidecker-Sandmann et al., 2023). Through the quantity and depth of coverage, the media can shape the level of public attention given to certain*
*events. In the context of disaster reporting, the government may influence the direction of media coverage to control public attention on specific disasters (Bai, 2022). For example, during the COVID-19 pandemic, research on government crisis communication showed that media agenda-setting was significantly*

*influenced by government press conferences (Hayek, 2024). Crisis communication theory further explains the government can swiftly steer public opinion in the aftermath of a disaster, reducing the spread of negative emotions and maintaining social stability (Zhou et al., 2023). As a result, the variability in disaster reporting by the media may be influenced by multiple factors, including government policies, public interest, and the media's own resource allocation, leading to a situation where the volume of media reports is not necessarily consistent with the actual number of disaster events."*

Moreover, we added the analysis of the flood trend in different population density and economically developed areas to provide insights from an urban and social perspective in Lines 371-386:

*"In addition, to analyze the relationship between flood changes and societal characteristics, Figure 7 uses average annual Gross Regional Product (GRP) in billion USD and population density in people per square kilometer as base maps to display the distribution of flood trends across different regions, with darker shades indicating higher values. Overall, most provinces exhibit an increasing trend in flood events, particularly in the northern, and western regions of China. These areas, including provinces such as Heilongjiang, Shandong, and Chongqing, are characterized by varying levels population density, both higher and lower, according to Figure 7(a). The provinces that exhibit a decreasing trend in flood events are primarily located in the central and southeastern regions, particularly in provinces like Jiangsu, Fujian, and Guangdong, which are notable for their higher population densities. This suggests that the rising flood events are not strictly tied to population density.*

*As for the trends in relation to economic output in Figure 7(b), the provinces with increasing flood trends are mostly those with lower to moderate GRP, such as those in the northern and western parts of China, despite Shandong and Zhejiang. These regions may not have received the same level of economic investment in flood control infrastructure as the more developed eastern provinces, which might explain the rising trend in flood events. On the other hand, the central and eastern provinces showing a decreasing trend, such as Jiangsu, Guangdong, and Sichuan, are among the most economically developed in China. This suggests that the availability of economic resources has allowed for more comprehensive flood management strategies, reducing the frequency of flood events in these areas.*

[Figure]

*Figure 7. The analysis of flood event trends across Chinese provinces from 2000 to 2022, shown in relation to (a) population density and (b) Gross Regional Product (GRP)."*

**Comment 7: Figure 6: This figure doesn't seem too relevant to the paper to warrant inclusion. A typhoon is certainly going to lead to flooding but the spatial scale is so wide that it's not a great verification in my opinion.**

We also agree that typhoons undoubtedly cause flooding, but the spatial scale is indeed quite large. Our intention in using Figure 6 was to demonstrate that our dataset successfully identifies disaster areas affected by typhoons, serving as evidence that our dataset can capture events with a broad impact. When considering the biases inherent in news data, a review of other literature revealed that the variability in disaster reporting by the media may be influenced by multiple factors, including government policies, public interest, and the media's own resource allocation, leading to a situation where the volume of media reports is not necessarily consistent with the actual number of disaster events. We speculate that this bias in media data is mainly due to the neglect of smaller-scale or less severe events. Therefore, we used these two significant cases to illustrate that larger-scale events are still likely to be reported.

However, we understand that this figure may not adequately demonstrate the accuracy or completeness of our data. After considering all the reviewers' comments, we decided to remove this figure.

**Comment 8: Figures 8 and 9: occurrence is here shown without any distinction of severity of flooding, whereas the latter one might be more relevant for actual use of the dataset.**

Thank you for your suggestions regarding Figures 8 and 9. These figures represent the heatmaps of flood occurrences and the number of flood-related news reports by year and month. The primary purpose is to illustrate the temporal distribution of flood events across China. These visualizations help to highlight the years and

seasons during which floods are most frequent, offering insights into the timing of flood events over the study period.

We acknowledge that distinguishing the severity of flooding could enhance the relevance of the dataset for certain applications. However, our current dataset does not provide detailed information on the severity of the flooding. Because this information is expressed with more variability and is more unstructured across different news sources, we will try to re-train the model or introduce new methods for extracting this type of information in future research.

**Comment 9: L230: I don't understand what the authors mean with '3 epochs' and a learning rate of 5 x 10^-5. Please elaborate.**

The term "3 epochs" refers to the number of times the entire training dataset is passed through the model during the training process. In our case, we trained the model for 3 epochs, meaning the dataset was fed into the model three times, which is a standard approach to ensure that the model learns the patterns effectively without overfitting.

The learning rate of 5 x 10^-5 is a hyperparameter that controls the step size at each iteration while moving toward a minimum of the loss function. A smaller learning rate, such as the one we used, allows the model to converge more slowly and steadily, reducing the risk of overshooting the optimal parameters. This value was chosen based on preliminary experiments to balance the learning speed and model performance.

We have added the explanation to make them more understandable in Lines 199-203:
*"The BERT-base model was fine-tuned for three epochs (i.e., the number of times the entire training dataset is passed through the model during the training process) with a learning rate (i.e., a hyperparameter that controls the step size at each iteration while moving toward a minimum of the loss function) of 5 x 10^-5 and a batch size (i.e., the number of training examples used in one iteration of the model) of 8, which were determined to be the most effective combination among the tested settings."*

**Comment 10: L 275: 'verify and revise': is this part of the preprocessing? What does this mean, exactly?**

The process of "verifying and revising" is not part of the preprocessing but rather a post-processing step. The location names are generated by the BiLSTM-CRF model, and since the data spans from 2000 to 2022, it includes periods during which several regions in China underwent administrative adjustments or renaming. To ensure accuracy and relevance when associating these locations with the administrative division shapefile for spatial visualization in ArcGIS, we updated the names to reflect the most current administrative divisions. This step was crucial for maintaining consistency and ensuring that the visualizations accurately represent the latest geographical boundaries.

We modified this part to make it clearer in Lines 237-243:
*"After identifying the flood locations, it was essential to verify and revise the list of places in accordance with the latest national administrative divisions. Because the data spans from 2000 to 2022, it includes periods during which several regions in China underwent administrative adjustments or renaming. To ensure accuracy and relevance when associating these locations with the administrative division shapefile for spatial visualization in ArcGIS, the changed names of districts or counties should be checked to reflect the current administrative divisions. This step was crucial for maintaining consistency and ensuring that the visualizations accurately represent the latest geographical boundaries. After that, flood locations were matched with the administrative division shape file and visualized using ArcGIS."*

**Comment 11: Figure 4: Any idea what causes the large biases? This is hardly discussed.**

Identifying specific biases is challenging because the *China Flood and Drought Bulletin* provides only the total number of flooded cities, without listing specific locations. This limitation makes it impossible to pinpoint which specific events or regions are underreported in our dataset. As the response to Comment 6, we modified the discussion in Lines 304-317:
*"However, we can hypothesize that the biases stem from the intrinsic characteristics of news data. Some previous studies have also reflected the bias of constructing disaster catalogs with reports (Gall et al., 2009; Kron et al., 2012). From the perspective of media communication studies, agenda-setting theory posits that by choosing which events to report on, the media effectively signals to the public which issues are important (Leidecker-Sandmann et al., 2023). Through the quantity and depth of coverage, the media can shape the level of public attention given to certain events. In the context of disaster reporting, the government may influence the direction of media coverage to control public attention on specific disasters (Bai, 2022). For example, during the COVID-19 pandemic, research on government crisis communication showed that media agenda-setting was significantly influenced by government press conferences (Hayek, 2024). Crisis communication theory further explains the government can swiftly steer public opinion in the aftermath of a disaster, reducing the spread of negative emotions and maintaining social stability (Zhou et al., 2023). As a result, the variability in disaster reporting by the media may be influenced by multiple factors, including government policies, public interest, and the media's own resource allocation, leading to a situation where the volume of media reports is not*

*necessarily consistent with the actual number of disaster events."*

Although our data consistently underestimates the number of flooded cities each year, likely due to the influence of the factors mentioned above, the trend in our data closely follows the overall temporal pattern observed in the *China Flood and Drought Bulletin*. This suggests that our dataset can still be effectively used to explore variations in flood events across regions. Then, it can be leveraged to analyze potential influencing factors of these changes, such as socioeconomic changes, climate change, alterations in land surface characteristics, and modifications in flood control measures, ultimately providing recommendations for flood management.

On the other hand, in future research, we plan to incorporate more available data sources to continuously update and validate this dataset. By expanding the dataset's coverage and adding descriptions of damages, its comprehensiveness will be improved.

**Dear Referee #3,**

Thank you very much for your time involved in reviewing the manuscript. Those comments are constructive for improving our manuscript. All the revisions related to your comments are noted in underline in the marked-up manuscript. Our responses to your comments are as follows:

**Comment 1:The authors provide a comprehensive introduction to existing natural disaster datasets that record flood events created by official sources, other governments, or organizations. However, the manuscript would benefit from a more detailed discussion on how this study specifically addresses the gaps in these existing datasets. It is essential to clearly state the novelty and significance of your work in the context of existing datasets. For instance, do the deficiencies in these existing datasets affect the analysis, modeling, and prediction of flood events to some extent? How does the new dataset you have developed alleviate these issues at both theoretical and practical application levels?**

Thanks for this suggestion. We have added more statements of the novelty and significance of our dataset in the revision:

In Section 1, Introduction (Lines 60-66):

*"Existing datasets provide valuable insights into urban flooding; however, they often operate at the provincial level or have limited event coverage across China. This broader scale can obscure important local variations in flood characteristics and risk factors. In China, the administrative structure consists of several levels: the highest is the provincial level, followed by the prefectural level (i.e., cities in*

*the usual sense), and then county-level. Given China's vast area of approximately 9.6 million square kilometers, flood characteristics exhibit significant spatial variability across provincial and prefectural regions (Wang et al., 2013; Shang et al., 2023). With around 2,844 county-level areas, each spanning roughly 1,000-3,000 square kilometers, this scale offers a more granular perspective for analyzing flood patterns across diverse locations."*

In Section 5.2, Main Findings (Lines 391-397):

*"The dataset created in this study serves as the first county-level urban flood inventory across China from 2000, addressing a gap in existing datasets that often fail to provide county-level flood distributions or coverage across the country. While China Flood and Drought Bulletin offers authoritative data on flood disasters, focusing on economic losses, casualties, and agricultural damages at the provincial level, it lacks detailed inventories for specific cities. Our dataset*
*shows trends that are largely consistent with those reported in the China Flood and Drought Bulletin, indicating that it can reliably reflect changes in flood events across different regions. This dataset's county-level granularity also allows for resampling and aggregation to city or provincial levels, facilitating deeper analyses of flood dynamics and influencing factors at different spatial scales."*

**Comment 2:Line 143 "After a manual review to remove duplicates and irrelevant entries, including those referring to flash floods which occur suddenly in mountainous areas and are not the focus of this study, the final dataset consisted of 253 relevant news articles". The data preparation section needs more details. Please explain the criteria used for manually**
**reviewing and removing irrelevant news articles from the CNKI database. Additionally, discuss any potential biases or limitations introduced by this manual selection process.**

We have added more explanation in Lines 126-130 and the discussion of biases in Lines 425-427:

*"Once the CNKI data was collected, duplicate reports of the same urban flood events from different regional newspapers were manually removed by the lead author. Articles containing search keywords but focused on flood prevention measures, seasonal warnings, or other non-flood-related topics were also excluded. Additionally, as the focus of this study is on urban flooding, reports*
*concerning flash floods and landslides were omitted. The final dataset, following manual review and verification by researchers in the group, consisted of 253 relevant news articles."*

*"Although thorough double-checking was conducted during the data preparation phase, the possibility of biases remains due to subjective differences in*
*interpretation. Future research could incorporate language models for correlation*

*analysis or involve more domain experts to cross-validate the accuracy of the results."*

**Comment 3:Similarly, Line145 "These relevant news articles were then segmented into paragraphs and reorganized into 633 distinct samples. Among them, 503 samples were used to fine-tune the BERT model, alongside data from the CMRC2018 dataset, enhancing the model's stability to accurately extract flood disaster information. The remaining 130 samples served as a test set to evaluate the model's performance." Please clarify how the 503 samples were selected from the 633 distinct samples, and explain why the remaining 130 samples were used to evaluate the model's performance. This selection process is currently unclear and confusing.**

Thank you for your insightful feedback. Regarding the selection of the 503 samples from the 633 distinct samples, we would like to clarify that this process was done through random sampling, without any subjective selection. The intent behind this random division was to ensure that the training and testing datasets were representative of the entire dataset, thereby minimizing any potential bias.

The remaining 130 samples were designated as the test set after the training samples were randomly selected, without any manual intervention or predefined criteria. This allows for an independent evaluation of the model's ability to generalize to new, unseen data.

We have modified the description of data preparation in Lines 160-163:
*"The CNKI news articles were then divided into 633 distinct samples. Of these, 503 were randomly selected as training samples, and the remaining 130 samples were set aside as test samples to evaluate the model performance. For training, 80\% of the samples (402) were combined with the CMRC2018 dataset to fine-tune the BERT model, while 20\% (101) were used for validation to optimize the model's hyperparameters."*

**Comment 4:For the identification of flood locations, I have a general question. From my understanding, news media reports about flooding occurrences typically mention the affected city or, at most, the district. However, actual urban flooding can occur at the street level or even smaller scales. Could you please provide a detailed explanation of how the BiLSTM-CRF model was trained and applied to recognize flood locations?**

Thank you for raising this important point. It is indeed true that news reports often only mention the affected city or district, with only some reports specifying smaller-scale locations like streets or buildings. However, our goal is to obtain results at the county or district level rather than more granular details.

To clarify our approach, we used the BERT model to identify the flood-affected areas mentioned in each news report firstly. The answer sentences typically include cities, counties or districts, and in some cases, specific streets or buildings. After obtaining these mentions, the BiLSTM-CRF model was employed to extract the pure location names from BERT's output.

The BiLSTM-CRF model was trained using the MSRA named entity recognition corpus, a widely used dataset developed by Microsoft Research Asia, which contains a large amount of annotated Chinese sentences with named entities such as location names, person names, and organization names. Then, we standardized the spatial information by using county/district names (in China, counties and districts are the same administrative level and both included in cities).

**Comment 5:Regarding the performance of the BERT model (Table 4), it appears that the authors have only examined results based on a binary classification (flood vs. non-flood). If this is the case, the task seems too simple and lacks sufficient novelty. Could the authors also provide an evaluation of the model's performance in identifying the time and location of flood events?**

Table 4 does not only present the results of categorizing events as flood or non-flood. Only the first row is for classification. The second row shows the evaluation results of spatiotemporal information extraction. Two evaluation systems were used for the recognition of time and location information. This is explained in Section 3.4, Evaluation Metrics (Lines 260-270):

*"The first index is called exact match (EM), which measures the matching degree between the prediction and ground truths. The score is 1 for the EM of both the time and location information extracted. Otherwise, the score is 0. There is usually more than one disaster location in one flood event and maybe the model can output several but not completely accurate locations. Therefore, a fuzzy match was used to evaluate the location extraction using precision, recall, and F1 score. Unlike the classical formula, the precision and recall were calculated as:*

$$Precision = \frac{P}{M}$$

$$Recall = \frac{P}{N}$$

*Where P represents the number of accurately extracted flood locations, M is the total number of predicted flood locations and N is the total number of actual flood locations observed in the texts."*

**Table 4.** The performance of the BERT model (EM index was not applied to evaluate event identification)

|  | Precision | Recall | F1-score | EM |
|---|---|---|---|---|
| Flood-event identification | 0.98 | 0.98 | 0.98 | N/A |
| Flood-information extraction | 0.96 | 0.78 | 0.86 | 0.82 |

**Comment 6:It seems that the number of identified flooded cities is significantly underestimated by the news media compared to the China Flood and Drought Bulletin (Figure 4). The authors suggest this discrepancy is related to the low attention given to low GDP areas.**

**However, this raises a significant concern about the reliability of the developed dataset. As mentioned in section 4.3, the dataset records urban flood events reported in news articles from 2000 to 2022. If the news media is so inaccurate that it fails to record a large number of flood events, how can the authors ensure the reliability of the data generated from these news**

**sources?**

We visualized the year-on-year difference in the number of flooded cities between the two datasets to provide a more intuitive representation of interannual variations. As shown in the figure below, the trend in our data is largely consistent with that reported in the Bulletin. This suggests that our dataset can reliably reflect changes in flood events across different regions, though news media may underestimate the number of cities affected by floods.

[Figure]

On the other hand, while the *China Flood and Drought Bulletin* provides a summary of the number of flooded cities, it does not offer a detailed inventory of specific locations. The spatial distribution of flood disaster loss information in the bulletin is limited to the province level, which encompasses multiple city-level areas. There may be biases inherent in the news data, but we contend that our dataset serves as a valuable reference in the absence of more detailed and comprehensive data sources.

In the future, we will introduce more data source to improve the data coverage, such as social media data, available disaster reports in some cities or provinces and so on.

We also added the explanation in Lines 297-301:

*"In addition, the year-on-year difference in the number of flooded cities between the two datasets was visualized to provide a more intuitive representation of interannual variations. As shown in Figure 4(b), the trend of the news-based dataset closely follows the overall temporal pattern observed in the China Flood and Drought Bulletin. This suggests that the dataset created in this present study can reliably reflect changes in flood events across different regions, though news*
*media consistently underestimate the number of cities affected by floods."*

**Comment 7:Figure 6 is not directly related to your results, I think you can put it into supplementary materials.**

We have considered all the reviewers' comments and have ultimately decided to remove this figure and the related statements from the manuscript.

**Dear Referee #4,**

Thank you very much for your time involved in reviewing the manuscript. All the
revisions related to your comments are noted in highlight in the marked-up manuscript. Our responses to your comments are as follows:

**Comment 1: It's a bit surprising that this work is still based on BERT and doesn't mention anything about the emerging large language model (LLM)**
**techniques (e.g., GPT-4). Please comment on this choice and discuss potential improvements if newer techniques could be used.**

When we designed this study, emerging LLMs had not yet developed to their current impressive state. Later, we compared our approach with GPT-3.5, and found that the accuracy of information extraction from the same test corpus was
similar. On the other hand, considering this might be a project requiring long-term maintenance, we decided to continue using BERT, an open-source model that performs adequately, rather than switching to another model.

Moreover, we added the discussion of potential improvements if newer techniques could be used in Lines 443-454:

*"Furthermore, the approach used in the present study also has its limitations. We employed a BERT model fine-tuned by a Chinese corpus for question-answering tasks, which has proven efficient in information extraction. However, with the rapid advancement of large language models (LLMs), newer models such as*

*GPT series offer significant improvements in natural language processing tasks. For example, Colverd et al. (2023) successfully used several LLMs, including GPT-3.5, GPT-4, and PaLM-Text-Bison, to generate flood disaster impact reports by extracting information from the web, finding strong correlations between LLM-generated and human-authored reports. Additionally, Hu et al. (2023) proposed a method that combines geospatial knowledge with GPT models to extract location descriptions from disaster-related social media posts, achieving a 40\% improvement over traditional NER approaches. Given these advancements, future research should explore the use of LLMs to extract nuanced information from flood-related text data, which includes distinguishing flood types, causes, and the specific losses associated with each flooding event. On the other hand, BERT models require language-specific fine-tuning, which can limit adaptability across languages. In contrast, LLMs that adopt a zero-shot strategy (i.e., direct application without the need for fine-tuning) may solve the transferability problem."*

**Comment 2: Given the focus of this dataset on cities, the analysis of the contributed dataset seems somewhat less pertinent. For instance, the large-scale climate zone analysis is rather off-topic. Instead, one would expect to see if such a dataset could be linked with urban-specific features (e.g., built-up area, urban volumetric density, GDP) to reveal more city-scale findings.**

We agree that our dataset should be more linked with urban-specific features. We have added the following analysis in Section Spatial Distribution of Flood Events (Lines 371-386):

*"In addition, to analyze the relationship between flood changes and societal characteristics, Figure 7 uses average annual Gross Regional Product (GRP) in billion USD and population density in people per square kilometer as base maps to display the distribution of flood trends across different regions, with darker shades indicating higher values. Overall, most provinces exhibit an increasing trend in flood events, particularly in the northern, and western regions of China. These areas, including provinces such as Heilongjiang, Shandong, and Chongqing, are characterized by varying levels population density, both higher and lower, according to Figure 7(a). The provinces that exhibit a decreasing trend in flood events are primarily located in the central and southeastern regions, particularly in provinces like Jiangsu, Fujian, and Guangdong, which are notable for their higher population densities. This suggests that the rising flood events are not strictly tied to population density.*

*As for the trends in relation to economic output in Figure 7(b), the provinces with increasing flood trends are mostly those with lower to moderate GRP, such as those in the northern and western parts of China, despite Shandong and Zhejiang. These regions may not have received the same level of economic*

*investment in flood control infrastructure as the more developed eastern provinces, which might explain the rising trend in flood events. On the other hand, the central and eastern provinces showing a decreasing trend, such as Jiangsu, Guangdong, and Sichuan, are among the most economically developed in China. This suggests that the availability of economic resources has allowed for more*
*comprehensive flood management strategies, reducing the frequency of flood events in these areas.*

[Figure]

*Figure 7. The analysis of flood event trends across Chinese provinces from 2000 to 2022, shown in relation to (a) population density and (b) Gross Regional*
*Product (GRP)."*

**Minor Comment 1: Line 375: "Lanzhou Province" - Lanzhou is \*\*not\*\* a province but the capital city of Gansu Province.**

Sorry for this mistake. We have addressed this issue in Lines 348-349.

**Minor Comment 2: The dataset should be archived more appropriately**
**following the FAIR principle as suggested by reviewer 1. In addition, the GitHub repo needs more necessary README info, such as a description of the dataset, citation, etc. Also, `xlsx` is not recommended for simple tabular formats—please consider publishing this dataset in `csv` for better accessibility to allow better open research.**

Thanks for this suggestion and we have made the modifications as follows:
(1) We have changed the dataset sharing website to Zenodo, which is an open-access repository that allows researchers to share and preserve their datasets. It is operated by CERN and OpenAIRE and provides DOI for citations, which supports the FAIR principles. The DOI of our dataset is:
10.5281/zenodo.14000094
(2) We have improved our README information to describe the dataset including the data source, the data resolution, time span, and so on. The current data description is as follows:

*"This dataset is a catalog of urban floods in China from 2000 to 2022. The*

*data is sourced from Chinese news text, with the BERT model used to extract information on the timing and location of flood events. The temporal scale is daily, with entries showing only the month for cases where specific dates could not be extracted. The spatial scale is at the county level in China. Details of the data collection and creation process are thoroughly explained in the relevant*

*paper. Each year's flood events are stored in a CSV file named after that year, containing the following fields:*

 *Year: the year the event occurred;*

 *PAC: the administrative code for the county where the event occurred;*

 *Province: the province of the affected area;*

 *City: the city of the affected area;*

 *County: the county where the event occurred;*

 *Occurrence: the number of times floods occurred in that area during the year;*

 *Time: the specific date of each flood event in that area.*

*The ".shp," ".shx," and ".dbf" files are shapefiles of China's administrative regions used by the author to visualize the data."*

(3) We have changed the data files into 'csv' format.